# Dissecting Hessian: Understanding Common Structure of Hessian in Neural Networks

**Yikai Wu**[*]
Department of Computer Science
Duke University
Durham, NC 27708
yikai.wu@duke.edu

**Xingyu Zhu**[*]
Department of Computer Science
Duke University
Durham, NC 27708
xingyu.zhu@duke.edu

**Chenwei Wu**
Department of Computer Science
Duke University
Durham, NC 27708
chenwei.wu592@duke.edu

**Annie Wang**
Department of Computer Science
Duke University
Durham, NC 27708
annie.wang029@duke.edu

**Rong Ge**
Department of Computer Science
Duke University
Durham, NC 27708
rongge@cs.duke.edu

## Abstract

Hessian captures important properties of the deep neural network loss landscape. Previous works have observed low rank structure in the Hessians of neural networks. We make several new observations about the top eigenspace of layer-wise Hessian – top eigenspaces for different models have surprisingly high overlap, and top eigenvectors form low rank matrices when they are reshaped into the same shape as the corresponding weight matrix. Towards formally explaining such structures of the Hessian, we show that the new eigenspace structure can be explained by approximating the Hessian using Kronecker factorization; we also prove the low rank structure for random data at random initialization for over-parametrized two-layer neural nets. Our new understanding can explain why some of these structures become weaker when the network is trained with batch normalization. The Kronecker factorization also leads to better explicit generalization bounds.

## 1 Introduction

The loss landscape for neural networks is crucial for understanding training and generalization. In this paper we focus on the structure of Hessians, which capture important properties of the loss landscape. For optimization, Hessian information is used explicitly in second order algorithms, and even for gradient-based algorithms properties of the Hessian are often leveraged in analysis (Sra et al., 2012). For generalization, the Hessian captures the local structure of the loss function near a local minimum, which is believed to be related to generalization gaps (Keskar et al., 2017).

---

[*]Contributed equally, listed in alphabetical order

Several previous results including (Sagun et al., 2018; Papyan, 2018) observed interesting structures in Hessians for neural networks – it often has around $c$ large eigenvalues where $c$ is the number of classes. In this paper we ask:

*Why does the Hessian of neural networks have special structures in its top eigenspace?*

A rigorous analysis of the Hessian structure would potentially allow us to understand what the top eigenspace of the Hessian depends on (e.g., the weight matrices or data distribution), as well as predicting the behavior of the Hessian when the architecture changes.

Towards this goal, we first focus on the structure for the top *eigenspace* of layer-wise Hessians. We observe that the top eigenspace of Hessians are far from random – models trained with different random initializations still have a large overlap in their top eigenspace, and the top eigenvectors are close to rank 1 when they are reshaped into the same shape as the corresponding weight matrix.

We formalize a conjecture that allows us to understand all these structures using a Kronecker decomposition. We also analyze the Hessian in an over-parametrized two-layer neural network for random data, proving that the output Hessian is approximately rank $c - 1$ and its top eigenspace can be easily computed based on weight matrices.

## 1.1  Our Results

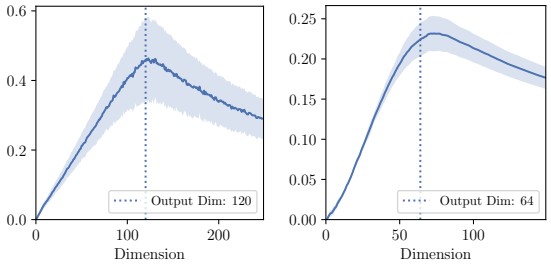
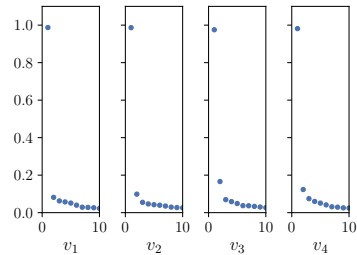

(a) Overlap between dominate eigenspace of layer-wise Hessian at different minima for fc1:LeNet5 (**left**) with output dimension 120 and conv11:ResNet18-W64 (**right**) with output dimension 64.

(b) Top 10 singular values of the top 4 eigenvectors of the layer-wise Hessian of fc1:LeNet5 after reshaped as matrix.

Figure 1: Some interesting observations on the structure of layer-wise Hessians. The eigenspace overlap is defined in Definition 4.1 and the reshape operation is defined in Definition 4.2

**Structure of Top Eigenspace for Hessians:** Consider two neural networks trained with different random initializations and potentially different hyper-parameters; their weights are usually nearly orthogonal. One might expect that the top eigenspace of their layer-wise Hessians are also very different. However, this is surprisingly false: the top eigenspace of the layer-wise Hessians have a very high overlap, and the overlap peaks at the dimension of the layer's output (see Fig. 1a). Another interesting phenomenon is that if we express the top eigenvectors of a layer-wise Hessian as a matrix with the same dimensions as the weight matrix, then the matrix is approximately rank 1. In Fig. 1b we show the singular values of several such reshaped eigenvectors.

**Understanding Hessian Structure using Kronecker Factorization:** We show that both of these new properties of layer-wise Hessians can be explained by a Kronecker Factorization. Under a decoupling conjecture, we can approximate the layer-wise Hessian using the Kronecker product of the output Hessian and input auto-correlation. This Kronecker approximation directly implies that the eigenvectors of the layer-wise Hessian should be approximately rank 1 when viewed as a matrix. Moreover, under stronger assumptions, we can generalize the approximation for the top eigenvalues and eigenvectors of the full Hessian.

**Structure of auto-correlation:** The auto-correlation of the input is often very close to a rank 1 matrix. We show that when the input auto-correlation component is approximately rank 1, the layer-wise Hessians indeed have high overlap at the dimension of the layer's output, and the spectrum of the layer-wise Hessian is similar to the spectrum of the output Hessian. On the contrary, when the model is trained with batch normalization, the input auto-correlation matrix is much farther from rank 1 and the layer-wise Hessian often does not have the same low rank structure.

**Structure of output Hessian:** For the output Hessian, we prove that in an over-parametrized two-layer neural network on random data, the Hessian is approximately rank $c - 1$. Further, we can compute the top $c - 1$ eigenspace directly from weight matrices. We show that this calculation can be extended to multiple layers and the result has a high overlap with the actual top eigenspace of Hessian in most settings.

**Applications:** As a direct application of our results, we show that the Hessian structure can be used to improve the PAC-Bayes bound computed in Dziugaite & Roy (2017).

## 2 Related Works

**Hessian-based analysis for neural networks (NNs):** Hessian matrices for NNs reflect the second order information about the loss landscape, which is important in characterizing SGD dynamics (Jastrzebski et al., 2019) and related to generalization (Li et al., 2020), robustness to adversaries (Yao et al., 2018) and interpretation of NNs (Singla et al., 2019). People have empirically observed several interesting phenomena of the Hessian, e.g., the gradient during training converges to the top eigenspace of Hessian (Gur-Ari et al., 2018; Ghorbani et al., 2019), and the eigenspectrum of Hessian contains a "spike" which has about $c - 1$ large eigenvalues and a continuous "bulk" (Sagun et al., 2016, 2018; Papyan, 2018).

**Understanding structures of Hessian:** People have developed different frameworks to explain the low rank structure of the Hessians including hierarchical clustering of logit gradients (Papyan, 2019, 2020), independent Gaussian model for logit gradients (Fort & Ganguli, 2019), and Neural Tangent Kernel (Jacot et al., 2020). These results work in different settings but are not directly comparable (among themselves and our paper). A distinguishing feature of this work is that we can characterize the top eigenspace of the Hessian directly by the weight matrices of the network.

**Theoretical analysis for large eigenvalues of Fisher Information Matrices (FIM):** The FIM can be seen as a component of the Hessian matrix for neural networks. Karakida et al. (2019b) showed that the largest $c$ eigenvalues of the FIM for a randomly initialized neural network are much larger than the others. Their results rely on the eigenvalue spectrum analysis in Karakida et al. (2019c,a), which assumes the weights used during forward propagation are drawn independently from the weights used in back propagation (Schoenholz et al., 2017).

**Layer-wise Kronecker factorization for training NNs:** The idea of approximating the FIM using Kronecker factorizations can be dated back to Heskes (2000). More recently Martens & Grosse (2015) proposed Kronecker-factored approximate curvature (K-FAC) which approximates the inverse of FIM using layer-wise Kronecker product and perform approximated natural gradient descent (NGD) in training NNs. Kronecker factored eigenbasis has also been utilized (George et al., 2018). K-FAC has been generalized to convolutional and recurrent NNs (Grosse & Martens, 2016; Martens et al., 2018), Bayesian deep learning (Zhang et al., 2018), and structured pruning (Wang et al., 2019). Unlike these previous works which focus on accelerating computations, in this paper we use Kronecker factorization to explain the structures of Hessians.

**PAC-Bayes generalization bounds:** People have established generalization bounds for neural networks under PAC-Bayes framework by McAllester (1999). This bound was further tightened by Langford & Seeger (2001), and Catoni (2007) proposed a faster-rate version. For neural networks, Dziugaite & Roy (2017) proposed the first non-vacuous generalization bound, which used PAC-Bayesian approach with optimization to bound the generalization error for a stochastic neural network. Their bound was then extended to ImageNet scale by Zhou et al. (2019) using compression.

## 3 Preliminaries and Notations

**Basic Notations:** In this paper, we generally follow the default notation suggested by Goodfellow et al. (2016). Additionally, for a matrix $\boldsymbol{M}$, let $\|\boldsymbol{M}\|_F$ denote its Frobenius norm and $\|\boldsymbol{M}\|$ denote its spectral norm. For two matrices $\boldsymbol{M} \in \mathbb{R}^{a_1 \times b_1}, \boldsymbol{N} \in \mathbb{R}^{a_2 \times b_2}$, let $\boldsymbol{M} \otimes \boldsymbol{N} \in \mathbb{R}^{(a_1 a_2) \times (b_1 b_2)}$ be their Kronecker product such that $[\boldsymbol{M} \otimes \boldsymbol{N}]_{(i_1-1) \times a_2 + i_2, (j_1-1) \times b_2 + j_2} = \boldsymbol{M}_{i_1, i_2} \boldsymbol{N}_{j_1, j_2}$.

**Neural Networks:** We consider classification problems with cross-entropy loss. For a $c$-class classification problem, we are given a collection of training samples $S = \{(\boldsymbol{x}_i, \boldsymbol{y}_i)\}_{i=1}^N$ where $\forall i \in [N], (\boldsymbol{x}_i, \boldsymbol{y}_i) \in \mathbb{R}^d \times \mathbb{R}^c$. We assume $S$ is i.i.d. sampled from the underlying data distribution $\mathcal{D}$.

Consider an $L$-layer fully connected ReLU neural network without skip connection $f_{\boldsymbol{\theta}} : \mathbb{R}^d \to \mathbb{R}^c$. With $\sigma(x) = x\mathbf{1}_{x \geq 0}$ as the Rectified Linear Unit (ReLU) function, the output of this network is a series of logits $\boldsymbol{z} \in \mathbb{R}^c$ computed recursively as

$$\boldsymbol{z}^{(p)} := \boldsymbol{W}^{(p)}\boldsymbol{x}^{(p)} + \boldsymbol{b}^{(p)}. \tag{1}$$

$$\boldsymbol{x}^{(p)} := \sigma(\boldsymbol{z}^{(p)}). \tag{2}$$

Here $\boldsymbol{x}^{(p)}$ and $\boldsymbol{z}^{(p)}$ are called the input and output of the $p$-th layer, and we set $\boldsymbol{x}^{(1)} = \boldsymbol{x}$, $\boldsymbol{z} := f_{\boldsymbol{\theta}}(\boldsymbol{x}) = \boldsymbol{z}^{(L)}$. We denote $\boldsymbol{\theta} := (\boldsymbol{w}^{(1)}, \boldsymbol{b}^{(1)}, \boldsymbol{w}^{(2)}, \boldsymbol{b}^{(2)}, \cdots, \boldsymbol{w}^{(L)}, \boldsymbol{b}^{(L)}) \in \mathbb{R}^P$ the parameters of the network. In particular, $\boldsymbol{w}^{(i)}$ is the flattened $i$-th layer weight coefficient matrix $\boldsymbol{W}^{(i)}$ and $\boldsymbol{b}^{(i)}$ is its bias vector. For convolutional networks, a similar analogue is presented in Appendix A.2.

For a single input $\boldsymbol{x} \in \mathbb{R}^d$ with label $\boldsymbol{y}$ and logit output $\boldsymbol{z}$, let $n^{(p)}$ and $m^{(p)}$ be the lengths of $\boldsymbol{x}^{(p)}$ and $\boldsymbol{z}^{(p)}$. For convolutional layers, we consider the number of output channels as $m^{(p)}$ and width of unfolded input as $n^{(p)}$. Note that $\boldsymbol{x}^{(1)} = \boldsymbol{x}, \boldsymbol{z}^{(L)} = \boldsymbol{z} = f_{\boldsymbol{\theta}}(\boldsymbol{x})$. We also denote $\boldsymbol{p} := \mathrm{softmax}(\boldsymbol{z}) = e^{\boldsymbol{z}} / \sum_{i=1}^c e^{\boldsymbol{z}_i}$ the output confidence.

With the loss function $\ell(\boldsymbol{p}, \boldsymbol{y}) = -\sum_{i=1}^c \boldsymbol{y}_i \log(\boldsymbol{p}_i) \in \mathbb{R}^+$ being the cross-entropy loss between the softmax of logits $\boldsymbol{z} = f_{\boldsymbol{\theta}}(\boldsymbol{x}_i) \in \mathbb{R}^c$ and the one-hot label $\boldsymbol{y} \in \mathbb{R}^c$, the training process of the neural network optimizes parameter $\boldsymbol{\theta}$ to minimize the empirical training loss:

$$\mathcal{L}(\boldsymbol{\theta}) = \frac{1}{N} \sum_{i=1}^N \ell(f_{\boldsymbol{\theta}}(\boldsymbol{x}_i), \boldsymbol{y}_i) = \mathop{\mathbb{E}}_{(\boldsymbol{x}, \boldsymbol{y}) \in S} \left[ \ell(\boldsymbol{z}, \boldsymbol{y}) \right]. \tag{3}$$

**Hessians:** Fixing the parameter $\boldsymbol{\theta}$, we use $\boldsymbol{H}_\ell(\boldsymbol{v}, \boldsymbol{x})$ to denote the Hessian of some vector $\boldsymbol{v}$ with respect to scalar loss function $\ell$ at input $\boldsymbol{x}$.

$$\boldsymbol{H}_\ell(\boldsymbol{v}, \boldsymbol{x}) = \nabla_{\boldsymbol{v}}^2 \ell(f_{\boldsymbol{\theta}}(\boldsymbol{x}), \boldsymbol{y}) = \nabla_{\boldsymbol{v}}^2 \ell(\boldsymbol{z}, \boldsymbol{y}). \tag{4}$$

Note $\boldsymbol{v}$ here can be any vector. For example, the parameter Hessian is $\boldsymbol{H}_\ell(\boldsymbol{\theta}, \boldsymbol{x})$, where $\boldsymbol{v} = \boldsymbol{\theta}$. The layer-wise weight Hessian of the $p$-th layer is $\boldsymbol{H}_\ell(\boldsymbol{w}^{(p)}, \boldsymbol{x})$.

For simplicity, define $\mathbb{E}$ as the empirical expectation over the training sample $S$ unless explicitly stated otherwise. We focus on the layer-wise weight Hessians $\boldsymbol{H}_{\mathcal{L}}(\boldsymbol{w}^{(p)}) = \mathbb{E}[\boldsymbol{H}_\ell(\boldsymbol{w}^{(p)}, \boldsymbol{x})]$ with respect to loss, which are diagonal blocks in the full Hessian $\boldsymbol{H}_{\mathcal{L}}(\boldsymbol{\theta}) = \mathbb{E}[\boldsymbol{H}_\ell(\boldsymbol{\theta}, \boldsymbol{x})]$ corresponding to the cross terms between the weight coefficients of the same layer. We define $\boldsymbol{M}_{\boldsymbol{x}}^{(p)} := \boldsymbol{H}_\ell(\boldsymbol{z}^{(p)}, \boldsymbol{x})$ as the Hessian of output $\boldsymbol{z}^{(p)}$ with respect to empirical loss. With the notations defined above, we have the $p$-th layer-wise hessian for a single input as

$$\boldsymbol{H}_\ell(\boldsymbol{w}^{(p)}, \boldsymbol{x}) = \nabla_{\boldsymbol{w}^{(p)}}^2 \ell(\boldsymbol{z}, \boldsymbol{y}) = \boldsymbol{M}_{\boldsymbol{x}}^{(p)} \otimes (\boldsymbol{x}^{(p)}\boldsymbol{x}^{(p)T}). \tag{5}$$

It follows that

$$\boldsymbol{H}_{\mathcal{L}}(\boldsymbol{w}^{(p)}) = \mathbb{E}\left[ \boldsymbol{M}_{\boldsymbol{x}}^{(p)} \otimes \boldsymbol{x}^{(p)}\boldsymbol{x}^{(p)T} \right] = \mathbb{E}\left[ \boldsymbol{M} \otimes \boldsymbol{x}\boldsymbol{x}^T \right]. \tag{6}$$

The subscription $\boldsymbol{x}$ and the superscription $(p)$ will be omitted when there is no confusion, as our analysis primarily focuses on the same layer unless otherwise stated.

## 4 Kronecker Factorization of Hessian

The fact that layer-wise Hessian for a single sample can be decomposed into Kronecker product of two components naturally leads to the following conjecture:

**Conjecture** (Decoupling Conjecture). The layer-wise Hessian can be approximated by a Kronecker product of the expectation of its two components, that is

$$\boldsymbol{H}_{\mathcal{L}}(\boldsymbol{w}^{(p)}) = \mathbb{E}\left[ \boldsymbol{M} \otimes \boldsymbol{x}\boldsymbol{x}^T \right] \approx \mathbb{E}[\boldsymbol{M}] \otimes \mathbb{E}[\boldsymbol{x}\boldsymbol{x}^T]. \tag{7}$$

In particular, the top eigenvalues and eigenspace of $\boldsymbol{H}_{\mathcal{L}}(\boldsymbol{w}^{(p)})$ is close to those of $\mathbb{E}[\boldsymbol{M}] \otimes \mathbb{E}[\boldsymbol{x}\boldsymbol{x}^T]$.

Note that this conjecture is certainly true when $\boldsymbol{M}$ and $\boldsymbol{x}\boldsymbol{x}^T$ are approximately statistically independent. In Section 4.1 and Section 4.2 we will show that this conjecture is true in practice. We then analyze the two components separately in Sections 4.3 and F.3.3.

Assuming the decoupling conjecture, we can analyze the layer-wise Hessian by analyzing the two components separately. Note that $\mathbb{E}[\boldsymbol{M}]$ is the Hessian of the layer-wise output with respect to empirical loss, and $\mathbb{E}[\boldsymbol{x}\boldsymbol{x}^T]$ is the auto-correlation matrix of the layer-wise inputs. For simplicity we call $\mathbb{E}[\boldsymbol{M}]$ the output Hessian and $\mathbb{E}[\boldsymbol{x}\boldsymbol{x}^T]$ the input auto-correlation. For convolutional layers, we define a similar factorization $\mathbb{E}[\boldsymbol{M}] \otimes \mathbb{E}[\boldsymbol{x}\boldsymbol{x}^T]$ for the layer-wise Hessian, but with a different $\boldsymbol{M}$ motivated by Grosse & Martens (2016). (See Appendix A.2)

In this paper we also note that the off-diagonal blocks of the full Hessian can also be decomposed similarly. We can then approximate each block using the Kronecker factorization, and when the input auto-correlation matrices are close to rank 1, this allows us to approximate the eigenvalues and eigenvectors of the full parameter Hessian. The details of this approximation is stated in Appendix B.

**Experiment Setup:** We conduct experiments on the CIFAR-10, CIFAR-100 (MIT) (Krizhevsky, 2009), and MNIST (CC BY-SA 3.0) (LeCun et al., 1998) datasets as well as their random labeled versions, namely MNIST-R and CIFAR10-R. We use PyTorch (Paszke et al., 2019) framework for all experiments. We used several different fully connected (fc) networks (a fc network with $m$ hidden layers and $n$ neurons each hidden layer is denoted as F-$n^m$), several variations of LeNet (LeCun et al., 1998), VGG11 (Simonyan & Zisserman, 2015), and ResNet18 (He et al., 2016). More representative results are included in Appendix E. The eigenvalues and eigenvectors of the exact layer-wise Hessians are approximated using a modified Lanczos algorithm (Golmant et al., 2018) which is described in detail in Appendix C. We use "layer:network" to denote a layer of a particular network. For example, conv2:LeNet5 refers to the second convolutional layer in LeNet5.

## 4.1 Approximation of Layer-wise Hessian and Full Hessian

We compare the top eigenvalues and eigenspaces of the approximated Hessian and the true Hessian. We use the standard definition of subspace overlap to measure the similarity between top eigenspaces.

**Definition 4.1** (Subspace Overlap). For $k$-dimensional subspaces $\boldsymbol{U}, \boldsymbol{V}$ in $\mathbb{R}^d$ ($d \geq k$) where the basis vectors $\boldsymbol{u}_i$'s and $\boldsymbol{v}_i$'s are column vectors, with $\boldsymbol{\phi}$ as the size $k$ vector of canonical angles between $\boldsymbol{U}$ and $\boldsymbol{V}$, we define the subspace overlap of $\boldsymbol{U}$ and $\boldsymbol{V}$ as

$$\text{Overlap}(\boldsymbol{U}, \boldsymbol{V}) := \|\boldsymbol{U}^T\boldsymbol{V}\|_F^2/k = \|\cos\boldsymbol{\phi}\|_2^2/k. \tag{8}$$

Note that when $k = 1$, the overlap is equivalent to the squared dot product between the two vectors.

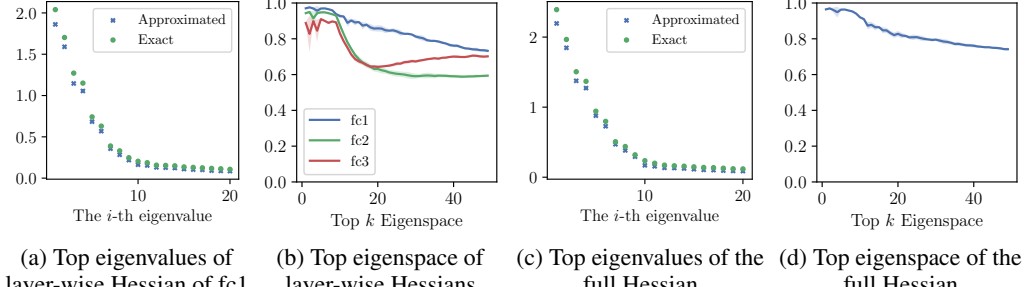

(a) Top eigenvalues of layer-wise Hessian of fc1  (b) Top eigenspace of layer-wise Hessians  (c) Top eigenvalues of the full Hessian  (d) Top eigenspace of the full Hessian

Figure 2: Comparison between the approximated and true layer-wise Hessian of F-$200^2$.

As shown in Fig. 2, this approximation works reasonably well for the top eigenvalues and eigenspaces of both layer-wise weight Hessians and the full parameter Hessian.

## 4.2 Eigenvector Correspondence for Layer-wise Hessians

Suppose the $i$-th eigenvector for $\mathbb{E}[\boldsymbol{x}\boldsymbol{x}^T]$ is $\boldsymbol{v}_i$ and the $j$-th eigenvector for $\mathbb{E}[\boldsymbol{M}]$ is $\boldsymbol{u}_j$. Then the Kronecker product $\mathbb{E}[\boldsymbol{M}] \otimes \mathbb{E}[\boldsymbol{x}\boldsymbol{x}^T]$ has an eigenvector $\boldsymbol{u}_j \otimes \boldsymbol{v}_i$. Therefore if the decoupling conjecture holds, one would expect that the top eigenvector of the layer-wise Hessian has a clear correspondence with the top eigenvectors of its two components. Since $\boldsymbol{u} \otimes \boldsymbol{v}$ is just the flattened matrix $\boldsymbol{u}\boldsymbol{v}^T$, we may naturally define the following reshape operation.

**Definition 4.2** (Layer-wise Eigenvector Matricization). Consider a layer with input dimension $n$ and output dimension $m$. For an eigenvector $\boldsymbol{h} \in \mathbb{R}^{mn}$ of its layer-wise Hessian, the matricized form of $\boldsymbol{h}$ is $\text{Mat}(\boldsymbol{h}) \in \mathbb{R}^{m \times n}$ where $\text{Mat}(\boldsymbol{h})_{i,j} = \boldsymbol{h}_{(i-1)m+j}$.

More concretely, to demonstrate the correspondence between the eigenvectors of the layer-wise Hessian and the eigenvectors of matrix $\mathbb{E}[M]$ and $\mathbb{E}[xx^T]$, we introduce "eigenvector correspondence matrices" as shown in Fig. 3.

**Definition 4.3** (Eigenvector Correspondence Matrices). For layer-wise Hessian matrix $H \in \mathbb{R}^{mn \times mn}$ with eigenvectors $h_1, \cdots, h_{mn}$, and its corresponding auto-correlation matrix $\mathbb{E}[xx^T] \in \mathbb{R}^{n \times n}$ with eigenvectors $v_1, \cdots, v_n$. The correspondence between $v_i$ and $h_j$ can be defined as

$$\operatorname{Corr}(v_i, h_j) := \| \operatorname{Mat}(h_j)v_i \|^2. \tag{9}$$

For the output Hessian matrix $\mathbb{E}[M] \in \mathbb{R}^{m \times m}$ with eigenvectors $u_1, \cdots, u_m$, we can likewise define correspondence between $v_i$ and $h_j$ as

$$\operatorname{Corr}(u_i, h_j) := \| \operatorname{Mat}(h_j)^T u_i \|^2 \tag{10}$$

We may then define the eigenvector correspondence matrix between $H$ and $\mathbb{E}[xx^T]$ as a $n \times mn$ matrix whose $i, j$-th entry is $\operatorname{Corr}(v_i, h_j)$, and the eigenvector correspondence matrix between $H$ and $\mathbb{E}[M]$ as a $m \times mn$ matrix whose $i, j$-th entry is $\operatorname{Corr}(u_i, h_j)$.

Intuitively, if the $i, j$-th entry of the correspondence matrix is close to 1, then the eigenvector $h_j$ is likely to be the Kronecker product of $v_i$ (or $u_i$) with some vector. If the decoupling conjecture holds, every eigenvector of the layer-wise Hessian (column of the correspondence matrices) should have a perfect correlation of 1 with exactly one of $v_i$ and one of $u_i$.

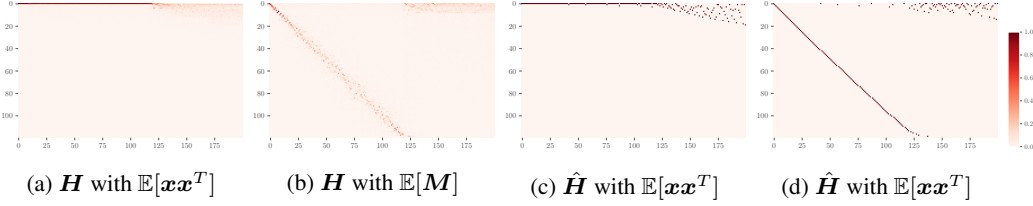

(a) $H$ with $\mathbb{E}[xx^T]$     (b) $H$ with $\mathbb{E}[M]$     (c) $\hat{H}$ with $\mathbb{E}[xx^T]$     (d) $\hat{H}$ with $\mathbb{E}[xx^T]$

Figure 3: Heatmap of Eigenvector Correspondence Matrices for fc1:LeNet5, which has 120 output neurons. Here we take the top left corner of the eigenvector correspondence matrices. Similarities between (a)(c) and (b)(d) respectively verify the decoupling conjecture.

In Fig. 3 we can see that the correspondence matrices for the true layer-wise Hessian $H$ approximately satisfies this property for top eigenvectors. The similarity between the correspondence patterns for the true and Kroneckor product approximated Hessian $\hat{H}$ also verifies the validity of the Kronecker approximation for dominate eigenspace. From Fig. 3a and Fig. 3c, the top $m$ eigenvectors of the true layer-wise Hessian and the approximated Hessian are all highly correlated with $v_1$, the first eigenvector of $\mathbb{E}[xx^T]$. From Fig. 3b and Fig. 3d, the correspondence with the $\mathbb{E}[M]$ component has a near diagonal pattern for both the true Hessian and the Kronecker approximation. Thus for small $i$ we have $h_i \approx v_1 \otimes u_i$.

## 4.3 Structure of Input Auto-correlation Matrix $\mathbb{E}[xx^T]$

To understand the structure of the auto-correlation matrix, a key observation is that the input $x$ for most layers are outputs of a ReLU, hence it is nonnegative. We can decompose the auto-correlation matrix as $\mathbb{E}[xx^T] = \mathbb{E}[x]\mathbb{E}[x]^T + \mathbb{E}[(x - \mathbb{E}[x])(x - \mathbb{E}[x])^T]$. We denote $\Sigma_x := \mathbb{E}[(x - \mathbb{E}[x])(x - \mathbb{E}[x])^T]$ the auto-covariance matrix. As every sample $x$ is nonnegative, the expectation $\mathbb{E}[x]\mathbb{E}[x]^T$ has a large norm and usually dominates the covariance matrix $\Sigma_x$.

We empirically verified this phenomenon on various networks and datasets. The first eigenvector of $\mathbb{E}[xx^T]$ has a very high overlap with $\mathbb{E}[x]$ (squared dot product mean: 0.997, range: 0.964-1.000). Meanwhile $\|\mathbb{E}[x]\mathbb{E}[x]^T\|$ is significantly larger than $\|\Sigma_x\|$ in our experiments ($\|\mathbb{E}[x]\mathbb{E}[x]^T\|/\|\Sigma_x\|$ mean: 12.08, range: 2.28-30.03). This suggests that $\mathbb{E}[x]\mathbb{E}[x]^T$ is approximately equal to $\mathbb{E}[xx^T]$ and dominates the covariance $\Sigma_x$. Similar phenomenon also exists for convolution layers. The complete experiment results are provided in Appendix E.1. We also observe the $\mathbb{E}[xx^T]$ matrices are all close to rank 1 throughout the training trajectory as shown in Appendix E.4.

## 4.4 Low Rank Structure of $\mathbb{E}[M]$

Previous works observed the gap in Hessian eigenspectrum around the number of classes $c$ (where $c = 10$ in our experiments on CIFAR10 and MNIST). Since $\mathbb{E}[\boldsymbol{x}\boldsymbol{x}^T]$ is close to rank 1 and the Kronecker factorization is a good approximation for top eigenspace, the top eigenvalues of layer-wise Hessian can be approximated as the top eigenvalues of $\mathbb{E}[M]$ multiplied by the first eigenvalue of $\mathbb{E}[\boldsymbol{x}\boldsymbol{x}^T]$. Thus, the top eigenvalues of Hessians should have the same relative ratios as the top eigenvalues of their corresponding $\mathbb{E}[M]$'s. Therefore, the outliers should also appear in $\mathbb{E}[M]$.

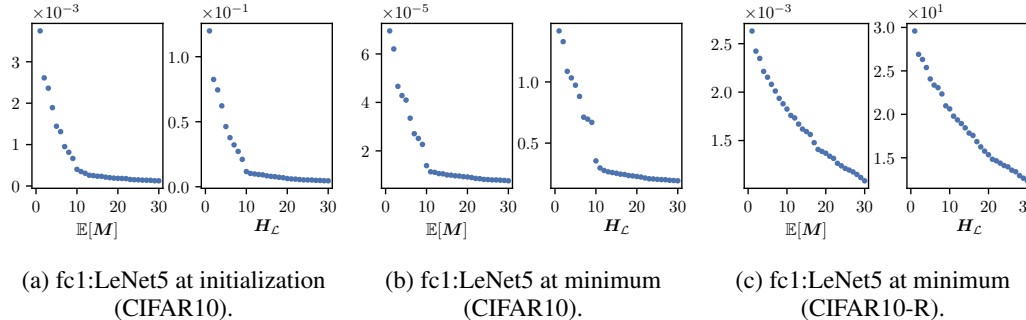

(a) fc1:LeNet5 at initialization (CIFAR10).

(b) fc1:LeNet5 at minimum (CIFAR10).

(c) fc1:LeNet5 at minimum (CIFAR10-R).

Figure 4: Eigenspectrum of the layer-wise output Hessian $\mathbb{E}[M]$ and the layer-wise weight Hessian $\boldsymbol{H}_{\mathcal{L}}(\boldsymbol{w}^{(p)})$. The vertical axes denote the eigenvalues. Similarity between the two eigenspectra is a direct consequence of a low rank $\mathbb{E}[\boldsymbol{x}\boldsymbol{x}^T]$ and the decoupling conjecture.

Fig. 4 shows the similarity of eigenvalue spectrum between $\mathbb{E}[M]$ and layer-wise Hessians in different situations, which agrees with our prediction. However, the eigengap only appear at initialization and at minimum for true labels (Fig. 4a and Fig. 4b), but not at minimum for random labels (Fig. 4c). To understand the structure of $\mathbb{E}[M]$ itself, we consider a simplified setting where we have a random two-layer neural network with random data.

**Theorem 4.1** (informal). *For a two-layer neural network with Gaussian input, at initialization, when the network is large, the output Hessian of the first layer is approximately rank $(c - 1)$ and the corresponding top eigenspace is $\mathcal{R}(\boldsymbol{W}^{(2)})\backslash\{\boldsymbol{W}^{(2)} \cdot \boldsymbol{1}\}$ and $\mathcal{R}(\boldsymbol{W}^{(2)})$ denotes the row space of the weight matrix $\boldsymbol{W}^{(2)}$ of the second layer.*

The formal statement of this theorem and the full proof is in Appendix H. The closed form calculation can be heuristically extended to the case with multiple layers, that the top eigenspace of the output Hessian of the $k$-layer would be approximately $\mathcal{R}(\boldsymbol{S}^{(k)}) \setminus \{\boldsymbol{S}^{(k)} \cdot \boldsymbol{1}\}$, where $\boldsymbol{S}^{(k)} = \boldsymbol{W}^{(n)}\boldsymbol{W}^{(n-1)} \cdots \boldsymbol{W}^{(k+1)}$ and $\mathcal{R}(\boldsymbol{S}^{(k)})$ is the row space of $\boldsymbol{S}^{(k)}$.

Though our result was only proven for random initialization and random data, we observe that this subspace also has high overlap with the top eigenspace of output Hessian at the minima of models trained with real datasets. The corresponding empirical results are shown in Appendix H.4.

# 5 Understanding Structures of Layer-wise Hessian

## 5.1 Eigenspace Overlap of Different Models

Several interesting structures of layer-wise Hessians can be explained using the decoupling conjecture and eigenvalue correspondence. Consider models with the same network structure that are trained on the same dataset using different random initializations, despite no obvious similarity between their parameters, we observe surpisingly high overlap between the dominating eigenspace of their layer-wise Hessians.

Fig. 5 includes 4 different variants of LeNet5 trained on CIFAR10, 3 different variants of ResNet18 trained on CIFAR100, and 3 different variants of VGG11 trained on CIFAR100. For each structural variant, 5 models are trained from independent random initializations. We plot the average pairwise overlap between the top eigenspaces of those models' layer-wise Hessians. In each figure, we vary the number of output neuron/channels. For the same structure, the top eigenspaces of different models exhibits a highly non-trivial overlap which peaks near $m$ – the dimension of the layer's output.

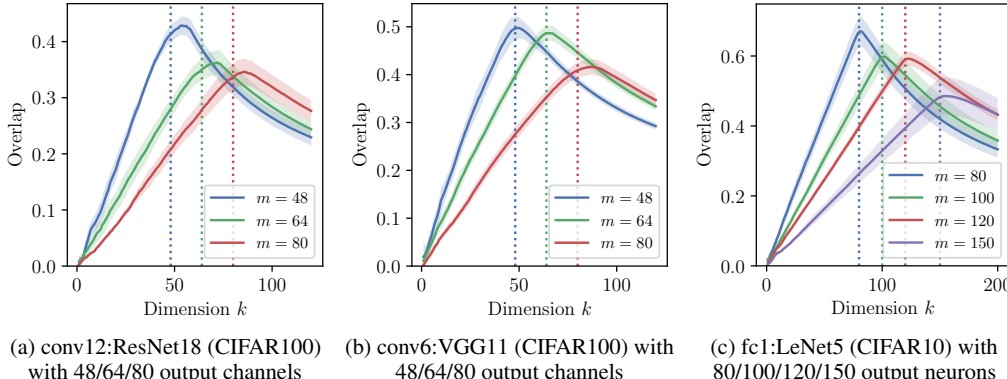

(a) conv12:ResNet18 (CIFAR100) with 48/64/80 output channels

(b) conv6:VGG11 (CIFAR100) with 48/64/80 output channels

(c) fc1:LeNet5 (CIFAR10) with 80/100/120/150 output neurons

Figure 5: Overlap between the top $k$ dominating eigenspace of different independently trained models. The overlap peaks at the output dimension $m$. The eigenspace overlap is defined in Definition 4.1.

As we observed in Section 4.3, the auto-correlation matrix $\mathbb{E}[\boldsymbol{x}\boldsymbol{x}^T]$ is approximately $\mathbb{E}[\boldsymbol{x}]\mathbb{E}[\boldsymbol{x}]^T$. Thus if the $i$-th eigenvector of $\mathbb{E}[\boldsymbol{M}]$ is $\boldsymbol{u}_i$, the $i$-th eigenvector of the layer-wise Hessian would be close to $\boldsymbol{u}_i \otimes \hat{\mathbb{E}}[\boldsymbol{x}]$, where $\hat{\mathbb{E}}[\boldsymbol{x}] = \mathbb{E}[\boldsymbol{x}]/\|\mathbb{E}[\boldsymbol{x}]\|$, is normalized $\mathbb{E}[\boldsymbol{x}]$. Even though the directions of $\boldsymbol{u}_i$'s can be very different for different models, at rank $m$ these vectors always span the entire space, as a result the top-$m$ eigenspace for layer-wise Hessian is close to $\boldsymbol{I}_m \otimes \hat{\mathbb{E}}[\boldsymbol{x}]$.

Now suppose we have two different models with $\hat{\mathbb{E}}[\boldsymbol{x}]_1$ and $\hat{\mathbb{E}}[\boldsymbol{x}]_2$ respectively. Their top-$m$ eigenspaces are close to $\boldsymbol{I}_m \otimes \hat{\mathbb{E}}[\boldsymbol{x}]_1$ and $\boldsymbol{I}_m \otimes \hat{\mathbb{E}}[\boldsymbol{x}]_2$ respectively. In this case, it is easy to check that the overlap at $m$ is approximately $(\hat{\mathbb{E}}[\boldsymbol{x}]_1^T \hat{\mathbb{E}}[\boldsymbol{x}]_2)^2$. Since $\hat{\mathbb{E}}[\boldsymbol{x}]_1$ and $\hat{\mathbb{E}}[\boldsymbol{x}]_2$ are the same for the input layer and all non-negative for other layers, the inner-product between them is large and the overlap is expected to be high at dimension $m$.

Our explanations of the overlap relies on two properties: the auto-correlation matrix needs to be very close to rank 1 and the eigenspectrum of output Hessian should have a heavy-tail. While both are true in many shallow networks and in later layers of deeper networks, they are not satisfied for earlier layers of deeper networks. In Appendix F.3 we explain how one can still understand the overlap using correspondence matrices when the above simplified argument does not hold.

## 5.2 Dominating Eigenvectors of Layer-wise Hessian are Low Rank

A natural corollary for the Kronecker factorization approximation of layer-wise Hessians is that the eigenvectors of the layer-wise Hessians are low rank. Let $\boldsymbol{h}_i$ be the $i$-th eigenvector of a layer-wise Hessian. The rank of $\mathrm{Mat}(\boldsymbol{h}_i)$ can be considered as an indicator of the complexity of the eigenvector. Consider the case that $\boldsymbol{h}_i$ is one of the top eigenvectors. From Section 5.1, we have $\boldsymbol{h}_i \approx \boldsymbol{u}_i \otimes \hat{\mathbb{E}}[\boldsymbol{x}]$. Thus, $\mathrm{Mat}(\boldsymbol{h}_i) \approx \boldsymbol{u}_i \hat{\mathbb{E}}[\boldsymbol{x}]^T$, which is approximately rank 1.

Experiments shows that first singular values of $\mathrm{Mat}(\boldsymbol{h}_i)$ divided by its Frobenius Norm are usually much larger than 0.5, indicating the top eigenvectors of the layer-wise Hessians are very close to rank 1. Fig. 22 shows first singular values of $\mathrm{Mat}(\boldsymbol{h}_i)$ divided by its Frobenius Norm for $i$ from 1 to 200. We can see that the top eigenvectors of the layer-wise Hessians are very close to rank 1.

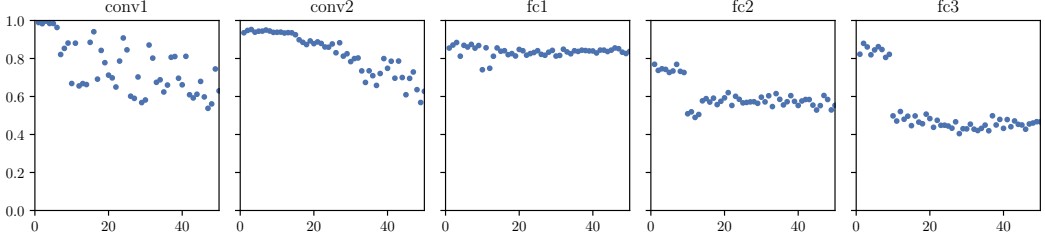

Figure 6: Ratio between top singular value and Frobenius norm of matricized dominating eigenvectors. (LeNet5 on CIFAR10). The horizontal axes denote the index $i$ of eigenvector $\boldsymbol{h}_i$, and the vertical axes denote $\|\mathrm{Mat}(\boldsymbol{h}_i)\|/\|\mathrm{Mat}(\boldsymbol{h}_i)\|_F$.

### 5.3 Batch Normalization and Zero-mean Input

According to our explanation, the good approximation and high overlap of top eigenspace both depend on the low rank structure of $\mathbb{E}[\boldsymbol{x}\boldsymbol{x}^T]$. Also, the low rank structure is caused by the fact that $\mathbb{E}[\boldsymbol{x}]\mathbb{E}[\boldsymbol{x}]^T$ dominates $\boldsymbol{\Sigma}_{\boldsymbol{x}}$ in most cases. Therefore, it's natural to conjecture that models trained using Batch Normalization (BN) (Ioffe & Szegedy, 2015) will change these phenomena as $\mathbb{E}[\boldsymbol{x}]$ will be zero and $\mathbb{E}[\boldsymbol{x}\boldsymbol{x}^T] = \boldsymbol{\Sigma}_{\boldsymbol{x}}$ for those models. Indeed, as shown in Ghorbani et al. (2019), BN suppresses the outliers in the Hessian eigenspectrum and Papyan (2020) provided an explanation.

We experiment on our networks with BN. The results are shown in Appendix F.4. We found that $\mathbb{E}[\boldsymbol{x}\boldsymbol{x}^T]$ is no longer close to rank 1 for models trained with BN. However, $\mathbb{E}[\boldsymbol{x}\boldsymbol{x}^T]$ still have a few large eigenvalues. In this case, all the previous structures ($c$ outliers, high eigenspace overlap, low rank eigenvectors) become weaker. The decoupling conjecture itself also becomes less accurate. However, the approximation still gives meaningful information.

## 6 Tighter PAC-Bayes Bound with Hessian Information

PAC-Bayes bound is commonly used to derive upper bounds on the generalization error.

**Theorem 6.1** (PAC-Bayes Bound). (McAllester, 1999; Langford & Seeger, 2001) With the hypothesis space $\mathcal{H}$ parametrized by model parameters. For any prior distribution $P$ in $\mathcal{H}$ that is chosen independently from the training set $S$, and any posterior distribution $Q$ in $\mathcal{H}$ whose choice may inference $S$, with probability $1 - \delta$,

$$D_{\mathrm{KL}}\left(\hat{e}(Q)\|e(Q)\right) \leq \frac{D_{\mathrm{KL}}(Q\|P) + \log\frac{|S|}{\delta}}{|S| - 1}, \tag{11}$$

where $e(Q)$ is the expected classification error for the posterior over the underlying data distribution and $\hat{e}(Q)$ is the classification error for the posterior over the training set.

Intuitively, if one can find a posterior distribution $Q$ that has low loss on the training set, and is close to the prior $P$, then the generalization error on $Q$ must be small. Dziugaite & Roy (2017) uses optimization techniques to find an optimal posterior in the family of Gaussians with diagonal covariance. They showed that the bound is nonvacuous for several neural network models.

We follow Dziugaite & Roy (2017) to set the prior $P$ to be a multi-variant Gaussian. The covariance is a multiple of identity. Thus, it is invariant with respect to the change of basis. For the posterior, when the variance in one direction is larger, the distance with the prior decreases; however this also has the risk of increasing the empirical loss over the posterior. In general, one would expect the variance to be larger along a flatter direction in the loss landscape. However, since the covariance matrix of $Q$ is fixed to be diagonal in Dziugaite & Roy (2017), the search of optimal deviation happens in standard basis vectors which are not aligned with the local loss landscape.

Using the Kronecker factorization as in Eq. (7), we can approximate the layer-wise Hessian's eigenspace. We set $Q$ to be a Gaussian whose covariance is diagonal in the eigenbasis of the layer-wise Hessians. We expect the alignment of sharp and flat directions will result in a better optimized posterior $Q$ and thus a tighter bound on classification error.

We perform the same optimization process as proposed by Dziugaite & Roy (2017). Our algorithm is called *Approx Hessian* (APPR) when we fix the layer-wise Hessian eigenbasis to the one at the minimum and *Iterative Hessian* (ITER) when we update the eigenbasis dynamically with the mean of the Gaussian which is being optimized. To accelerate *Iterative Hessian*, we used generalization of Theorem 4.1 to directly approximate the output hessian, which is then used to compute the eigenbasis. We call this variant algorithm of *Iterative Hessian with approximated output Hessian* (ITER.M). The results of this variant are only slightly worse than *Iterative Hessian*, which also suggests the approximation of the output hessian is reasonable.

We used identical dataset, network structures and experiment settings as in Dziugaite & Roy (2017), with a few adjustments in hyperparameters. We also added T-$200^2$ used in Section 4. T-$600_{10}$ and T-$200_{10}^2$ are trained on standard MNIST while all others are trained on MNIST-2 (see Appendix D.1). The results are shown in Table 1 with a confidence of 0.965.

Table 1: Optimized PAC-Bayes bounds using different methods. T-$n^m$ and R-$n^m$ represents network F-$n^m$ trained with true/random labels. TESTER. gives the empirical generalization gap. BASE represents the bound given by the algorithm proposed by Dziugaite & Roy (2017). APPR, ITER, and ITER.M represents the bound given by our algorithms.

| Model | T-600 | T-1200 | T-$300^2$ | T-$600^2$ | R-600 | T-$600_{10}$ | T-$200_{10}^2$ |
|---|---|---|---|---|---|---|---|
| TestEr. | 0.015 | 0.016 | 0.015 | 0.015 | 0.493 | 0.018 | 0.021 |
| BASE | 0.154 | 0.175 | 0.169 | 0.192 | 0.605 | 0.287 | 0.417 |
| APPR (ours) | 0.146 | 0.173 | 0.142 | 0.171 | 0.565 | 0.242 | 0.273 |
| ITER (ours) | **0.120** | **0.142** | **0.125** | **0.146** | 0.568 | **0.213** | **0.215** |
| ITER.M (ours) | 0.126 | 0.149 | 0.131 | 0.150 | **0.562** | 0.223 | 0.273 |

We also plotted the final posterior variance, $s$ for network T-$200_{10}^2$ in Fig. 7. For our algorithms, APPR, ITER, and ITER.M, we can see that direction associated with larger eigenvalue has a smaller variance. This agrees with our presumption that top eigenvectors are aligned with sharper directions and should have smaller variance after optimization. Detailed algorithm description and experiment results are shown in Appendix G.

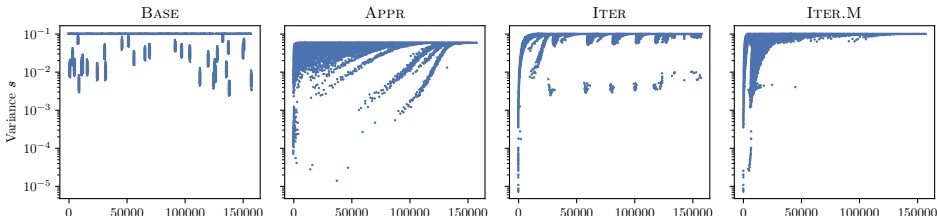

Figure 7: Optimized posterior variance $s$ using different algorithms (fc1:T-$200^2$ trained on MNIST). The horizontal axis denotes the eigenbasis ordered with decreasing eigenvalues. The abbreviation of algorithms are the same as in Table 1.

# 7   Conclusions

In this paper we identified two new surprising structures in the dominating eigenspace of layerwise Hessians for neural networks. Specifically, the eigenspace overlap reveals a novel similarity between different models. We showed that under a decoupling conjecture, these structures can be explained by a Kronecker factorization. We analyze each component in the factorization, in particular we prove that the output Hessian is approximately rank $c - 1$ for random two-layer neural networks. Our proof gives a simple heuristic formula to estimate the top-eigenspace of the Hessian. As a proof of concept, we showed that these structures can be used to find better explicit generalization bounds. Since the dominating eigenspace of Hessian, which is the sharpest directions on the loss landscape, plays an important role in both generalization (Keskar et al., 2017; Jiang et al., 2020) and optimization (Gur-Ari et al., 2018), we believe our new understanding can benefit both fields. We hope this work would be a starting point towards formally proving the structures of neural network Hessians.

**Limitations and Open Problems**   Most of our work focuses on the layerwise Hessian except for Section 4.1 and Appendix B. The eigenspace overlap phenomenon depends on properties of auto-correlation and output Hessian, which are weaker for earlier layers of larger networks. Our theoretical results need to assume the parameters are random, and only applies to fully-connected networks. The immediate open problems include why the decoupling conjecture is correct and why the output Hessian is low rank in more general networks.

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

## Appendix Roadmap

1. In Appendix A, we provide the detailed derivations of Hessian for fully-connected and convolutional layers.

2. In Appendix B, we provide detailed description on the approximation of dominating eigenvectors of the full hessian.

3. In Appendix C, we explain how we compute the eigenvalues and eigenvectors of full and layer-wise Hessian numerically.

4. In Appendix D, we give the detailed experiment setups, including the datasets, network structures, and the training settings we use.

5. In Appendix E, we provide detailed experimental results that are not fully included in the main text.

6. In Appendix F, we provide additional and more general explanations of the phenomena we found.

7. In Appendix G, we give a detailed description of the PAC-Bayes bound that we optimize and the algorithm we use to optimize the bound.

8. In Appendix H, we provided the complete statement and proof for Theorem 4.1.

## A    Detailed Derivations

### A.1    Derivation of Hessian

For an input $\boldsymbol{x}$ with label $\boldsymbol{y}$, we define the Hessian of single input loss with respect to vector $\boldsymbol{v}$ as

$$\boldsymbol{H}_\ell(\boldsymbol{v}, \boldsymbol{x}) = \nabla_{\boldsymbol{v}}^2 \ell(f_{\boldsymbol{\theta}}(\boldsymbol{x}), \boldsymbol{y}) = \nabla_{\boldsymbol{v}}^2 \ell(\boldsymbol{z}_{\boldsymbol{x}}, \boldsymbol{y}). \tag{12}$$

We define the Hessian of loss with respect to $\boldsymbol{v}$ for the entire training sample as

$$\boldsymbol{H}_{\mathcal{L}}(\boldsymbol{v}) = \nabla_{\boldsymbol{v}}^2 \mathcal{L}(\boldsymbol{\theta}) = \sum_{i=1}^{N} \nabla_{\boldsymbol{v}}^2 \ell(f_{\boldsymbol{\theta}}(\boldsymbol{x}_i), \boldsymbol{y}_i) = \sum_{i=1}^{N} \boldsymbol{H}_\ell(\boldsymbol{v}, \boldsymbol{x}_i) = \mathbb{E}\left[\boldsymbol{H}_\ell(\boldsymbol{v}, \boldsymbol{x})\right]. \tag{13}$$

We now derive the Hessian for a fixed input label pair $(\boldsymbol{x}, \boldsymbol{y})$. Following the definition and notations in Section 3, we also denote output as $\boldsymbol{z} = f_{\boldsymbol{\theta}}(\boldsymbol{x})$. We fix a layer $p$ for the layer-wise Hessian. Here the layer-wise weight Hessian is $\boldsymbol{H}_\ell(\boldsymbol{w}^{(p)}, \boldsymbol{x})$. We also have the output for the layer as $\boldsymbol{z}^{(p)}$. Since $\boldsymbol{w}^{(p)}$ only appear in the layer but not the subsequent layers, we can consider $\boldsymbol{z} = f_{\boldsymbol{\theta}}(\boldsymbol{x}) = g_{\boldsymbol{\theta}}(\boldsymbol{z}^{(p)}(\boldsymbol{w}, \boldsymbol{x}))$ where $g_{\boldsymbol{\theta}}$ only contains the layers after the $p$-th layer and does not depend on $\boldsymbol{w}^{(p)}$. Thus, using the Hessian Chain rule (Skorski, 2019), we have

$$\boldsymbol{H}_\ell(\boldsymbol{w}^{(p)}, \boldsymbol{x}) = \left(\frac{\partial \boldsymbol{z}^{(p)}}{\partial \boldsymbol{w}^{(p)}}\right)^T \boldsymbol{H}_\ell(\boldsymbol{z}^{(p)}, \boldsymbol{x}) \left(\frac{\partial \boldsymbol{z}^{(p)}}{\partial \boldsymbol{w}^{(p)}}\right) + \sum_{i=1}^{m^{(p)}} \frac{\partial \ell(\boldsymbol{z}, \boldsymbol{y})}{\partial z_i^{(p)}} \nabla_{\boldsymbol{w}^{(p)}}^2 z_i^{(p)}, \tag{14}$$

where $z_i^{(p)}$ is the $i$th entry of $\boldsymbol{z}^{(p)}$ and $m^{(p)}$ is the number of neurons in $p$-th layer (size of $\boldsymbol{z}^{(p)}$).

Since $\boldsymbol{z}^{(p)} = \boldsymbol{W}^{(p)} \boldsymbol{x}^{(p)} + \boldsymbol{b}^{(p)}$ and $\boldsymbol{w}^{(p)} = \operatorname{vec}(\boldsymbol{W}^{(p)})$ we have

$$\frac{\partial \boldsymbol{z}^{(p)}}{\partial \boldsymbol{w}^{(p)}} = \boldsymbol{I}_{m^{(p)}} \otimes \boldsymbol{x}^{(p)T}. \tag{15}$$

Since $\frac{\partial \boldsymbol{z}^{(p)}}{\partial \boldsymbol{w}^{(p)}}$ does not depend on $\boldsymbol{w}^{(p)}$, for all $i$ we have $\nabla_{\boldsymbol{w}^{(p)}}^2 z_i^{(p)} = 0$. Thus,

$$\boldsymbol{H}_\ell(\boldsymbol{w}^{(p)}, \boldsymbol{x}) = \left(\boldsymbol{I}_{m^{(p)}} \otimes \boldsymbol{x}^{(p)}\right) \boldsymbol{H}_\ell(\boldsymbol{z}^{(p)}, \boldsymbol{x}) \left(\boldsymbol{I}_{m^{(p)}} \otimes \boldsymbol{x}^{(p)T}\right). \tag{16}$$

We define $\boldsymbol{M}_{\boldsymbol{x}}^{(p)} = \boldsymbol{H}_\ell(\boldsymbol{z}^{(p)}, \boldsymbol{x})$ as in Section 3 so that

$$\boldsymbol{H}_\ell(\boldsymbol{w}^{(p)}, \boldsymbol{x}) = \left(\boldsymbol{I}_{m^{(p)}} \otimes \boldsymbol{x}^{(p)}\right) \boldsymbol{M}_{\boldsymbol{x}}^{(p)} \left(\boldsymbol{I}_{m^{(p)}} \otimes \boldsymbol{x}^{(p)T}\right) = \boldsymbol{M}_{\boldsymbol{x}}^{(p)} \otimes \boldsymbol{x}^{(p)} \boldsymbol{x}^{(p)T}. \tag{17}$$

We now look into $M_x^{(p)} = H_\ell(z^{(p)}, x)$. Again we have $z = g_\theta(z^{(p)})$ and can use chain rule here,

$$H_\ell(z^{(p)}, x) = \left(\frac{\partial z}{\partial z^{(p)}}\right)^T H_\ell(z, x) \left(\frac{\partial z}{\partial z^{(p)}}\right) + \sum_{i=1}^{c} \frac{\partial \ell(z, y)}{\partial z_i} \nabla^2_{z^{(p)}} z_i \tag{18}$$

By letting $p := \operatorname{softmax}(z)$ be the output confidence vector, we define the Hessian with respect to output logit $z$ as $A_x$ and have

$$A_x := H_\ell(z, x) = \nabla^2_z l(z, y) = \operatorname{diag}(p) - pp^T, \tag{19}$$

according to Singla et al. (2019).

We also define the Jacobian of $z$ with respect to $z^{(p)}$ (informally logit gradient for layer $p$) as $G_x^{(p)} := \frac{\partial z}{\partial z^{(p)}}$. For FC layers with ReLUs, we can consider ReLU after the $p$-th layer as multiplying $z^{(p)}$ by an indicator function $\mathbf{1}_{z^{(p)}>0}$. To use matrix multiplication, we can turn the indicator function into a diagonal matrix and define it as $D^{(p)}$ where

$$D^{(p)} := \operatorname{diag}\left(\mathbf{1}_{z^{(p)}>0}\right). \tag{20}$$

Thus, we have the input of the next layer as $x^{(p+1)} = D^{(p)} z^{(p)}$. The FC layers can then be considered as a sequential matrix multiplication and we have the final output as

$$z = W^{(L)} D^{(L-1)} W^{(L-1)} D^{(L-2)} \cdots D^{(p)} z^{(p)}. \tag{21}$$

Thus,

$$G_x^{(p)} = \frac{\partial z}{\partial z^{(p)}} = W^{(L)} D^{(L-1)} W^{(L-1)} D^{(L-2)} \cdots D^{(p)}. \tag{22}$$

Since $G_x^{(p)}$ is independent of $z^{(p)}$, we have

$$\nabla^2_{z^{(p)}} z_i = 0, \forall i. \tag{23}$$

Thus,

$$M_x^{(p)} = H_\ell(z^{(p)}, x) = G_x^{(p)T} A_x G_x^{(p)}. \tag{24}$$

Moreover, loss Hessian with respect to the bias term $b^{(p)}$ equals to that with respect to the output of that layer $z^{(p)}$. We thus have

$$H_\ell(b^{(p)}, x) = M_x^{(p)} = G_x^{(p)T} A_x G_x^{(p)}. \tag{25}$$

The Hessians of loss for the entire training sample are simply the empirical expectations of the Hessian for single input. We have the formula as the following:

$$H_{\mathcal{L}}(w^{(p)}) = \mathbb{E}\left[H_\ell(w^{(p)}, x)\right] = \mathbb{E}\left[M_x^{(p)} \otimes x^{(p)} x^{(p)T}\right], \tag{26}$$

$$H_{\mathcal{L}}(b^{(p)}) = H_{\mathcal{L}}(z^{(p)}) = \mathbb{E}\left[M_x^{(p)}\right] = \mathbb{E}\left[G_x^{(p)T} A_x G_x^{(p)}\right]. \tag{27}$$

Note that we can further decompose $A_x = Q_x^T Q_x$, where

$$Q_x = \operatorname{diag}\left(\sqrt{p}\right)\left(I_c - \mathbf{1}_c p^T\right), \tag{28}$$

with $\mathbf{1}_c$ is a all one vector of size $c$, proved in Papyan (2019).

We can further extend the close form expression to off diagonal blocks and the bias entries to get the full Gauss-Newton term of Hessian. Let

$$F_x^T = \begin{pmatrix} G_x^{(1)T} \otimes x^{(1)} \\ G_x^{(1)T} \\ G_x^{(2)T} \otimes x^{(2)} \\ G_x^{(2)T} \\ \vdots \\ G_x^{(L)T} \otimes x^{(n)} \\ G_x^{(L)T} \end{pmatrix}. \tag{29}$$

The full Hessian is given by

$$H_{\mathcal{L}}(\theta) = \mathbb{E}\left[F_x^T A_x F_x\right] + \mathbb{E}\left[\sum_{i=1}^{c} \frac{\partial \ell(z, y)}{z_i} \nabla^2_\theta z_i\right]. \tag{30}$$

## A.2 Approximating Weight Hessian of Convolutional Layers

The approximation of weight Hessian of convolutional layer is a trivial extension from the approximation of Fisher information matrix of convolutional layer by Grosse & Martens (2016).

Consider a two dimensional convolutional layer of neural network with $m$ input channels and $n$ output channels. Let its input feature map **X** be of shape $(n, X_1, X_2)$ and output feature map **Z** be of shape $(m, P_1, P_2)$. Let its convolution kernel be of size $K_1 \times K_2$. Then the weight **W** is of shape $(m, n, K_1, K_2)$, and the bias $\boldsymbol{b}$ is of shape $(m)$. Let $P$ be the number of patches slide over by the convolution kernel, we have $P = P_1 P_2$.

Follow Dangel et al. (2020), we define $\boldsymbol{Z} \in \mathbb{R}^{m \times P}$ as the reshaped matrix of **Z** and $\boldsymbol{W} \in \mathbb{R}^{m \times n K_1 K_2}$ as the reshaped matrix of **W**. Define $\boldsymbol{B} \in \mathbb{R}^{m \times P}$ by broadcasting $\boldsymbol{b}$ to $P$ dimensions. Let $\boldsymbol{X} \in \mathbb{R}^{n K_1 K_2 \times P}$ be the unfolded **X** with respect to the convolutional layer. The unfold operation (Paszke et al., 2019) is commonly used in computation to model convolution as matrix operations.

After the above transformation, we have the linear expression of the $p$-th convolutional layer similar to FC layers:

$$\boldsymbol{Z}^{(p)} = \boldsymbol{W}^{(p)} \boldsymbol{X}^{(p)} + \boldsymbol{B}^{(p)} \tag{31}$$

We still omit superscription of $(p)$ for dimensions for simplicity. We also denote $\boldsymbol{z}^{(p)}$ as the vector form of $\boldsymbol{Z}^{(p)}$ and has size $mP$. Similar to fully connected layer, we have analogue of Eq. (17) for convolutional layer as

$$\boldsymbol{H}_\ell(\boldsymbol{w}^{(p)}, \boldsymbol{X}) = \left( \boldsymbol{I}_m \otimes \boldsymbol{X}^{(p)} \right) \boldsymbol{M}_{\boldsymbol{x}}^{(p)} \left( \boldsymbol{I}_m \otimes \boldsymbol{X}^{(p)T} \right), \tag{32}$$

where $\boldsymbol{M}_{\boldsymbol{x}}^{(p)} = \boldsymbol{H}_\ell(\boldsymbol{z}^{(p)}, \boldsymbol{X})$ and is a $mP \times mP$ matrix. Also, since convolutional layers can also be considered as linear operations (matrix multiplication with reshape) together with FC layers and ReLUs, Eq. (23) still holds. Thus, we still have

$$\boldsymbol{H}_\ell(\boldsymbol{z}^{(p)}, \boldsymbol{X}) = \boldsymbol{M}_{\boldsymbol{x}}^{(p)} = \boldsymbol{G}_{\boldsymbol{x}}^{(p)T} \boldsymbol{A}_{\boldsymbol{x}} \boldsymbol{G}_{\boldsymbol{x}}^{(p)}, \tag{33}$$

where $\boldsymbol{G}_{\boldsymbol{x}}^{(p)} = \frac{\partial \boldsymbol{z}}{\partial \boldsymbol{z}^{(p)}}$ and has dimension $c \times mP$, although is cannot be further decomposed as direct multiplication of weight matrices as in the FC layers.

However, for convolutional layers, $\boldsymbol{X}^{(p)}$ is a matrix instead of a vector. Thus, we cannot make Eq. (32) into the form of a Kronecker product as in Eq. (17).

Despite this, it is still possible to have a Kronecker factorization of the weight Hessian in the form

$$\boldsymbol{H}_\ell(\boldsymbol{w}^{(p)}, \boldsymbol{X}) \approx \tilde{\boldsymbol{M}}_{\boldsymbol{x}}^{(p)} \otimes \boldsymbol{X}^{(p)} \boldsymbol{X}^{(p)T}, \tag{34}$$

using further approximation motivated by Grosse & Martens (2016). Note that $\tilde{\boldsymbol{M}}_{\boldsymbol{x}}^{(p)}$ need to have a different shape $(m \times m)$ from $\boldsymbol{M}_{\boldsymbol{x}}^{(p)}$ ($mP \times mP$), since $\boldsymbol{H}_\ell(\boldsymbol{w}^{(p)}, \boldsymbol{X})$ is $mnK1K2 \times mnK1K2$ and $\boldsymbol{X}^{(p)} \boldsymbol{X}^{(p)T}$ is $nK1K2 \times nK1K2$.

Since we can further decompose $\boldsymbol{A}_{\boldsymbol{x}} = \boldsymbol{Q}_{\boldsymbol{x}}^T \boldsymbol{Q}_{\boldsymbol{x}}$, we then have

$$\boldsymbol{M}_{\boldsymbol{x}}^{(p)} = \boldsymbol{G}_{\boldsymbol{x}}^{(p)T} \boldsymbol{A}_{\boldsymbol{x}} \boldsymbol{G}_{\boldsymbol{x}}^{(p)} = \left( \boldsymbol{Q}_{\boldsymbol{x}} \boldsymbol{G}_{\boldsymbol{x}}^{(p)} \right)^T \left( \boldsymbol{Q}_{\boldsymbol{x}} \boldsymbol{G}_{\boldsymbol{x}}^{(p)} \right). \tag{35}$$

We define $\boldsymbol{N}_{\boldsymbol{x}}^{(p)} = \boldsymbol{Q}_{\boldsymbol{x}} \boldsymbol{G}_{\boldsymbol{x}}^{(p)}$. Here $\boldsymbol{Q}_{\boldsymbol{x}}$ is $c \times c$ and $\boldsymbol{G}_{\boldsymbol{x}}^{(p)}$ is $c \times mP$ so that $\boldsymbol{N}_{\boldsymbol{x}}^{(p)}$ is $c \times mP$. We can reshape $\boldsymbol{N}_{\boldsymbol{x}}^{(p)}$ into a $cP \times m$ matrix $\tilde{\boldsymbol{N}}_{\boldsymbol{x}}^{(p)}$. We then reduce $\boldsymbol{M}_{\boldsymbol{x}}^{(p)}$ ($mP \times mP$) into a $m \times m$ matrix as

$$\tilde{\boldsymbol{M}}_{\boldsymbol{x}}^{(p)} = \frac{1}{P} \tilde{\boldsymbol{N}}_{\boldsymbol{x}}^{(p)T} \tilde{\boldsymbol{N}}_{\boldsymbol{x}}^{(p)}. \tag{36}$$

The scalar $\frac{1}{P}$ is a normalization factor since we squeeze a dimension of size $P$ into size 1.

Thus, we can have similar Kronecker factorization approximation as

$$\boldsymbol{H}_{\mathcal{L}}(\boldsymbol{w}^{(p)}) = \mathbb{E}\left[ \boldsymbol{H}_\ell(\boldsymbol{w}^{(p)}, \boldsymbol{X}) \right] = \mathbb{E}\left[ \left( \boldsymbol{I}_m \otimes \boldsymbol{X}^{(p)} \right) \boldsymbol{M}_{\boldsymbol{x}}^{(p)} \left( \boldsymbol{I}_m \otimes \boldsymbol{X}^{(p)T} \right) \right] \tag{37}$$

$$\approx \mathbb{E}\left[ \tilde{\boldsymbol{M}}_{\boldsymbol{x}}^{(p)} \otimes \boldsymbol{X}^{(p)} \boldsymbol{X}^{(p)T} \right] \approx \mathbb{E}\left[ \tilde{\boldsymbol{M}}_{\boldsymbol{x}}^{(p)} \right] \otimes \mathbb{E}\left[ \boldsymbol{X}^{(p)} \boldsymbol{X}^{(p)T} \right]. \tag{38}$$

## B  Structure of Dominating Eigenvectors of the Full Hessian.

Although it is not possible to apply Kronecker factorization to the full Hessian directly, we can construct an approximation of the top eigenvectors and eigenspace using similar ideas and our findings.

In this section, we will always have superscript $(p)$ for all layer-wise matrices and vectors in order to distinguish them from the full versions.

As shown in Appendix A.1 Eq. (30), we have the full Hessian of fully connected networks as

$$\boldsymbol{H}_{\mathcal{L}}(\boldsymbol{\theta}) = \mathbb{E}\left[\boldsymbol{F}_{\boldsymbol{x}}^T \boldsymbol{A}_{\boldsymbol{x}} \boldsymbol{F}_{\boldsymbol{x}}\right] + \mathbb{E}\left[\sum_{i=1}^c \frac{\partial \ell(\boldsymbol{z}, \boldsymbol{y})}{z_i} \nabla_{\boldsymbol{\theta}}^2 z_i\right], \tag{39}$$

where

$$\boldsymbol{F}_{\boldsymbol{x}}^T = \begin{pmatrix} \boldsymbol{G}_{\boldsymbol{x}}^{(1)T} \otimes \boldsymbol{x}^{(1)} \\ \boldsymbol{G}_{\boldsymbol{x}}^{(1)T} \\ \boldsymbol{G}_{\boldsymbol{x}}^{(2)T} \otimes \boldsymbol{x}^{(2)} \\ \boldsymbol{G}_{\boldsymbol{x}}^{(2)T} \\ \vdots \\ \boldsymbol{G}_{\boldsymbol{x}}^{(L)T} \otimes \boldsymbol{x}^{(n)} \\ \boldsymbol{G}_{\boldsymbol{x}}^{(L)T} \end{pmatrix}. \tag{40}$$

In order to simplify the formula, we define

$$\tilde{\boldsymbol{x}}^{(p)} = \begin{pmatrix} \boldsymbol{x}^{(p)} \\ 1 \end{pmatrix} \tag{41}$$

to be the extended input of the $p$-th layer. Thus, the terms in the Hessian attributed to the bias can be included in the Kronecker product with the extended input, and $\boldsymbol{F}_{\boldsymbol{x}}^T$ can be simplified as

$$\boldsymbol{F}_{\boldsymbol{x}}^T = \begin{pmatrix} \boldsymbol{G}_{\boldsymbol{x}}^{(1)T} \otimes \tilde{\boldsymbol{x}}^{(1)} \\ \boldsymbol{G}_{\boldsymbol{x}}^{(2)T} \otimes \tilde{\boldsymbol{x}}^{(2)} \\ \vdots \\ \boldsymbol{G}_{\boldsymbol{x}}^{(L)T} \otimes \tilde{\boldsymbol{x}}^{(n)} \end{pmatrix}. \tag{42}$$

As discussed in several previous works (Sagun et al., 2016; Papyan, 2018, 2019; Fort & Ganguli, 2019), the full Hessian can be decomposed in to the G-term and the H-term. Specifically, the G-term is $\mathbb{E}\left[\boldsymbol{F}_{\boldsymbol{x}}^T \boldsymbol{A}_{\boldsymbol{x}} \boldsymbol{F}_{\boldsymbol{x}}\right]$, and the H-term is $\mathbb{E}\left[\sum_{i=1}^c \frac{\partial \ell(\boldsymbol{z}, \boldsymbol{y})}{z_i} \nabla_{\boldsymbol{\theta}}^2 z_i\right]$ in Eq. (39).

Empirically, the G-term usually dominates the H-term, and the top eigenvalues and eigenspace of the Hessian are mainly attributed to the G-term. Since we focus on the top eigenspace, we can approximate our full Hessian using the G-term, as

$$\boldsymbol{H}_{\mathcal{L}}(\boldsymbol{\theta}) \approx \mathbb{E}\left[\boldsymbol{F}_{\boldsymbol{x}}^T \boldsymbol{A}_{\boldsymbol{x}} \boldsymbol{F}_{\boldsymbol{x}}\right]. \tag{43}$$

In our approximation of the layer-wise Hessian $\boldsymbol{H}_{\mathcal{L}}(\boldsymbol{w}^{(p)})$ Eq. (6), the two parts of the Kronecker factorization are the layer-wise output Hessian $\mathbb{E}[\boldsymbol{M}_{\boldsymbol{x}}^{(p)}]$ and the auto-correlation matrix of the input $\mathbb{E}[\boldsymbol{x}^{(p)}\boldsymbol{x}^{(p)T}]$. Although we cannot apply Kronecker factorization to $\mathbb{E}\left[\boldsymbol{F}_{\boldsymbol{x}}^T \boldsymbol{A}_{\boldsymbol{x}} \boldsymbol{F}_{\boldsymbol{x}}\right]$, we can still approximate its eigenspace using the eigenspace of the full output Hessian.

Note here that the full output Hessian is not a common definition. Let $\hat{m} = \sum_{p=1}^L m^{(p)}$ be the sum of output dimension of each layer. We define a full output vector $\tilde{\boldsymbol{z}} \in \mathbb{R}^{\hat{m}}$ by concatenating all the layerwise outputs together,

$$\tilde{\boldsymbol{z}} := \begin{pmatrix} \boldsymbol{z}^{(1)} \\ \boldsymbol{z}^{(2)} \\ \vdots \\ \boldsymbol{z}^{(L)} \end{pmatrix}. \tag{44}$$

We then define the full output Hessian is the Hessian w.r.t. $\tilde{z}$. Let the full output Hessian for a single input $x$ be $M_x \in \mathbb{R}^{\hat{m} \times \hat{m}}$. Similar to Eq. (26), it can be expressed as

$$M_x := H_\ell(\tilde{z}, x) = G_x^T A_x G_x, \tag{45}$$

where

$$G_x^T = \begin{pmatrix} G_x^{(1)T} \\ G_x^{(2)T} \\ \vdots \\ G_x^{(L)T} \end{pmatrix} \tag{46}$$

similar to Eq. (42). The full output Hessian for the entire training sample is thus

$$H_\mathcal{L}(\tilde{z}) = \mathbb{E}[M_x] = \mathbb{E}[G_x^T A_x G_x]. \tag{47}$$

We can then approximate the eigenvectors of the full Hessian $H_\mathcal{L}(\theta)$ using the eigenvectors of $\mathbb{E}[M_x]$. Let the $i$-th eigenvector of $H_\mathcal{L}(\theta)$ be $v_i$ and that of $\mathbb{E}[M_x]$ be $u_i$. We may then break up $u_i$ into segments corresponding to different layers as in

$$u_i = \begin{pmatrix} u_i^{(1)} \\ u_i^{(2)} \\ \vdots \\ u_i^{(L)} \end{pmatrix}, \tag{48}$$

where for all layer $p$, $u_i^{(p)} \in \mathbb{R}^{m^{(p)}}$. Motivated by the relation between $G_x$ and $F_x$, the $i$-th eigenvector of $H_\mathcal{L}(\theta)$ can be approximated as the following. Let

$$w_i = \begin{pmatrix} u_i^{(1)} \otimes \mathbb{E}[x^{\tilde{(1)}}] \\ u_i^{(2)} \otimes \mathbb{E}[x^{\tilde{(2)}}] \\ \vdots \\ u_i^{(L)} \otimes \mathbb{E}[x^{\tilde{(L)}}] \end{pmatrix}. \tag{49}$$

We then have

$$v_i \approx \frac{w_i}{\|w_i\|} \tag{50}$$

We can then use the Gram–Schmidt process to get the basis vectors of the approximated eigenspace.

Another reason for this approximation is that the expectation is the input of each layer $\mathbb{E}[x^{(p)}]$ dominates its covariance as shown in Appendix E.1. Thus, the approximate is accurate for top eigenvectors and also top eigenspace. For latter eigenvectors, the approximation would not be as accurate since this approximate loses all information in the covariance of the inputs.

We also approximated the eigenvalues using this approximation. Let the $i$-th eigenvalue of $H_\mathcal{L}(\theta)$ be $\lambda_i$ and that of $\mathbb{E}[M_x]$ be $\sigma_i$. We have

$$\lambda_i \approx \sigma_i \|w_i\|^2. \tag{51}$$

Below we show the approximation of the eigenvalues top eigenspace using this method. The eigenspace overlap is defined as in Definition 4.1. We experimented on several fully connected networks, the results shown below are for F-$200^2$ (same as Fig. 2d in the main text), F-$200^4$, F-$600^4$, and F-$600^8$, all with dimension 50.

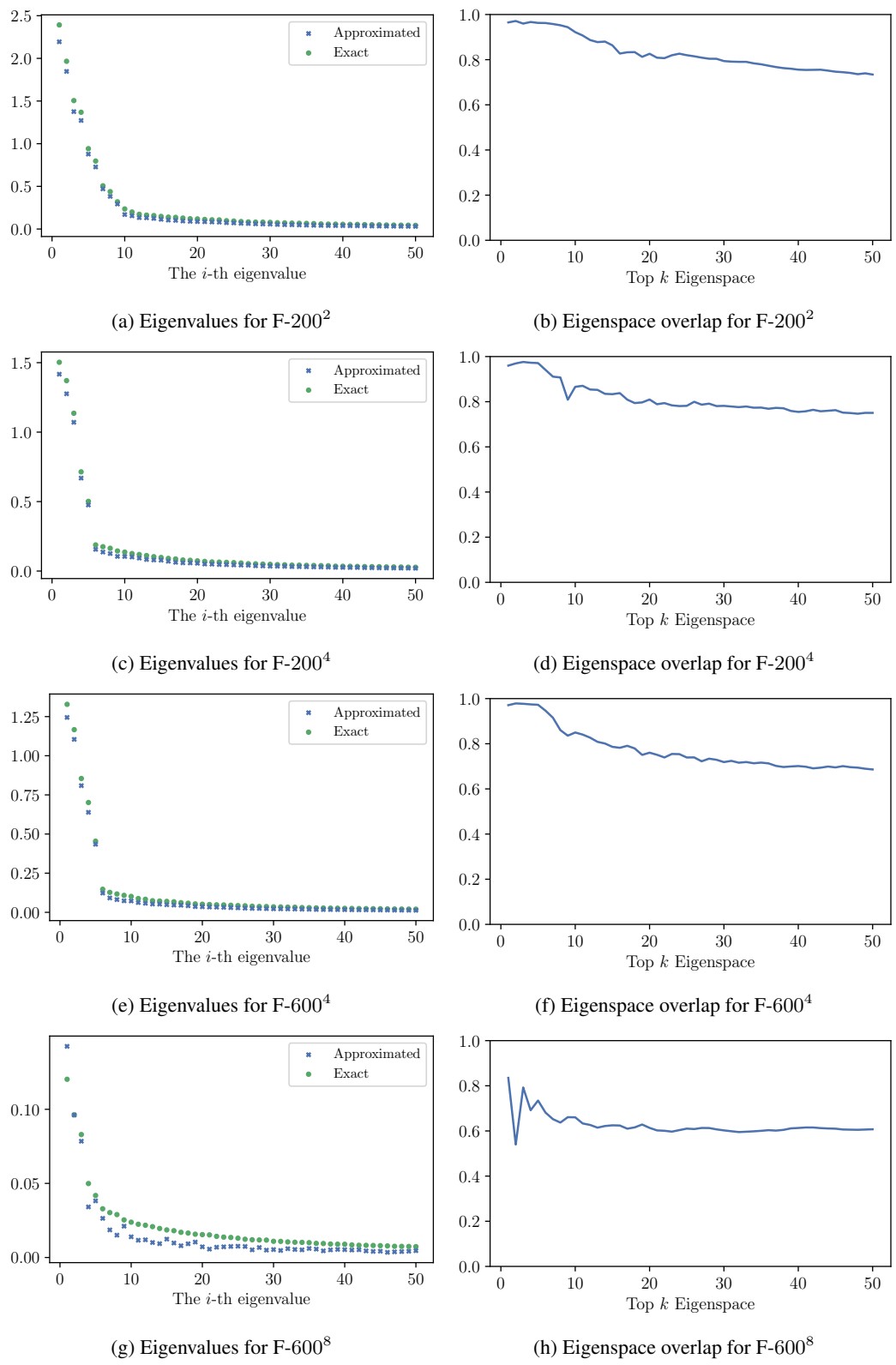

(a) Eigenvalues for F-200$^2$

(b) Eigenspace overlap for F-200$^2$

(c) Eigenvalues for F-200$^4$

(d) Eigenspace overlap for F-200$^4$

(e) Eigenvalues for F-600$^4$

(f) Eigenspace overlap for F-600$^4$

(g) Eigenvalues for F-600$^8$

(h) Eigenspace overlap for F-600$^8$

Figure 8: Top 50 Eigenvalues and Eigenspace approximation for full Hessian

## C   Computation of Hessian Eigenvalues and Eigenvectors

For Hessian approximated using Kronecker factorization, we compute $\mathbb{E}[\boldsymbol{M}]$ and $\mathbb{E}[\boldsymbol{x}\boldsymbol{x}^T]$ explicitly. Let $\boldsymbol{m}$ and $\boldsymbol{v}$ be an eigenvector of $\mathbb{E}[\boldsymbol{M}]$ and $\mathbb{E}[\boldsymbol{x}\boldsymbol{x}^T]$ respectively, with corresponding eigenvalues $\lambda_{\boldsymbol{m}}$ and $\lambda_{\boldsymbol{v}}$. Since both matrices are positive semi-definite, $\boldsymbol{m} \otimes \boldsymbol{v}$ is an eigenvector of $\mathbb{E}[\boldsymbol{M}] \otimes \mathbb{E}[\boldsymbol{x}\boldsymbol{x}^T]$ with eigenvalue $\lambda_{\boldsymbol{m}}\lambda_{\boldsymbol{v}}$. In this way, since $\mathbb{E}[\boldsymbol{M}]$ has $m$ eigenvectors and $\mathbb{E}[\boldsymbol{x}\boldsymbol{x}^T]$ has $n$ eigenvectors, we can approximate all $mn$ eigenvectors for the layer-wise Hessian. All these calculation can be done directly.

However, it is almost prohibitive to calculate the true Hessian explicitly. Thus, we use numerical methods with automatic differentiation (Paszke et al., 2017) to calculate them. The packages we use is Golmant et al. (2018) and we use the Lanczos method in most of the calculations. We also use package in Yao et al. (2019) as a reference.

For layer-wise Hessian, we modified the Golmant et al. (2018) package. In particular, the package relies on the calculation of Hessian-vector product $\boldsymbol{H}\boldsymbol{v}$, where $\boldsymbol{v}$ is a vector with the same size as parameter $\theta$. To calculate eigenvalues and eigenvectors for layer-wise Hessian at the $p$-th layer, we cut the $\boldsymbol{v}$ into different layers. Then, we only leave the part corresponding to weights of the $p$-th layer and set all other entries to 0. Note that the dimension does not change. We let the new vector be $\boldsymbol{v}^{(p)}$ and get the value of $\boldsymbol{u} = \boldsymbol{H}\boldsymbol{v}^{(p)}$ using auto differentiation. Then, we do the same operation to $\boldsymbol{u}$ and get $\boldsymbol{u}^{(p)}$.

# D  Detailed Experiment Setup

## D.1  Datasets

We conduct experiment on CIFAR-10, CIFAR-100 (MIT) (Krizhevsky, 2009) (`https://www.cs.toronto.edu/~kriz/cifar.html`), and MNIST (CC BY-SA 3.0) (LeCun et al., 1998) (`http://yann.lecun.com/exdb/mnist/`). The datasets are downloaded through torchvision (Paszke et al., 2019) (`https://pytorch.org/vision/stable/index.html`). We used their default splitting of training and testing set.

To compare our work on PAC-Bayes bound with the work of Dziugaite & Roy (2017), we created a custom dataset MNIST-2 by setting the label of images 0-4 to 0 and 5-9 to 1. We also created random-labeled datasets MNIST-R and CIFAR10-R by randomly labeling the images from the training set of MNIST and CIFAR10. The dataset information is summarized in Table 2

Table 2: Datasets

| Dataset | # Data Points | | Input Size | # Classes | Label |
| | Train | Test | | | |
| --- | --- | --- | --- | --- | --- |
| CIFAR10 | 50000 | 10000 | $3 \times 32 \times 32$ | 10 | True |
| CIFAR10-R | 50000 | 10000 | $3 \times 32 \times 32$ | 10 | Random |
| CIFAR100 | 50000 | 10000 | $3 \times 32 \times 32$ | 100 | True |
| MNIST | 60000 | 10000 | $28 \times 28$ | 10 | True |
| MNIST-2 | 60000 | 10000 | $28 \times 28$ | 2 | True |
| MNIST-R | 60000 | 10000 | $28 \times 28$ | 10 | Random |

All the datasets (MNIST, CIFAR-10, and CIFAR-100) we used are publicly available. According to their descriptions on the contents and collection methods, they should not contain any personal information or offensive content. MNIST is a remix of datasets from the National Institute of Standards and Technology (NIST), which obtained consent for collecting the data. However, we also note that CIFAR-10 and CIFAR-100 are subsets of the dataset 80 Million Tiny Image (Torralba et al., 2008) (`http://groups.csail.mit.edu/vision/TinyImages/`), which used automatic collection and includes some offensive images.

## D.2  Network Structures

**Fully Connected Network:**  We used several different fully connected networks varying in the number of hidden layers and the number of neurons for each hidden layer. The output of all layers except the last layer are passed into ReLU before feeding into the subsequent layer. As described in Section 4.1, we denote a fully connected network with $m$ hidden layers and $n$ neurons each hidden layer by F-$n^m$. For networks without uniform layer width, we denote them by a sequence of numbers (e.g. for a network with three hidden layers, where the first two layers has 200 neurons each and the third has 100 neurons, we denote it as F-$200^2$-100). For example, the structure of F-$200^2$ is shown in Table 3.

Table 3: Structure of F-$200^2$ on MNIST

| # | Name | Module | In Shape | Out Shape |
| --- | --- | --- | --- | --- |
| 1 | | Flatten | (28,28) | 784 |
| 2 | fc1 | Linear(784, 200) | 784 | 200 |
| 3 | | ReLU | 200 | 200 |
| 4 | fc2 | Linear(200, 200) | 200 | 200 |
| 5 | | ReLU | 200 | 200 |
| 6 | fc3 | Linear(200, 10) | 200 | 10 |
| | | *output* | | |

**LeNet5:**   We adopted the LeNet5 structure proposed by LeCun et al. (1998) for MNIST, and slightly modified the input convolutional layers to adapt the input of CIFAR-10 dataset. The standard LeNet5 structure we used in the experiments is shown in Table 4. We further modified the dimension of fc1 and conv2 to create several variants for the experiment in Section 5.1. Take the model whose first fully connected layer is adjusted to have 80 neurons as an example, we denote it as LeNet5-(fc1-80).

Table 4: Structure of LeNet5 on CIFAR-10

| # | Name | Module | In Shape | Out Shape |
|---|------|--------|----------|-----------|
| 1 | conv1 | Conv2D(3, 6, 5, 5) | (3, 32, 32) | (6, 28, 28) |
| 2 | | ReLU | (6, 28, 28) | (6, 28, 28) |
| 3 | maxpool1 | MaxPooling2D(2,2) | (6, 28, 28) | (6, 14, 14) |
| 4 | conv2 | Conv2D(6, 16, 5, 5) | (6, 14, 14) | (16, 10, 10) |
| 5 | | ReLU | (16, 10, 10) | (16, 10, 10) |
| 6 | maxpool2 | MaxPooling2D(2,2) | (16, 10, 10) | (16, 5, 5) |
| 7 | | Flatten | (16, 5, 5) | 400 |
| 8 | fc1 | Linear(400, 120) | 400 | 120 |
| 9 | | ReLU | 120 | 120 |
| 10 | fc2 | Linear(120, 84) | 120 | 84 |
| 11 | | ReLU | 84 | 84 |
| 12 | fc3 | Linear(84, 10) | 84 | 10 |
| | | *output* | | |

**Networks with Batch Normalization:**   In Appendix F.4 we conducted several experiments regarding the effect of batch normalization on our results. For those experiments, we use the existing structures and add batch normalization layer for each intermediate output after it passes the ReLU module. In order for the Hessian to be well-defined, we fix the running statistics of batch normalization and treat it as a linear layer during inference. We also turn off the learnable parameters $\theta$ and $\beta$ (Ioffe & Szegedy, 2015) for simplicity. For network structure X, we denote the variant with batch normalization after all hidden layers X-BN. For example, the detailed structure LeNet5-BN is shown in Table 5.

Table 5: Structure of LeNet5-BN on CIFAR-10

| # | Name | Module | In Shape | Out Shape |
|---|------|--------|----------|-----------|
| 1 | conv1 | Conv2D(3, 6, 5, 5) | (3, 32, 32) | (6, 28, 28) |
| 2 | | ReLU | (6, 28, 28) | (6, 28, 28) |
| 3 | | BatchNorm2D | (6, 28, 28) | (6, 28, 28) |
| 4 | maxpool1 | MaxPooling2D(2,2) | (6, 28, 28) | (6, 14, 14) |
| 5 | conv2 | Conv2D(6, 16, 5, 5) | (6, 14, 14) | (16, 10, 10) |
| 6 | | ReLU | (16, 10, 10) | (16, 10, 10) |
| 7 | | BatchNorm2D | (16, 10, 10) | (16, 10, 10) |
| 8 | maxpool2 | MaxPooling2D(2,2) | (16, 10, 10) | (16, 5, 5) |
| 9 | | Flatten | (16, 5, 5) | 400 |
| 10 | fc1 | Linear(400, 120) | 400 | 120 |
| 11 | | ReLU | 120 | 120 |
| 12 | | BatchNorm1D | 120 | 120 |
| 13 | fc2 | Linear(120, 84) | 120 | 84 |
| 14 | | ReLU | 84 | 84 |
| 15 | | BatchNorm1D | 84 | 84 |
| 16 | fc3 | Linear(84, 10) | 84 | 10 |
| | | *output* | | |

**Variants of VGG11:** To verify that our results apply to larger networks, we trained a number of variant of VGG11 (originally named VGG-A in the paper, but commonly refered as VGG11) proposed by Simonyan & Zisserman (2015). For simplicity, we removed the dropout regularization in the original network. To adapt the structure, which is originally designed for the $3 \times 224 \times 224$ input of ImageNet, to $3 \times 32 \times 32$ input of CIFAR-10.

Since the original VGG11 network is too large for computing the top eigenspace up to hundreds of dimensions, we reduce the number of output channels of each convolution layer in the network to 32, 48, 64, 80, and 200. We denote the small size variants as VGG11-W32, VGG11-W48, VGG11-W64, VGG11-W80, and VGG11-W200 respectively. We use conv1 - conv8 and fc1 to denote the layers of VGG11 where conv1 is closest to the input feature and fc1 is the classification layer.

**Variants of ResNet18:** We also trained a number of variant of ResNet18 proposed by He et al. (2016). As batch normalization will change the low rank structure of the auto correlation matrix and reduce the overlap, we removed all batch normalization operations. Following the adaptation of ResNet to CIFAR dataset as in `https://github.com/kuangliu/pytorch-cifar`, we changed the input size to $3 \times 32 \times 32$ and added a 1x1 convolutional layer for each shortcut after the first block.

Similar to VGG11, we reduce the number of output channels of each convolution layer in the network to 48, 64, 80. We denote the small size variants as ResNet18-W48, ResNet18-W64, and ResNet18-W80 respectively.

We use conv1 - conv17 and fc1 to denote the layers of the ResNet18 backbone where conv1 is closest to the input feature and fc1 is the classification layer. For the 1x1 convolutional layers in the shortcut, we denote them by sc-conv1 - sc-conv3. where sc-conv1 is the convolutional layer on the shortcut of the second ResNet block and sc-conv3 is the convolutional layer on the shortcut of the fourth ResNet block.

### D.3 Training Process and Hyperparameter Configuration

For all datasets, we used the default splitting of training and testing set. All models (except explicitly stated otherwise) are trained using batched stochastic gradient descent (SGD) with batch-size 128 and fixed learning rate 0.01 for 1000 epochs. No momentum and weight decay regularization were used. The loss objective converges by the end of training, so we may assume that the final models are at local minima.

For generality we also used a training scheme with fixed learning rate at 0.001, and a training scheme with fixed learning rate at 0.01 with momentum of 0.9 and weight-decay factor of 0.0005. Models trained with these settings will be explicitly stated. Otherwise we assume they were trained with the default scheme mentioned above.

Follow the default initialization scheme of PyTorch(Paszke et al., 2019), the weights of linear layers and convolutional layers are initialized using the Xavier method (Glorot & Bengio, 2010), and bias of each layer are initialized to be zero.

# E  Additional Experiment Results

## E.1  Low Rank Structure of Auto-correlation Matrix $\mathbb{E}[\boldsymbol{x}\boldsymbol{x}^T]$

We have briefly discussed about the autocorrelation matrix $\mathbb{E}[\boldsymbol{x}\boldsymbol{x}^T]$ being approximately rank 1 in Section 4.3 in the main text. In particular, we claimed that the mean of layer input dominate the covariance, that $\mathbb{E}[\boldsymbol{x}\boldsymbol{x}^T] \approx \mathbb{E}[\boldsymbol{x}]\mathbb{E}[\boldsymbol{x}^T]$. In this section we provide some additional empirical results supporting that claim.

We use two metrics to quantify the quality of this approximation: the squared dot product between normalized $\mathbb{E}[\boldsymbol{x}]$ and the first eigenvector of $\mathbb{E}[\boldsymbol{x}\boldsymbol{x}^T]$ and the ratio between the first and second eigenvalue of $\mathbb{E}[\boldsymbol{x}\boldsymbol{x}^T]$. Intuitively if the first quantity is close to 1 and the second quantity is large, then the approximation is accurate.

Formally, for fully connected layers, define $\hat{\mathbb{E}}[\boldsymbol{x}]$ as the normalized expectation of the layer input $\boldsymbol{x}$, namely $\mathbb{E}[\boldsymbol{x}]/\|\mathbb{E}[\boldsymbol{x}]\|$. For convolutional layers, following the notations in Appendix A.2, define $\hat{\mathbb{E}}[\boldsymbol{x}]$ as the first left singular vector of $\mathbb{E}[\boldsymbol{X}]$ where $\hat{\mathbb{E}}[\boldsymbol{x}] \in \mathbb{R}^{nK_1 K_2}$. Abusing notations for simplicity, we use $\mathbb{E}[\boldsymbol{x}\boldsymbol{x}^T]$ to denote the $nK_1 K_2 \times nK_1 K_2$ matrix $\mathbb{E}[\boldsymbol{X}\boldsymbol{X}^T]$. In this section we consider the squared dot product between $\hat{\mathbb{E}}[\boldsymbol{x}]$ and the first eigenvector $v_1$ of $\mathbb{E}[\boldsymbol{x}\boldsymbol{x}^T]$, namely $(v_1^T \hat{\mathbb{E}}[\boldsymbol{x}])^2$.

For the spectral ratio, let $\lambda_1$ be the first eigenvalue of $\mathbb{E}[\boldsymbol{x}\boldsymbol{x}^T]$ and $\lambda_2$ be the second. We have

$$\frac{\lambda_1}{\lambda_2} \geq \frac{\|\mathbb{E}[\boldsymbol{x}]\mathbb{E}[\boldsymbol{x}]^T\| - \|\boldsymbol{\Sigma_x}\|}{\|\boldsymbol{\Sigma_x}\|} = \frac{\|\mathbb{E}[\boldsymbol{x}]\mathbb{E}[\boldsymbol{x}]^T\|}{\|\boldsymbol{\Sigma_x}\|} - 1, \tag{52}$$

where $\boldsymbol{\Sigma_x}$ is the covariance of $\boldsymbol{x}$. Thus, the spectral norm of $\mathbb{E}[\boldsymbol{x}]\mathbb{E}[\boldsymbol{x}]^T$ divided by that of $\boldsymbol{\Sigma_x}$ gives a lower bound to $\lambda_1/\lambda_2$. In our experiments, we usually have $\lambda_1/\lambda_2 \geq \|\mathbb{E}[\boldsymbol{x}]\mathbb{E}[\boldsymbol{x}]^T\|/\|\boldsymbol{\Sigma_x}\|$.

Table 6: Squared dot product $(v_1^T \hat{\mathbb{E}}[\boldsymbol{x}])^2$ and spectral ratio $\lambda_1/\lambda_2$ for fully connected layers in a selection of network structures and datasets. We independently trained 5 runs for each instance and compute the mean, minimum, and maximum of the two quantities over all layers (except the first layer which takes the input with mean-zero) in all runs.

| Dataset | Network | # fc | $(v_1^T \hat{\mathbb{E}}[\boldsymbol{x}])^2$ | | | $\lambda_1/\lambda_2$ | | |
|---|---|---|---|---|---|---|---|---|
| | | | mean | min | max | mean | min | max |
| MNIST | F-$200^2$ | 2 | 1.000 | 1.000 | 1.000 | 12.29 | 9.65 | 16.16 |
| | F-$600^2$ | 2 | 0.999 | 0.999 | 0.999 | 12.00 | 11.42 | 13.00 |
| | F-$600^4$ | 4 | 1.000 | 0.999 | 1.000 | 17.81 | 7.33 | 28.00 |
| | F-$600^8$ | 8 | 0.991 | 0.965 | 1.000 | 6.63 | 2.28 | 11.15 |
| CIFAR10 | F-$600^2$ | 2 | 0.999 | 0.998 | 1.000 | 9.24 | 4.74 | 13.74 |
| | F-$1500^3$ | 3 | 0.999 | 0.997 | 1.000 | 13.27 | 6.10 | 18.41 |
| | LeNet5 | 3 | 0.998 | 0.997 | 0.999 | 7.21 | 5.88 | 9.02 |
| | LeNet5-(fc1-80) | 3 | 0.998 | 0.996 | 0.999 | 7.80 | 6.77 | 11.01 |
| | LeNet5-(fc1-100) | 3 | 0.997 | 0.995 | 0.999 | 7.42 | 6.20 | 9.10 |
| | LeNet5-(fc1-150) | 3 | 0.998 | 0.992 | 0.999 | 7.35 | 5.34 | 9.62 |
| | VGG11-W32 | 1 | 0.990 | 0.988 | 0.993 | 6.02 | 5.57 | 6.51 |
| | VGG11-W64 | 1 | 0.996 | 0.993 | 0.999 | 5.87 | 5.32 | 6.26 |
| | VGG11-W64 | 1 | 0.995 | 0.993 | 0.996 | 6.24 | 5.97 | 6.70 |
| CIFAR100 | VGG11-W48 | 1 | 0.999 | 0.999 | 0.999 | 17.861 | 15.456 | 20.491 |
| | VGG11-W64 | 1 | 0.999 | 0.999 | 1.000 | 19.185 | 18.358 | 20.410 |
| | VGG11-W80 | 1 | 0.999 | 0.999 | 1.000 | 19.455 | 18.120 | 21.450 |
| | ResNet18-W48 | 1 | 1.000 | 1.000 | 1.000 | 28.23 | 27.37 | 29.27 |
| | ResNet18-W64 | 1 | 1.000 | 1.000 | 1.000 | 27.07 | 25.72 | 29.50 |
| | ResNet18-W80 | 1 | 1.000 | 1.000 | 1.000 | 28.23 | 25.98 | 30.03 |

Table 7: Squared dot product $(v_1^T \hat{\mathbb{E}}[\boldsymbol{x}])^2$ and spectral ratio $\lambda_1/\lambda_2$ for convolutional layers in the selection of network structures and datasets in Table 6.

| Dataset | Network | # conv | $(v_1^T \hat{\mathbb{E}}[\boldsymbol{x}])^2$ | | | $\lambda_1/\lambda_2$ | | |
| | | | mean | min | max | mean | min | max |
|---------|---------|--------|------|-----|-----|------|-----|-----|
| CIFAR10 | LeNet5 | 1 | 0.999 | 0.998 | 0.999 | 15.87 | 11.15 | 27.20 |
| | LeNet5-(fc1-80) | 1 | 0.998 | 0.998 | 0.999 | 12.36 | 9.53 | 13.36 |
| | LeNet5-(fc1-100) | 1 | 0.999 | 0.999 | 0.999 | 19.49 | 16.69 | 21.92 |
| | LeNet5-(fc1-150) | 1 | 0.999 | 0.998 | 0.999 | 12.86 | 7.65 | 16.34 |
| | VGG11-W32 | 7 | 0.995 | 0.991 | 0.999 | 5.31 | 2.39 | 9.09 |
| | VGG11-W64 | 7 | 0.997 | 0.993 | 1.000 | 5.76 | 2.50 | 9.98 |
| | VGG11-W64 | 7 | 0.998 | 0.995 | 1.000 | 5.81 | 2.53 | 10.62 |
| CIFAR100 | VGG11-W48 | 7 | 0.996 | 0.991 | 0.999 | 5.72 | 2.46 | 9.90 |
| | VGG11-W64 | 7 | 0.995 | 0.991 | 0.999 | 5.66 | 2.50 | 10.79 |
| | VGG11-W80 | 7 | 0.994 | 0.988 | 0.998 | 5.18 | 2.50 | 8.45 |
| | ResNet18-W48 | 19 | 0.981 | 0.917 | 0.998 | 3.79 | 1.89 | 7.56 |
| | ResNet18-W64 | 19 | 0.985 | 0.910 | 0.998 | 3.96 | 1.81 | 7.53 |
| | ResNet18-W80 | 19 | 0.987 | 0.954 | 0.997 | 4.16 | 2.11 | 7.04 |

As we can see from Table 6 and Table 7, in a variety of settings, $\mathbb{E}[\boldsymbol{x}]\mathbb{E}[\boldsymbol{x}]^T$ indeed dominated the autocorrelation matrix $\mathbb{E}[\boldsymbol{x}\boldsymbol{x}^T]$ for fully connected layers. Similar phenomenon also holds for convolutional layers in the modern architectures, but the spectral gap are generally smaller compared to that of the fully connected layers.

### E.2 Eigenspace Overlap Between Different Models

The non trivial overlap between top eigenspaces of layer-wise Hessians is one of our interesting observations that had been discusses in Section 5.1. Here we provide more related empirical results. Some will further verify our claim in Section 5.1 and some will appear to be challenge that. Both results will be explained discussed more extensively in Appendix F.

#### E.2.1 Overlap preserved when varying hyper-parameters:

We first verify that the overlap also exists for a set of models trained with the different hyper-parameters. Using the LeNet5 (defined in Table 4) as the network structure. We train 6 models using the default training scheme (SGD, lr=0.01, momentum=0), 5 models using a smaller learning rate (SGD, lr=0.001, momentum=0), and 5 models using a combination of optimization tricks (SGD, lr=0.01, momentum=0.9, weight decay=0.0005). With these 16 models, we compute the pairwise eigenspace overlap of their layer-wise Hessians (120 pairs in total) and plot their average in Fig. 9. The shade areas in the figure represents the standard deviation. The pattern of overlap is clearly preserved, and the position of the peak roughly agrees with the output dimension $m$, demonstrating that the phenomenon is caused by a common structure instead of similarities in training process.

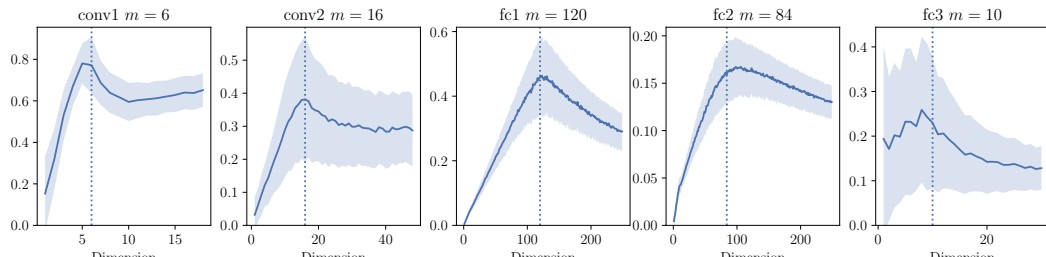

Figure 9: Eigenspace overlap of different models of LeNet5 trained with different hyper parameters.

Note that for fc3 (the final output layer), we are not observing a linear growth starting from 0 like other layers. This can be explained by the lack of neuron permutation. Related details will be discussed along with the reason for the linear growth pattern for other layers in Appendix F.3.

### E.2.2 Eigenspace overlap for convolutional layers in large models:

Even though the exact Kroneckor Factorization for layer-wise Hessians is only well-defined for fully connected layers, we also observe similar nontrivial eigenspace overlap for convolutional layers in larger and deeper networks including variants of VGG11 and ResNet18 on datasets CIFAR10 and CIFAR100. Some representative results are shown in Fig. 25 and Fig. 11. For each model on each dataset, we independently train 5 models and compute the average pairwise eigenspace overlap. The shade areas represents the standard deviation.

For most of the convolutional layers, the eigenspace overlap peaks around the dimension which is equal to the number of output channels of that layer, which is similar to the layers in LeNet5 as in Fig. 9. The eigenspace overlap of the final fully connected-layer also behaves similar to fc3:LeNet5, which remains around a constant then drops after exceeding the dimension of final output. However, there are also layers whose overlap does not peak around the output dimensions, (e.g. conv5 of Fig. 10b) and conv7 of Fig. 11a). We will cluster these "failure cases" in the following paragraph.

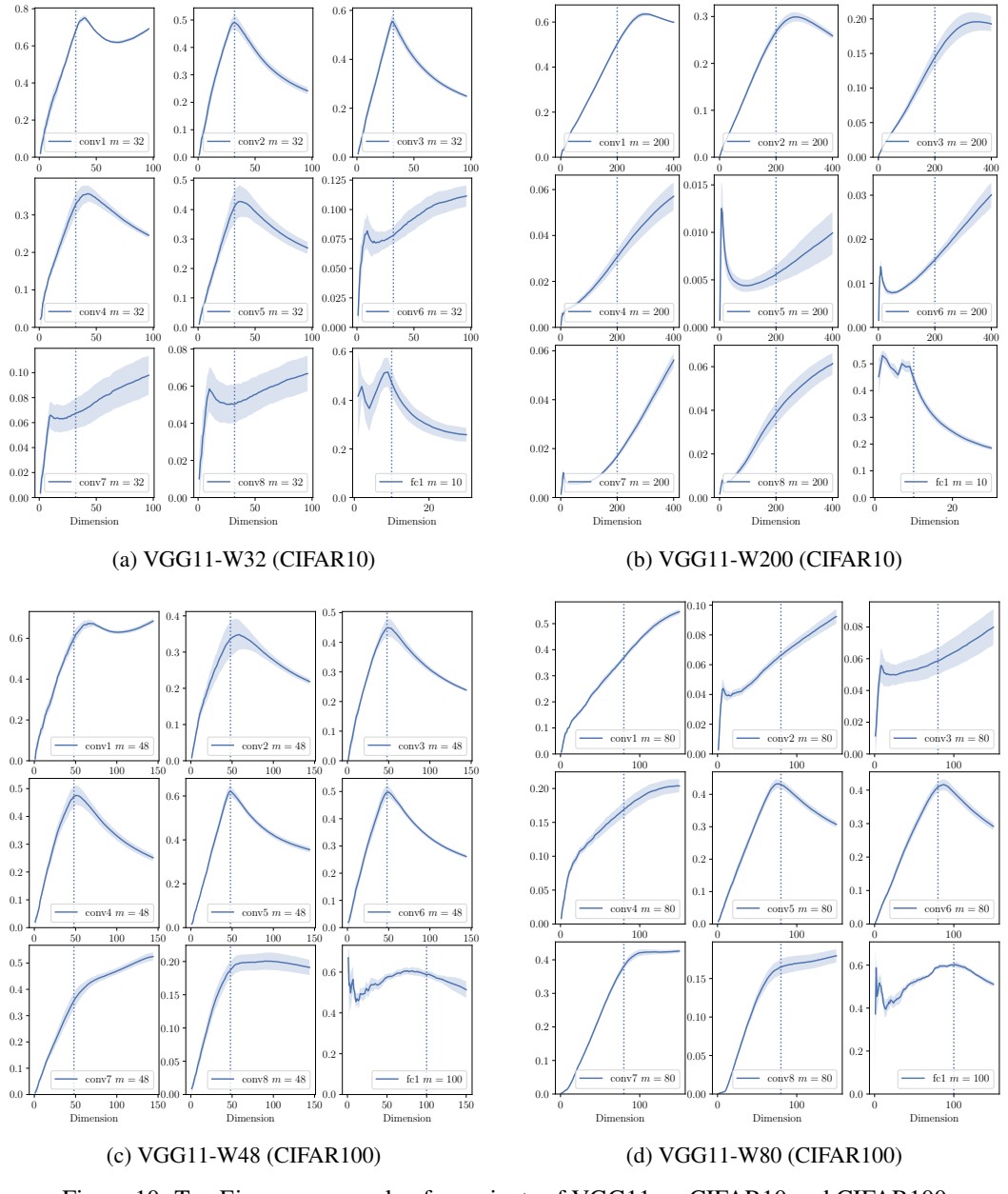

(a) VGG11-W32 (CIFAR10)  (b) VGG11-W200 (CIFAR10)

(c) VGG11-W48 (CIFAR100)  (d) VGG11-W80 (CIFAR100)

Figure 10: Top Eigenspace overlap for varients of VGG11 on CIFAR10 and CIFAR100

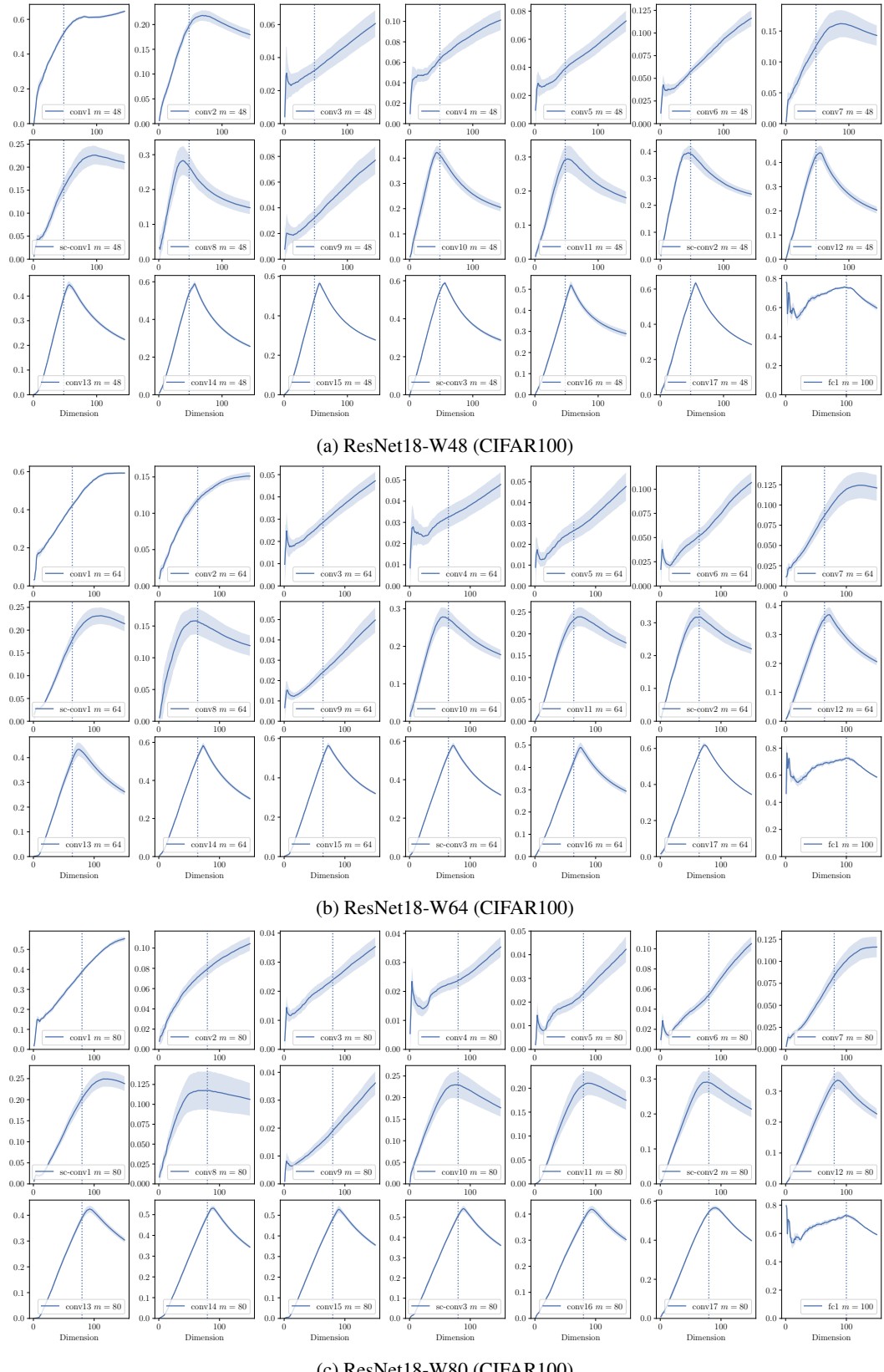

(a) ResNet18-W48 (CIFAR100)

(b) ResNet18-W64 (CIFAR100)

(c) ResNet18-W80 (CIFAR100)

Figure 11: Top Eigenspace overlap for variants of ResNet18 on CIFAR100

### E.2.3 Failure Cases

As seen in Fig. 25 and Fig. 11, there is a small portion of layers, usually closer to the input, whose eigenspace overlap does peak around the output dimensions. These layers can be clustered into the following two general cases.

**a. Early Peak of Low Overlap**

For layers shown in Fig. 12. The overlap of dominating eigenspaces are significantly lower than the other layers. Also there exists a small peak at very small dimensions.

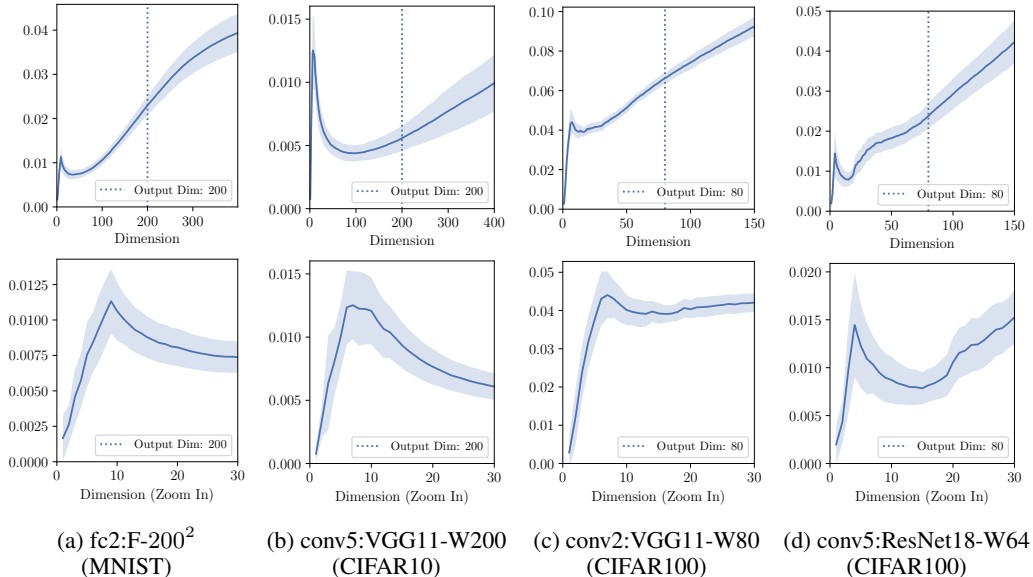

(a) fc2:F-200$^2$ (MNIST)  (b) conv5:VGG11-W200 (CIFAR10)  (c) conv2:VGG11-W80 (CIFAR100)  (d) conv5:ResNet18-W64 (CIFAR100)

Figure 12: Top eigenspace overlap for layers with an early low peak.
Figures in the second row are the zoomed in versions of the figures in the first row.

**b. Delayed Peak / Peak Doesn't Decline**

For layers shown in Fig. 13, the top eigenspaces has a nontrivial overlap, but the peak dimension is larger than predicted output dimension.

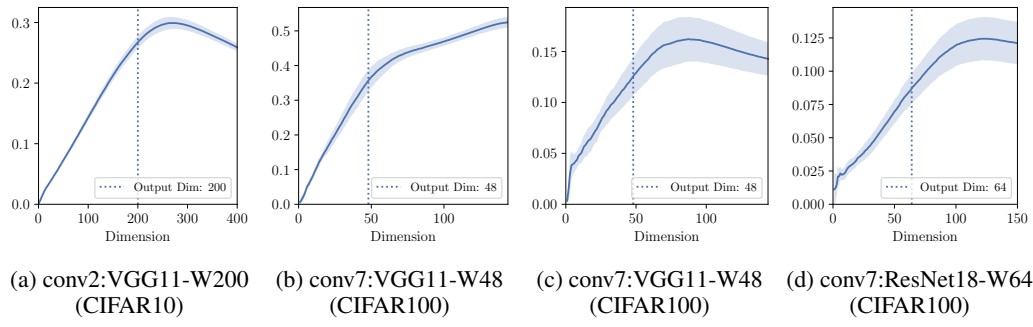

(a) conv2:VGG11-W200 (CIFAR10)  (b) conv7:VGG11-W48 (CIFAR100)  (c) conv7:VGG11-W48 (CIFAR100)  (d) conv7:ResNet18-W64 (CIFAR100)

Figure 13: Top eigenspace overlap for layers with a delayed peak.

However, the existence of such failure cases *does not* undermine the theory of Kronecker factorization approximation. In fact, both "failure cases" appear because the top hessian eigenspace is not completely spanned by $\mathbb{E}[x]$, and can be predicted by computing the auto correlation matrices and the output Hessians. The details will also be elaborated in Appendix F.3 with the help of correspondence matrices.

### E.3 Eigenvector Correspondence

Here we present the correspondence matrix for fc1, fc2, conv1, and conv2 layer of LeNet5. The top eigenvectors for all layers shows a strong correlation with the first eigenvector of $\mathbb{E}[\boldsymbol{x}\boldsymbol{x}^T]$ (which is approximately $\hat{\mathbb{E}}[\boldsymbol{x}]$). However, the diagonal pattern in the correspondence matrix with $\mathbb{E}[\boldsymbol{M}]$ for fc2 is not as clear as the one for fc1.

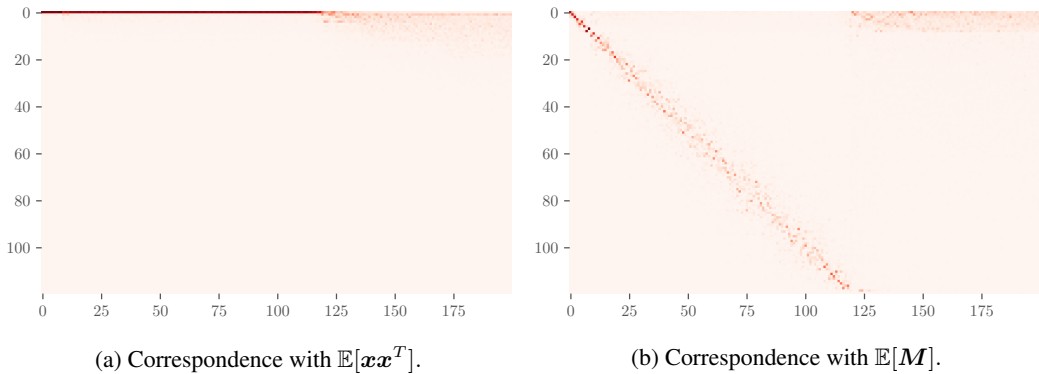

(a) Correspondence with $\mathbb{E}[\boldsymbol{x}\boldsymbol{x}^T]$.      (b) Correspondence with $\mathbb{E}[\boldsymbol{M}]$.

Figure 14: Eigenvector Correspondence for fc1:LeNet5. ($m$=120)

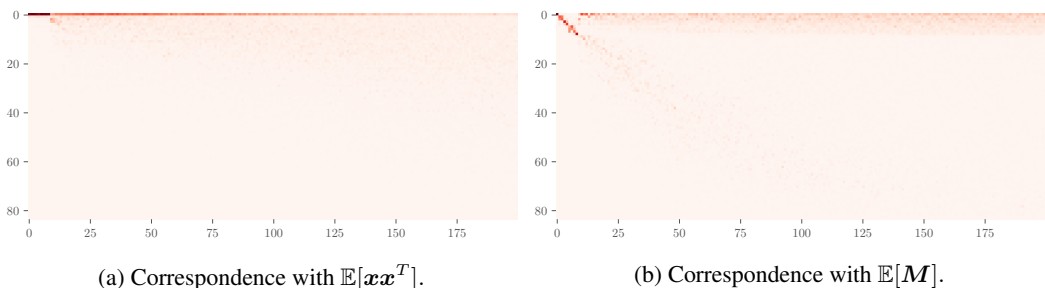

(a) Correspondence with $\mathbb{E}[\boldsymbol{x}\boldsymbol{x}^T]$.      (b) Correspondence with $\mathbb{E}[\boldsymbol{M}]$.

Figure 15: Eigenvector Correspondence for fc2:LeNet5. ($m$=84)

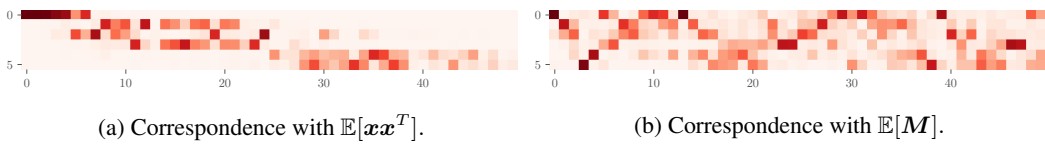

(a) Correspondence with $\mathbb{E}[\boldsymbol{x}\boldsymbol{x}^T]$.      (b) Correspondence with $\mathbb{E}[\boldsymbol{M}]$.

Figure 16: Eigenvector Correspondence for conv1:LeNet5. ($m$=6)

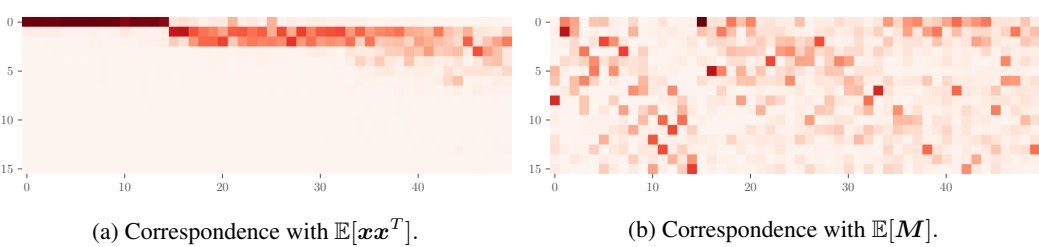

(a) Correspondence with $\mathbb{E}[\boldsymbol{x}\boldsymbol{x}^T]$.      (b) Correspondence with $\mathbb{E}[\boldsymbol{M}]$.

Figure 17: Eigenvector Correspondence for conv2:LeNet5. ($m$=16)

## E.4 Structure of $\mathbb{E}[\boldsymbol{x}\boldsymbol{x}^T]$ and $\mathbb{E}[\boldsymbol{M}]$ During Training

We observed the pattern of $\mathbb{E}[\boldsymbol{x}\boldsymbol{x}^T]$ matrix and $\mathbb{E}[\boldsymbol{M}]$ matrix along the training trajectory (Fig. 18, Fig. 19). It shows that $\mathbb{E}[\boldsymbol{x}\boldsymbol{x}^T]$ is always approximately rank 1, and $\mathbb{E}[\boldsymbol{M}]$ always have around $c$ large eigenvalues. According to our analysis, since the nontrivial eigenspace overlap is likely to be a consequence of a approximately rank 1 $\mathbb{E}[\boldsymbol{x}\boldsymbol{x}^T]$, we would conjecture that the overlap phenomenon is likely to happen on the training trajectory as well.

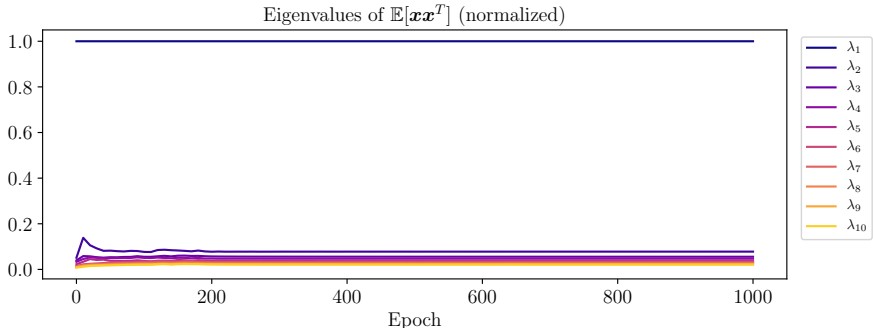

Figure 18: Top eigenvalues of $\mathbb{E}[\boldsymbol{x}\boldsymbol{x}^T]$ along training trajectory. (fc1:LeNet5)

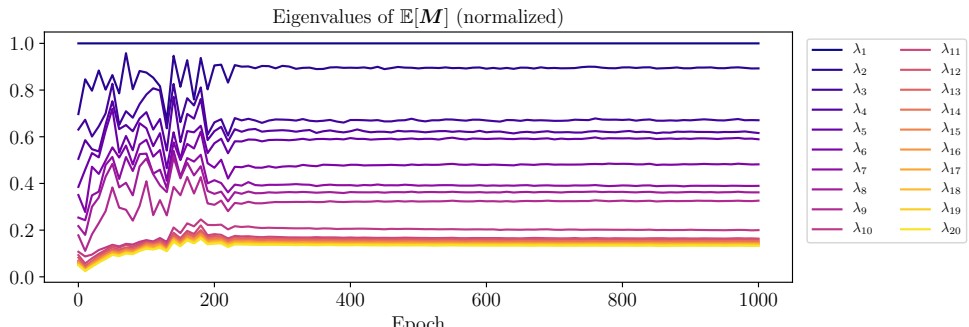

Figure 19: Top eigenvalues of $\mathbb{E}[\boldsymbol{M}]$ along training trajectory. (fc1:LeNet5)

# F Additional Explanations

## F.1 Outliers in Hessian Eigenspectrum

One characteristic of Hessian that has been mentioned by many is the outliers in the spectrum of eigenvalues. Sagun et al. (2018) suggests that there is a gap in Hessian eigenvalue distribution around the number of classes $c$ in most cases, where $c = 10$ in our case. A popular theory to explain the gap is the class / logit clustering of the logit gradients (Fort & Ganguli, 2019; Papyan, 2019, 2020). Note that these explanations can be consistent with our heuristic formula for the top eigenspace of output Hessian at initialization– in the two-layer setting we considered the logit gradients are indeed clustered.

In the layer-wise setting, the clustering claim can be formalized as follows: For each class $k \in [c]$ and logit entry $l \in [c]$, with $Q$ be defined as in Eq. (28), and $(\mathbf{x}, \mathbf{y})$ as the input, label pair, let

$$\Delta_{i,j} = \mathbb{E}\left[\left. Q_{\mathbf{x}} \frac{\partial z_{\mathbf{x}}}{\partial w_j^{(p)}} \right| \mathbf{y} = i \right]. \tag{53}$$

Then at the initialization, for each logit entry $j$, $\{\Delta_{i,j}\}_{i\in[c]}$ is clustered around the "logit center" $\hat{\Delta}_j \triangleq \mathbb{E}_{i\in[c]}[\Delta_{i,j}]$; at the minima, for each class $i$, $\{\Delta_{i,j}\}_{j\in[c]}$ is clustered around the "class center" $\check{\Delta}_i \triangleq \mathbb{E}_{j\in[c]}[\Delta_{i,j}]$. With the decoupling conjectures, we may also consider similar claims for output Hessians, where

$$\Gamma_{i,j} = \mathbb{E}\left[\left. Q_{\mathbf{x}} \frac{\partial z_{\mathbf{x}}}{\partial z_{\mathbf{x}}^{(p)}}_j \right| \mathbf{y} = i \right]. \tag{54}$$

A natural extension of the clustering phenomenon on output Hessians is then as follows: At the initialization, for each logit entry $j$, $\{\Gamma_{i,j}\}_{i\in[c]}$ is clustered around $\check{\Gamma}_j \triangleq \mathbb{E}_{i\in[c]}[\Gamma_{i,j}]$; at the minima, for each class $i$, $\{\Gamma_{i,j}\}_{j\in[c]}$ is clustered around $\hat{\Gamma}_i \triangleq \mathbb{E}_{j\in[c]}[\Gamma_{i,j}]$. Note that we have the layer-wise Hessian and layer-wise output Hessian satisfying

$$H_{\mathcal{L}}(w^{(p)}) = \underset{i,j\in[c]}{\mathbb{E}}[\Delta_{i,j}^T \Delta_{i,j}], \quad M^{(p)} = \underset{i,j\in[c]}{\mathbb{E}}[\Gamma_{i,j}^T \Gamma_{i,j}]. \tag{55}$$

**Low-rank Hessian at Random Initialization and Logit Gradient Clustering**
We first briefly recapture our explanation on the low-rankness of Hessian at random initialization. In Appendix H, we have shown that for a two layer ReLU network with Gaussian random initialization and Gaussian random input, the output hessian of the first layer $M^{(1)}$ is approximately $\frac{1}{4}W^{(2)T}AW^{(2)}$. We then heuristically extend this approximation to a randomly initialized $L$-layer network, that with $S^{(p)} = W^{(L)}W^{(L-1)}\cdots W^{(p+1)}$, the output Hessian of the $p$-th layer $H^{(p)}$ can be approximated by $\tilde{M}^{(p)}$ where

$$\tilde{M}^{(p)} \triangleq \frac{1}{4^{L-p}} S^{(p)T} A S^{(P)}. \tag{56}$$

Since $A$ is strictly rank $c - 1$ with null space of the all-one vector, $H^{(p)}$ is strictly rank $c - 1$. Thus $H^{(p)}$ is approximately rank $c - 1$, and so is the corresponding layerwise Hessian according to the decoupling conjecture.

Now we discuss the connection between our analysis with the theory of logit gradient clustering. As previously observed by Papyan (2019), for each logit entry $l$, $\{\Delta_{i,j}\}_{l\in[c]}$ are clustered around the logit gradients $\mathbb{E}_{l\in[c]}[\Delta_{i,j}]$. Similar clustering effects for $\{\Gamma_{i,j}\}_{l\in[c]}$ were also empirically observed by our experiments. Moreover, through the approximation above and the decoupling conjecture, for each logit entry $j$, the cluster centers $\check{\Gamma}_j$ and $\hat{\Delta}_j$ can be approximated by

$$\begin{aligned} \check{\Gamma}_j &\approx \check{\Gamma}_j \triangleq (S^T Q)_j \\ \check{\Delta}_j &\approx \check{\Delta}_j \triangleq ((\mathbb{E}[x] \otimes S^T)\mathbb{E}[Q])_j. \end{aligned} \tag{57}$$

Following Papyan (2019), we used t-SNE (Van der Maaten & Hinton, 2008) to visualize the logit gradients. As we see in Fig. 20, the "logit centers" of the clustering directly corresponds to the approximated dominating eigenvectors of the Hessian, which is consistent with our analysis.

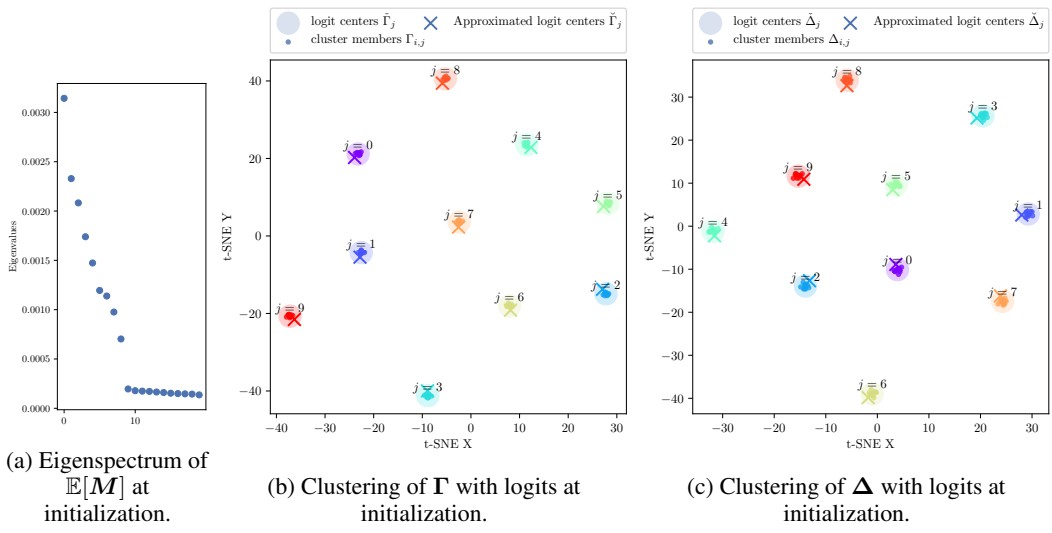

(a) Eigenspectrum of $\mathbb{E}[\boldsymbol{M}]$ at initialization.

(b) Clustering of $\boldsymbol{\Gamma}$ with logits at initialization.

(c) Clustering of $\boldsymbol{\Delta}$ with logits at initialization.

Figure 20: Logit clustering behavior of $\boldsymbol{\Delta}$ and $\boldsymbol{\Gamma}$ at initialization (fc1:T-200$^2$)

**Gradient Clustering at Minima**    Currently our theory does not provide an explanation to the low rank structure of Hessian at the minima. However we have observed that the class clustering of logit gradients does not universally apply to all models at the minima, even when the models have around $c$ significant large eigenvalues. As shown in Fig. 21, the class clustering is very weak but there are still around $c$ significant large eigenvalues. We conjecture that the class clustering of logit gradients may be a sufficient but not necessary condition for the Hessian to be low rank at minima.

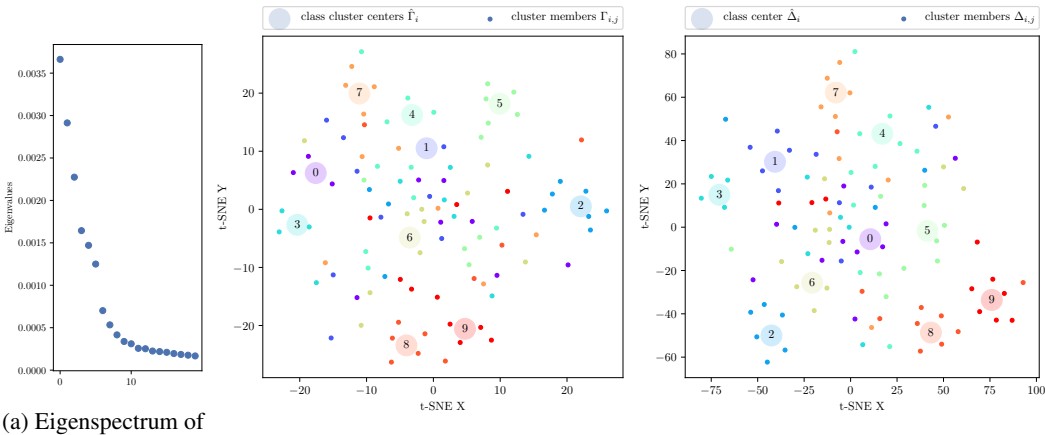

(a) Eigenspectrum of $\mathbb{E}[\boldsymbol{M}]$ at minimum.    (b) Clustering of $\boldsymbol{\Gamma}$ with class at minimum.    (c) Clustering of $\boldsymbol{\Delta}$ with class at minimum.

Figure 21: Class clustering behavior of $\boldsymbol{\Delta}$ and $\boldsymbol{\Gamma}$ at minimum. (fc1:T-200$^2$)

## F.2    Dominating Eigenvectors of Layer-wise Hessian are Low Rank

A natural corollary for the Kronecker factorization approximation of layer-wise Hessians is that the eigenvectors of the layer-wise Hessians are low rank. Let $\boldsymbol{h}_i$ be the $i$-th eigenvector of a layer-wise Hessian. The rank of $\mathrm{Mat}(\boldsymbol{h}_i)$ can be considered as an indicator of the complexity of the eigenvector. Consider the case that $\boldsymbol{h}_i$ is one of the top eigenvectors. From Section 5.1, we have $\boldsymbol{h}_i \approx \boldsymbol{u}_i \otimes \hat{\mathbb{E}}[\boldsymbol{x}]$. Thus, $\mathrm{Mat}(\boldsymbol{h}_i) \approx \boldsymbol{u}_i\hat{\mathbb{E}}[\boldsymbol{x}]^T$, which is approximately rank 1. Experiments shows that first singular values of $\mathrm{Mat}(\boldsymbol{h}_i)$ divided by its Frobenius Norm are usually much larger than 0.5, indicating the top eigenvectors of the layer-wise Hessians are very close to rank 1. Fig. 22 shows first singular values of $\mathrm{Mat}(\boldsymbol{h}_i)$ divided by its Frobenius Norm for $i$ from 1 to 200. We can see that the top eigenvectors of the layer-wise Hessians are very close to rank 1.

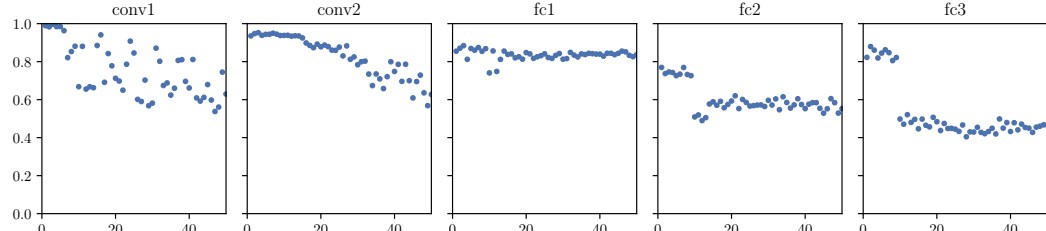

Figure 22: Ratio between top singular value and Frobenius norm of matricized dominating eigenvectors. (LeNet5 on CIFAR10). The horizontal axes denote the index $i$ of eigenvector $\boldsymbol{h}_i$, and the vertical axes denote $\| \operatorname{Mat}(\boldsymbol{h}_i)\| / \| \operatorname{Mat}(\boldsymbol{h}_i)\|_F$.

### F.3 Eigenspace Overlap of Different Models

From the experiment results in Appendix E together with Fig. 5, we can see that our approximation and explanation stated in Section 5.1 of the main text is approximately correct but may not be so accurate for some layers. We now present a more general explanation which addresses why the overlap before rank-$m$ grows linearly. We will also explain some exceptional cases as shown in Appendix E.2 and possible discrepancies of our approximation.

Let $\boldsymbol{h}_i$ be the $i$-th eigenvector of the layer-wise Hessian $\boldsymbol{H}_{\mathcal{L}}(\boldsymbol{w}^{(p)})$, under the assumption that the autocorrelation matrix $\mathbb{E}[\boldsymbol{x}\boldsymbol{x}^T]$ is approximately rank 1 that $\mathbb{E}[\boldsymbol{x}\boldsymbol{x}^T] \approx \mathbb{E}[\boldsymbol{x}]\mathbb{E}[\boldsymbol{x}]^T$, for all $i \leq m$, we can approximate the $\boldsymbol{h}_i$ as $\boldsymbol{u}_i \otimes (\mathbb{E}[\boldsymbol{x}]/\|\mathbb{E}[\boldsymbol{x}]\|)$ where $\boldsymbol{u}_i$ is the $i$-th eigenvector of $\mathbb{E}[\boldsymbol{M}]$. Formally, the trend of top eigenspace can be characterized by the following theorem. For simplicity of notations, we abuse the superscript within parentheses to refer the two models instead of layer number in this section.

**Theorem F.1.** Consider 2 different models with the same network structure trained on the same dataset. Fix the $p$-th hidden layer with input dimension $n$ and output dimension $m$. For the first model, denote its output Hessian as $\mathbb{E}[\boldsymbol{M}]^{(1)}$ with eigenvalues $\tau_1^{(1)} \geq \tau_2^{(1)} \geq \cdots \geq \tau_m^{(1)} \geq 0$ and eigenvectors $\boldsymbol{r}_1^{(1)}, \cdots, \boldsymbol{r}_m^{(1)} \in \mathbb{R}^m$; denote its autocorrelation matrix as $\mathbb{E}[\boldsymbol{x}\boldsymbol{x}^T]^{(1)}$, with eigenvalues $\gamma_1^{(1)} \geq \gamma_2^{(1)} \geq \cdots \geq \gamma_m^{(1)} \geq 0$ and eigenvectors $\boldsymbol{t}_1^{(1)}, \cdots, \boldsymbol{t}_n^{(1)} \in \mathbb{R}^n$. The variables for the second matrices are defined identically by changing 1 in the superscript parenthesis to 2.

Assume the Kronecker factorization approximation is accurate that $\boldsymbol{H}_{\mathcal{L}}(\boldsymbol{w}^{(p)})^{(1)} \approx \mathbb{E}[\boldsymbol{M}]^{(1)} \otimes \mathbb{E}[\boldsymbol{x}\boldsymbol{x}^T]^{(1)}$ and $\boldsymbol{H}_{\mathcal{L}}(\boldsymbol{w}^{(p)})^{(2)} \approx \mathbb{E}[\boldsymbol{M}]^{(2)} \otimes \mathbb{E}[\boldsymbol{x}\boldsymbol{x}^T]^{(2)}$. Also assume the autocorrelation matrices of two models are sufficiently close to rank 1 in the sense that $\tau_m^{(1)}\gamma_1^{(1)} > \tau_1^{(1)}\gamma_2^{(1)}$ and $\tau_m^{(2)}\gamma_1^{(2)} > \tau_1^{(2)}\gamma_2^{(2)}$. Then for all $k \leq m$, the overlap of top $k$ eigenspace between their layerwise Hessians $\boldsymbol{H}_{\mathcal{L}}(\boldsymbol{w}^{(p)})^{(1)}$ and $\boldsymbol{H}_{\mathcal{L}}(\boldsymbol{w}^{(p)})^{(2)}$ will be approximately $\frac{k}{m}(\boldsymbol{t}_1^{(1)} \cdot \boldsymbol{t}_1^{(2)})^2$. Consequently, the top eigenspace overlap will show a linear growth before it reaches dimension $m$. The peak at $m$ is approximately $(\boldsymbol{t}_1 \cdot \boldsymbol{t}_2)^2$.

*Proof.* Let $\boldsymbol{h}_i^{(2)}$ be the $i$-th eigenvector of the layer-wise Hessian for the first model $\boldsymbol{H}_{\mathcal{L}}(\boldsymbol{w}^{(p)})^{(1)}$, and $\boldsymbol{g}_i$ be that of the second model $\boldsymbol{H}_{\mathcal{L}}(\boldsymbol{w}^{(p)})^{(2)}$. Consider the first model. By the Kronecker factorization approximation, since $\tau_m^{(1)}\gamma_1^{(1)} > \tau_1^{(1)}\gamma_2^{(1)}$, the top $m$ eigenvalues of the layer-wise Hessian are $\gamma_1^{(1)}\tau_1^{(1)}, \cdots, \gamma_1^{(1)}\tau_m^{(1)}$. Consequently, for all $i \leq m$ we have $\boldsymbol{h}_i \approx \boldsymbol{r}_i^{(1)T} \otimes \boldsymbol{t}_1^{(1)}$. Thus, for any $k \leq m$, we have its top $k$ eigenspace as $\boldsymbol{V}_k^{(1)} \otimes \boldsymbol{t}_1^{(1)}$, where $\boldsymbol{V}_k^{(1)} \in \mathbb{R}^{m \times k}$ has column vectors $\boldsymbol{r}_1^{(1)}, \ldots, \boldsymbol{r}_k^{(1)}$. Similarly, for the second model we have $\boldsymbol{h}_i^{(2)} \approx \boldsymbol{r}_i^{(2)} \otimes \boldsymbol{t}_1^{(2)}$ and the top $k$ eigenspace as $\boldsymbol{V}_k^{(2)} \otimes \boldsymbol{t}_1^{(2)}$, where $\boldsymbol{V}_k^{(2)}$ has column vectors $\boldsymbol{r}_1^{(2)}, \ldots, \boldsymbol{r}_k^{(2)}$. The eigenspace overlap of the 2 models at dimension $k$ is thus

$$
\begin{aligned}
\operatorname{Overlap}\left(\boldsymbol{V}_k^{(1)} \otimes \boldsymbol{t}_1^{(1)}, \boldsymbol{V}_k^{(2)} \otimes \boldsymbol{t}_1^{(2)}\right) &= \frac{1}{k}\left\|\boldsymbol{V}_k^{(1)T}\boldsymbol{V}_k^{(2)} \otimes \boldsymbol{t}_1^{(1)T}\boldsymbol{t}_1^{(2)}\right\|_F^2 \\
&= \left(\boldsymbol{t}_1^{(1)} \cdot \boldsymbol{t}_1^{(2)}\right)^2 \operatorname{Overlap}\left(\boldsymbol{V}_k^{(1)}, \boldsymbol{V}_k^{(2)}\right).
\end{aligned}
\tag{58}
$$

Note that for all $i \leq m$, $\boldsymbol{r}_i^{(1)}, \boldsymbol{r}_i^{(2)} \in \mathbb{R}^n$, which is the space corresponding to the neurons. Since for hidden layers, the output neurons (channels for convolutional layers) can be arbitrarily permuted to give equivalent models while changing eigenvectors. For $\boldsymbol{h}_i \approx \boldsymbol{r}_i \otimes \boldsymbol{t}_1$, permuting neurons will permute entries in $\boldsymbol{r}_i$. Thus, we can assume that for two models, $\boldsymbol{r}_i^{(1)}$ and $\boldsymbol{r}_i^{(2)}$ are not correlated and thus have an expected inner product of $\sqrt{1/m}$.

It follows from Definition 4.1 that $\mathbb{E}[\text{Overlap}(\boldsymbol{V}_k^{(1)}, \boldsymbol{V}_k^{(2)})] = \sum_{i=1}^{k} \mathbb{E}[(\boldsymbol{r}_i^{(1)} \cdot \boldsymbol{r}_i^{(2)})^2] = k(\frac{1}{m}) = \frac{k}{m}$ and thus the eigenspace overlap of at dimension $k$ would be approximately $\frac{k}{m}(\boldsymbol{t}_1^{(1)} \cdot \boldsymbol{t}_1^{(2)})^2$. This explains the peak at dimension $m$ and the linear growth before it. $\qquad\square$

From our results on autocorrelation matrices in Section 4.3 and Appendix E.1, we have $\hat{\mathbb{E}}[\boldsymbol{x}]^{(1)} \approx \boldsymbol{t}_1^{(1)}$ and $\hat{\mathbb{E}}[\boldsymbol{x}]^{(2)} \approx \boldsymbol{t}_1^{(2)}$ where $\hat{\mathbb{E}}$ is the normalized expectation. Hence when $k = m$, the overlap is approximately $(\hat{\mathbb{E}}[\boldsymbol{x}]^{(1)} \cdot \hat{\mathbb{E}}[\boldsymbol{x}]^{(2)})^2$. Since $\hat{\mathbb{E}}[\boldsymbol{x}]^{(1)}$ and $\hat{\mathbb{E}}[\boldsymbol{x}]^{(2)}$ are the identical for the input layers, the overlap is expected to be very high at dimension $m$ for input layers. For other hidden layers in a ReLU network, $\boldsymbol{x}$ are output of ReLU and thus non-negative. Two non-negative vectors $\hat{\mathbb{E}}[\boldsymbol{x}]^{(1)}$ and $\hat{\mathbb{E}}[\boldsymbol{x}]^{(2)}$ still have relatively large dot product, which contributes to the high overlap peak.

### F.3.1 The Decreasing Overlap After Output Dimension

Consider the $(m+1)$-th eigenvector $\boldsymbol{h}_{m+1}^{(1)}$ of the first model. Following the Kronecker factorization approximation and assumptions in Theorem F.1, we have $\boldsymbol{h}_{m+1}^{(1)} \approx \boldsymbol{r}_1^{(1)} \otimes \boldsymbol{t}_2^{(1)}$. Since top $m$ eigenspace of the first model is approximately $\boldsymbol{I}_m \otimes \boldsymbol{t}_1^{(1)}$ and $\boldsymbol{t}_2^{(1)}$ is orthogonal to $\boldsymbol{t}_1^{(1)}$, the $\boldsymbol{h}_{m+1}^{(1)}$ eigenvector will be orthogonal to the top $m$ eigenspace of the first model. It will also have low overlap with $\boldsymbol{I}_m \otimes \boldsymbol{t}_1^{(2)}$ since $(\hat{\mathbb{E}}[\boldsymbol{x}]^{(1)} \cdot \hat{\mathbb{E}}[\boldsymbol{x}]^{(2)})^2$ is large.

Moreover, since the remaining eigenvectors of the autocorrelation matrix no longer has the all positive property as the first eigenvector and structure of the convariance $\Sigma_{\boldsymbol{x}}$ is directly associated with the ordering of the input neurons which are randomly permuted across different models, the overlap between other eigenvectors of the autocorrelation matrix across different models will be close to random, hence the overlap after the top $m$ dimension will decrease until the eigenspaces has sufficiently many basis vectors to make the random overlap large.

### F.3.2 The Output Layer

Note that for the last layer satisfying the assumptions in Theorem F.1, the overlap will stay high before dimension $m$ and be approximately $(\boldsymbol{t}_1 \cdot \boldsymbol{t}_2)^2$ since the output neurons directly correspondence to classes, and hence neurons cannot be permuted. In this case, the overlap will be approximately $(\boldsymbol{t}_1 \cdot \boldsymbol{t}_2)^2$ for all dimension $k \leq m$. This is consistent with our observations.

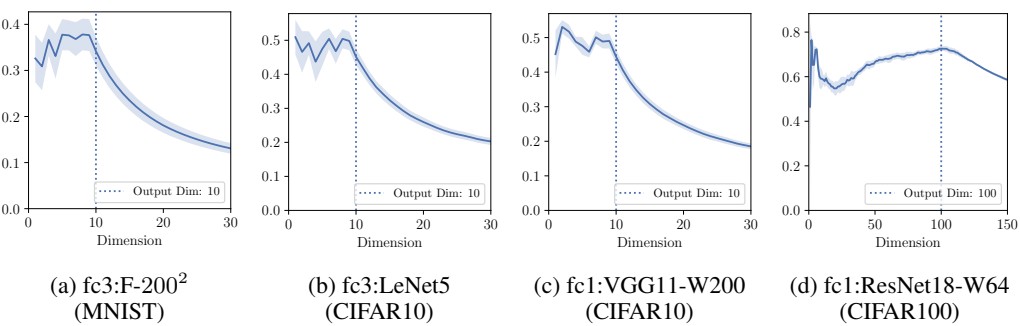

(a) fc3:F-$200^2$ (MNIST)  (b) fc3:LeNet5 (CIFAR10)  (c) fc1:VGG11-W200 (CIFAR10)  (d) fc1:ResNet18-W64 (CIFAR100)

Figure 23: Top eigenspace overlap for the final fully connected layer.

### F.3.3 Explaining "Failure Cases" of Eigenspace Overlap

As shown in Fig. 12 and Fig. 13, the nontrivial top eigenspace overlap does not necessarily peak at the output dimension for all layers. Some layers has a low peak at very small dimensions and others has a peak at a larger dimension. With the more complete analysis provided above, we now proceed to explain these two phenomenons. The major reason for such phenomenons is that the assumption of autocorrelation matrix being sufficiently close to rank 1 is not always satisfied. In particular, following the notations in Theorem F.1, for these exceptional layers we have $\tau_m \gamma_1 < \tau_1 \gamma_2$.

We first consider the first phenomenon (early peak of low overlap) and take fc2:F-$200^2$ (MNIST) in as an example. Here Fig. 24a is identical to Fig. 12a, which displays the early peak around $m = 10$.

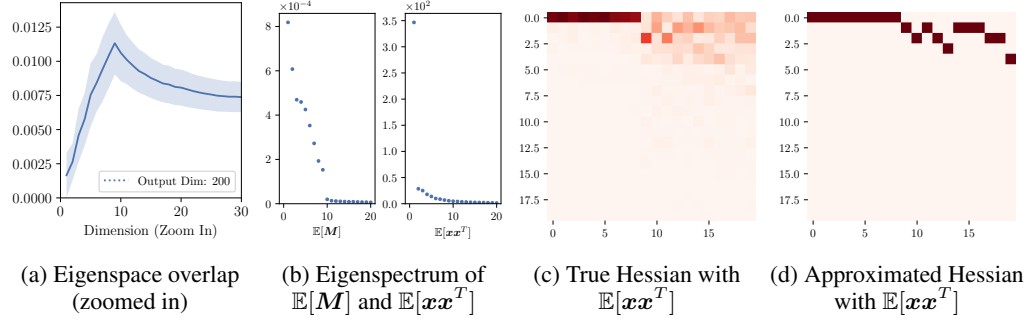

| (a) Eigenspace overlap (zoomed in) | (b) Eigenspectrum of $\mathbb{E}[\boldsymbol{M}]$ and $\mathbb{E}[\boldsymbol{x}\boldsymbol{x}^T]$ | (c) True Hessian with $\mathbb{E}[\boldsymbol{x}\boldsymbol{x}^T]$ | (d) Approximated Hessian with $\mathbb{E}[\boldsymbol{x}\boldsymbol{x}^T]$ |

Figure 24: Eigenspace overlap, eigenspectrum, and cropped (upper $20 \times 20$ block) eigenvector correspondence matrices for fc2:F-$200^2$ (MNIST)

As shown in Fig. 24b, the second eigenvalue of the auto correlation $\mathbb{E}[\boldsymbol{x}\boldsymbol{x}^T]$ is as large as approximately 1/10 of the first eigenvalue. With the output Hessian have $c - 1 = 9$ significant large eigenvalues as described in , it has $\tau_{10}\gamma_1 < \tau_1 \gamma_2$. Thus through the Kronecker factorization approximation, the top $m$ dimensional eigenspace is no longer simply $\boldsymbol{I}_m \otimes \hat{\mathbb{E}}[\boldsymbol{x}]$, but a subset of top eigenvectors of the output Hessian Kroneckered with a subset of top eigenvectors of $\mathbb{E}[\boldsymbol{x}\boldsymbol{x}^T]$ as reflected in Fig. 24d. This "mixture" of Kronecker product is moreover verified in Fig. 24c.

As reflected by the first row of Fig. 24c and Fig. 24d, for $i \leq 9$ we have $\boldsymbol{h}_i \approx \boldsymbol{r}_i \otimes \hat{\mathbb{E}}[\boldsymbol{x}]$, which falls in the regime of Theorem F.1. Hence we are seeing an linearly growing pattern of the overlap for dimension less than 10 and reaches a mean overlap of around 0.012 by dimension 9. If following this linear trend, the overlap would be close to 0.25 by the output dimension of 200. However, since the 10-th eigenvalue of the output Hessian is significantly smaller, little of the 10-19 dimensional eigenspace were contributed by $\hat{\mathbb{E}}[\boldsymbol{x}]$, hence the overlap of dimension larger than 10 falls into the regime discussed in Appendix F.3.1, for which we see a sharp decrease of overlap after dimension 9. Note that this example shows that Kronecker factorization can be used to predict when our conditions in Theorem F.1 fails and also predict the condition can be satisfied up to which dimension. As shown in Fig. 25, similar explanation also applies to convolutional layers in larger networks.

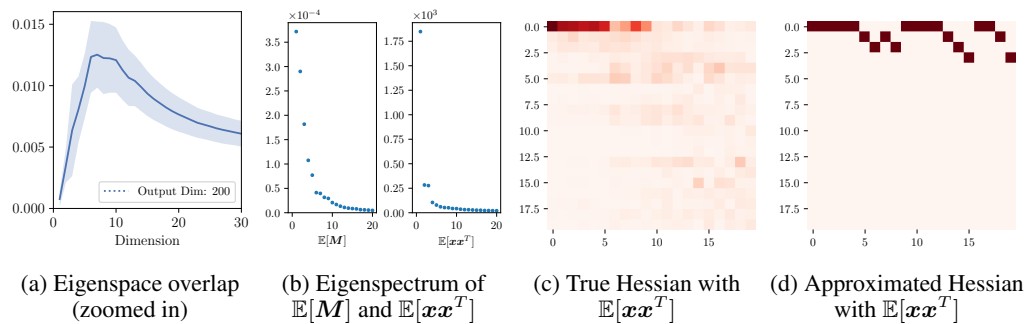

| (a) Eigenspace overlap (zoomed in) | (b) Eigenspectrum of $\mathbb{E}[\boldsymbol{M}]$ and $\mathbb{E}[\boldsymbol{x}\boldsymbol{x}^T]$ | (c) True Hessian with $\mathbb{E}[\boldsymbol{x}\boldsymbol{x}^T]$ | (d) Approximated Hessian with $\mathbb{E}[\boldsymbol{x}\boldsymbol{x}^T]$ |

Figure 25: Eigenspace overlap, eigenspectrum, and cropped (upper $20 \times 20$ block) eigenvector correspondence matrices for conv5:VGG11-W200 (CIFAR10)

We then consider the second phenomenon (delayed peak) and take conv2:VGG11-W200 (CIFAR10) in as an example. Here Fig. 26a is identical to Fig. 13a, which has the overlap peak later than the output dimension 200. In this case, the second eigenvalue of the auto correlation matrix is still not negligible compared to the top eigenvalue. What differentiate this case from the first phenomenon is that the eigenvalues of the output Hessian no longer has a significant peak – instead it has a heavy tail which is necessary for high overlap.

Towards dimension $m$ there gradually exhibits higher correspondence to later eigenvectors of the input autocorrelation matrix and hence less correspondence to $\hat{\mathbb{E}}[\boldsymbol{x}]$. This eventually results in the delayed and flattened peak.

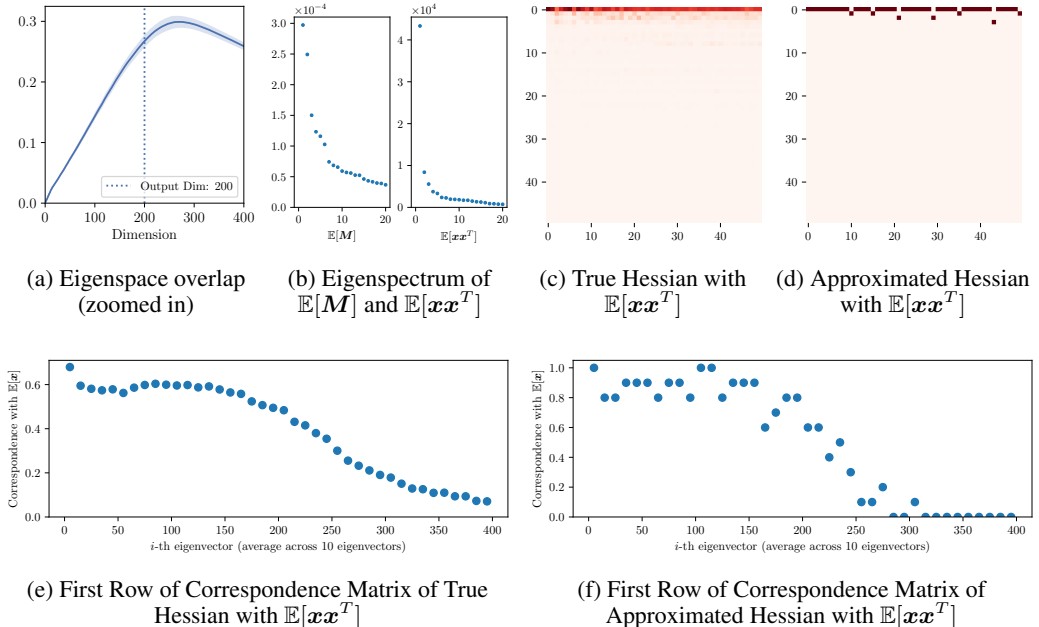

(a) Eigenspace overlap (zoomed in)

(b) Eigenspectrum of $\mathbb{E}[\boldsymbol{M}]$ and $\mathbb{E}[\boldsymbol{x}\boldsymbol{x}^T]$

(c) True Hessian with $\mathbb{E}[\boldsymbol{x}\boldsymbol{x}^T]$

(d) Approximated Hessian with $\mathbb{E}[\boldsymbol{x}\boldsymbol{x}^T]$

(e) First Row of Correspondence Matrix of True Hessian with $\mathbb{E}[\boldsymbol{x}\boldsymbol{x}^T]$

(f) First Row of Correspondence Matrix of Approximated Hessian with $\mathbb{E}[\boldsymbol{x}\boldsymbol{x}^T]$

Figure 26: Eigenspace overlap, eigenspectrum, and cropped (upper $50 \times 50$ block) eigenvector correspondence matrices for conv2:VGG11-W200 (CIFAR10)

Since the full correspondence matrices are too large to be visualized, we plotted their first rows up to 400 dimensions in Fig. 26e and Fig. 26f, in which each dot represents the average of correlation with $\hat{\mathbb{E}}[\boldsymbol{x}]$ for the 10 eigenvector nearby. From these figures it is straightforward to see the gradual decreasing correlation with $\hat{\mathbb{E}}[\boldsymbol{x}]$.

### F.4 Batch Normalization and Zero-mean Input

In this section, we show the results on networks with using Batch normalization (BN) (Ioffe & Szegedy, 2015). For layers after BN, we have $\mathbb{E}[\boldsymbol{x}] \approx 0$ so that $\mathbb{E}[\boldsymbol{x}]\mathbb{E}[\boldsymbol{x}]^T$ no longer dominates $\boldsymbol{\Sigma}_{\boldsymbol{x}}$ and the low rank structure of $\mathbb{E}[\boldsymbol{x}\boldsymbol{x}^T]$ should disappear. Thus, we can further expect that the overlap between top eigenspace of layer-wise Hessian among different models will not have a peak.

Table 8 shows the same experiments done in Table 6. The values for each network are the average of 3 different models. It is clear that the high inner product and large spectral ratio both do not hold here, except for the first layer where there is no normalization applied. Note that we had channel-wise normalization (zero-mean for each channel but not zero-mean for $\boldsymbol{x}$) for conv1 in LeNet5 so that the spectral ratio is also small.

Fig. 27a shows that $\mathbb{E}[\boldsymbol{x}\boldsymbol{x}^T]$ is no longer close to rank 1 when having BN. This is as expected. However, $\mathbb{E}[\boldsymbol{x}\boldsymbol{x}^T]$ still has a few large eigenvalues.

Fig. 27b shows the eigenvector correspondance matrix of True Hessian with $\mathbb{E}[\boldsymbol{x}\boldsymbol{x}^T]$ for fc1:LeNet5. Because $\mathbb{E}[\boldsymbol{x}\boldsymbol{x}^T]$ is no longer close to rank 1, only very few eigenvectors of the layer-wise Hessian will have high correspondance with the top eigenvector of $\mathbb{E}[\boldsymbol{x}\boldsymbol{x}^T]$, as expected. This directly leads

Table 8: Structure of $\mathbb{E}[\boldsymbol{x}\boldsymbol{x}^T]$ for BN networks

| Dataset | Network | # fc | $(v_1^T\hat{\mathbb{E}}[\boldsymbol{x}])^2$ | | | $\lambda_1/\lambda_2$ | | |
| | | | mean | min | max | mean | min | max |
|---|---|---|---|---|---|---|---|---|
| MNIST | F-200$^2$-BN | 2 | 0.062 | 0.001 | 0.260 | 1.16 | 1.04 | 1.30 |
| | F-600$^2$-BN | 2 | 0.026 | 0.000 | 0.063 | 1.13 | 1.02 | 1.26 |
| | F-600$^4$-BN | 4 | 0.027 | 0.000 | 0.146 | 1.11 | 1.03 | 1.19 |
| CIFAR10 | LeNet5-BN | 3 | 0.210 | 0.001 | 0.803 | 1.54 | 1.20 | 1.89 |

to the disappearance of peak in top eigenspace overlap of different models, as shown in Fig. 28. The peak still exists in conv1 because BN is not applied to the input.

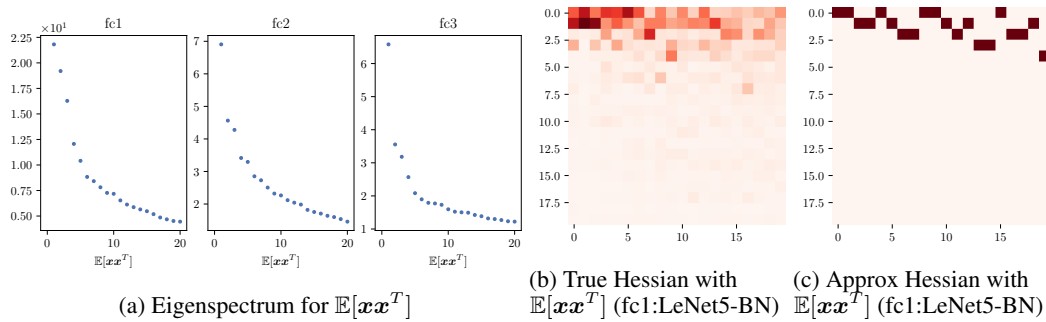

(a) Eigenspectrum for $\mathbb{E}[\boldsymbol{x}\boldsymbol{x}^T]$     (b) True Hessian with $\mathbb{E}[\boldsymbol{x}\boldsymbol{x}^T]$ (fc1:LeNet5-BN)     (c) Approx Hessian with $\mathbb{E}[\boldsymbol{x}\boldsymbol{x}^T]$ (fc1:LeNet5-BN)

Figure 27: Eigenspectrum and Eigenvector correspondence matrices with $\mathbb{E}[\boldsymbol{x}\boldsymbol{x}^T]$ for LeNet5-BN.

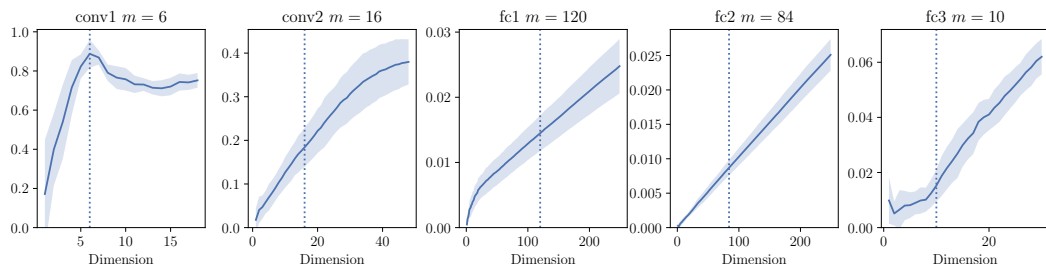

Figure 28: Eigenspace overlap of different models of LeNet5-BN.

Comparing Fig. 27b and Fig. 27c, we can see that the Kronecker factorization still gives a reasonable approximation for the eigenvector correspondence matrix with $\mathbb{E}[\boldsymbol{x}\boldsymbol{x}^T]$, although worse than the cases without BN (Fig. 3).

Fig. 29 compare the eigenvalues and top eigenspaces of the approximated Hessian and the true Hessian for LeNet5 with BN. The approximation using Kronecker factorization is also worse than the case without BN (Fig. 2). However, the approximation still gives meaningful information as the overlap of top eigenspace is still highly nontrivial.

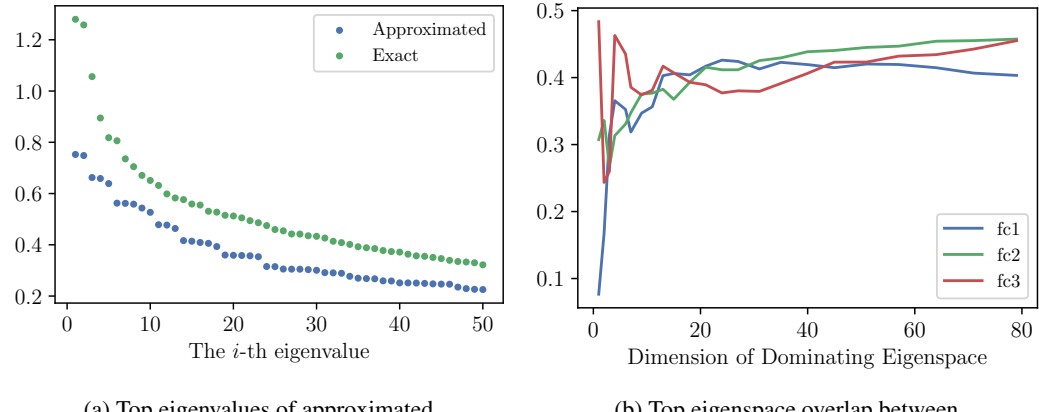

(a) Top eigenvalues of approximated and exact layer-wise Hessian for fc2.

(b) Top eigenspace overlap between approximated and true layer-wise Hessian.

Figure 29: Comparison between the true and approximated layer-wise Hessians for LeNet5-BN.

# G   Computing PAC-Bayes Bounds with Hessian Approximation

Given a model parameterized with $\theta$ and an input-label pair $(\boldsymbol{x}, \boldsymbol{y}) \in \mathbb{R}^d \times \mathbb{R}^c$, the classification error of $\theta$ over the input sample $\boldsymbol{x}$ is $\breve{l}(\theta, \boldsymbol{x}) := \mathbf{1}[\arg\max f_\theta(\boldsymbol{x}) = \arg\max \boldsymbol{y}]$. With the underlying data distribution $D$ and training set $S$ i.i.d. sampled from $D$, we define

$$e(\theta) := \mathbb{E}_{(\boldsymbol{x}, \boldsymbol{y}) \sim D}[\breve{l}(\theta, \boldsymbol{x})], \qquad \hat{e}(\theta) := \frac{1}{N} \sum_{i=1}^{N} [\breve{l}(\theta, \boldsymbol{x}_i)] \tag{59}$$

as the expected and empirical classification error of $\theta$, respectively. We define the measurable hypothesis space of parameters $\mathcal{H} := \mathbb{R}^P$. For any probabilistic measure $P$ in $\mathcal{H}$, let $e(P) = \mathbb{E}_{\theta \sim P} e(\theta)$, $\hat{e}(P) = \mathbb{E}_{\theta \sim P} \hat{e}(\theta)$, and $\breve{e}(P) = \mathbb{E}_{\theta \sim P} \mathcal{L}(\theta)$. Here $\breve{e}(P)$ serves as a differentiable convex surrogate of $\hat{e}(P)$.

**Theorem G.1** (Pac-Bayes Bound). (McAllester, 1999)(Langford & Seeger, 2001) For any prior distribution $P$ in $\mathcal{H}$ that is chosen independently from the training set $S$, and any posterior distribution $Q$ in $\mathcal{H}$ whose choice may inference $S$, with probability $1 - \delta$,

$$D_{\mathrm{KL}}\left(\hat{e}(Q) \| e(Q)\right) \leq \frac{D_{\mathrm{KL}}(Q \| P) + \log \frac{|S|}{\delta}}{|S| - 1}. \tag{60}$$

Fix some constant $b, c \geq 0$ and $\theta_0 \in \mathcal{H}$ as a random initialization, Dziugaite & Roy (2017) shows that when setting $Q = \mathcal{N}(\boldsymbol{w}, \mathrm{diag}(\boldsymbol{s}))$, $P = \mathcal{N}(\theta_0, \lambda \boldsymbol{I}_P)$, where $\boldsymbol{w}, \boldsymbol{s} \in \mathcal{H}$ and $\lambda = c \exp(-j/b)$ for some $j \in \mathbb{N}$, and solve the optimization problem

$$\min_{\boldsymbol{w}, \boldsymbol{s}, \lambda} \breve{e}(Q) + \sqrt{\frac{D_{\mathrm{KL}}(Q \| P) + \log \frac{|S|}{\delta}}{2(|S| - 1)}}, \tag{61}$$

with initialization $\boldsymbol{w} = \theta$, $\boldsymbol{s} = \theta^2$, one can achieved a nonvacuous PAC-Bayes bound by Eq. (60).

In order to avoid discrete optimization for $j \in \mathbb{N}$, Dziugaite & Roy (2017) uses the $B_{\mathrm{RE}}$ term to replace the bound in Eq. (60). The $B_{\mathrm{RE}}$ term is defined as

$$B_{\mathrm{RE}}(\boldsymbol{w}, \boldsymbol{s}, \lambda; \delta) = \frac{D_{\mathrm{KL}}(P \| Q) + 2 \log(b \log \frac{c}{\lambda}) + \log \frac{\pi^2 |S|}{6\delta}}{|S| - 1}, \tag{62}$$

where $Q = \mathcal{N}(\boldsymbol{w}, \mathrm{diag}(\boldsymbol{s}))$, $P = \mathcal{N}(\theta_0, \lambda \boldsymbol{I}_P)$. The optimization goal actually used in the implementation is thus

$$\min_{\boldsymbol{w} \in \mathbb{R}^P, \boldsymbol{s} \in \mathbb{R}_+^P, \lambda \in (0, c)} \breve{e}(Q) + \sqrt{\frac{1}{2} B_{\mathrm{RE}}(\boldsymbol{w}, \boldsymbol{s}, \lambda; \delta)}. \tag{63}$$

Algorithm 1 shows the algorithm for *Iterative Hessian* (ITER) PAC-Bayes Optimization. If we set $\eta = T$, the algorithm will be come *Approximate Hessian* (APPR) PAC-Bayes Optimization. It is based on Algorithm 1 in Dziugaite & Roy (2017). The initialization of $\boldsymbol{w}$ is different from Dziugaite & Roy (2017) because we believe what they wrote, $\mathrm{abs}(\boldsymbol{w})$ is a typo and $\log[\mathrm{abs}(\boldsymbol{w})]$ is what they actually means. It is more reasonable to initialize the variance $\boldsymbol{s}$ as $\boldsymbol{w}^2$ instead of $\exp[2\,\mathrm{abs}(\boldsymbol{w})]$.

---

**Algorithm 1** PAC-Bayes bound optimization using layer-wise Hessian eigenbasis

---

**Input:**
    $\boldsymbol{w}_0 \in \mathbb{R}^P$                                         ▷ Network parameters (Initialization)
    $\boldsymbol{w} \in \mathbb{R}^P$                                          ▷ Network parameters (SGD solution)
    $S$                                                     ▷ Training examples
    $\delta \in (0,1)$                                        ▷ Confidence parameter
    $b \in \mathbb{N}, c \in (0,1)$                             ▷ Precision and bound for $\lambda$
    $\tau \in (0,1), T \in \mathbb{N}$                        ▷ Learning rate; No. of iterations
    $\eta \in \mathbb{N}$                             ▷ Epoch interval for Hessian calculation
**Output**
    $\boldsymbol{w}$                                    ▷ Optimized network parameters
    $\boldsymbol{s}$                   ▷ Optimized posterior variances in Hessian eigenbasis
    $\lambda$                                    ▷ Optimized prior variancce

1:  **procedure** ITERATIVE-HESSIAN-PAC-BAYES
2:     $\varsigma \leftarrow \log[\mathrm{abs}(\boldsymbol{w})]$                           ▷ where $\boldsymbol{s}(\varsigma) = \exp(2\varsigma)$
3:     $\varrho \leftarrow -3$                               ▷ where $\lambda(\varrho) = \exp(2\varrho)$
4:     $R(\boldsymbol{w},\boldsymbol{s},\lambda) = \sqrt{\frac{1}{2}B_{\mathrm{RE}}(\boldsymbol{w},\boldsymbol{s},\lambda;\delta)}$               ▷ BRE term
5:     $B(\boldsymbol{w},\boldsymbol{s},\lambda,\boldsymbol{w}') = \mathcal{L}(\boldsymbol{w}') + R(\boldsymbol{w},\boldsymbol{s},\lambda)$       ▷ Optimization goal
6:     **for** $t = 0 \to T-1$ **do**                     ▷ Run SGD for T iterations
7:         **if** $t \bmod \eta == 0$ **then**
8:             HESSIANCALC$(w)$
9:         **end if**
10:        Sample $\boldsymbol{\xi} \sim \mathcal{N}(0,1)^P$
11:        $\boldsymbol{w}'(\boldsymbol{w},\varsigma) = \boldsymbol{w} + \mathrm{TOSTANDARD}\,(\boldsymbol{\xi} \odot \exp(\varsigma))$    ▷ Generate noisy parameter for SNN
12:        $\boldsymbol{w} \leftarrow \boldsymbol{w} - \tau\left[\nabla_{\boldsymbol{w}}R(\boldsymbol{w},\boldsymbol{s},\lambda) + \nabla_{\boldsymbol{w}'}\mathcal{L}(\boldsymbol{w}')\right]$
13:        $\varsigma \leftarrow \varsigma - \tau\left[\nabla_{\varsigma}R(\boldsymbol{w},\boldsymbol{s}(\varsigma),\lambda) + \mathrm{TOHESSIAN}\,(\nabla_{\boldsymbol{w}'}\mathcal{L}(\boldsymbol{w}')) \odot \boldsymbol{\xi} \odot \exp(\varsigma)\right]$
14:        $\varrho \leftarrow \varrho - \tau\nabla_{\varrho}R(\boldsymbol{w},\boldsymbol{s},\lambda(\varrho))$                ▷ Gradient descent
15:     **end for**
16:     **return** $w, s(\varsigma), \lambda(\varrho)$
17: **end procedure**

---

In the algorithm, HESSIANCALC$(\boldsymbol{w})$ is the process to calculate Hessian information with respect to the posterior mean $\boldsymbol{w}$ in order to produce the Hessian eigenbasis to perform the change of basis. For very small networks, we can calculate Hessian explicitly but it is prohibitive for most common networks. However, efficient approximate change of basis can be performed using our approximated layer-wise Hessians. In this case, we would just need to calculate the full eigenspace of $\mathbb{E}[\boldsymbol{M}]$ and that of $\mathbb{E}[\boldsymbol{x}\boldsymbol{x}^T]$ for each layer. For $p$th layer, we denote them as $\boldsymbol{U}^{(p)}$ and $\boldsymbol{V}^{(p)}$ respectively with eigenvectors as columns. We can also store the corresponding eigenvalues by doing pairwise multiplications between eigenvalues of $\mathbb{E}[\boldsymbol{M}]$ and $\mathbb{E}[\boldsymbol{x}\boldsymbol{x}^T]$.

After getting the eigenspaces, we can perform the change of basis. Note that we perform change of basis on vectors with the same dimensionality as the parameter vector (or the posterior mean). TOHESSIAN$(\boldsymbol{u})$ is the process to put a vector $\boldsymbol{u}$ in the standard basis to the Hessian eigenbasis. We first break $\boldsymbol{u}$ into different layers and let $\boldsymbol{u}^{(p)}$ be the vector for the $p$th layer. We then define $\mathrm{Mat}^{(p)}$ as the reshape of a vector to the shape of the parameter matrix $\boldsymbol{W}^{(p)}$ of that layer. We have the new vector $\boldsymbol{v}^{(p)}$ in Hessian basis as

$$\boldsymbol{v}^{(p)} = \mathrm{vec}\left[\boldsymbol{U}^{(p)T}\,\mathrm{Mat}^{(p)}(\boldsymbol{u}^{(p)})\boldsymbol{V}^{(p)}\right]. \tag{64}$$

The new vector $\boldsymbol{v} = \mathrm{TOHESSIAN}(\boldsymbol{u})$ is thus the concatenation of all the $\boldsymbol{v}^{(p)}$.

TOSTANDARD($\boldsymbol{v}$) is the process to put a vector $\boldsymbol{v}$ in the Hessian eigenbasis to the standard basis. It is the reverse process to TOHESSIAN. We also break $\boldsymbol{v}$ into layers and let the vector for the $p$th layer be $\boldsymbol{v}^{(p)}$. Then, the new vector $\boldsymbol{u}^{(p)}$ is

$$\boldsymbol{u}^{(p)} = \text{vec}\left[\boldsymbol{U}^{(p)}\,\text{Mat}^{(p)}(\boldsymbol{v}^{(p)})\boldsymbol{V}^{(p)T}\right], \tag{65}$$

The new vector $\boldsymbol{u} = $ TOSTANDARD($\boldsymbol{v}$) is thus the concatenation of all $\boldsymbol{u}^{(p)}$.

After getting optimized $\boldsymbol{w}, \boldsymbol{s}, \lambda$, we compute the final bound using Monte Carlo methods same as in Dziugaite & Roy (2017). Note that the prior $P$ is invariant with respect to the change of basis, since its covariance matrix is a multiple of identity $\lambda \boldsymbol{I}_P$. Thus, the KL divergence can be calculate in the Hessian eigenbasis without changing the value of $\lambda$. In the *Iterative Hessian with approximated output Hessian* (ITER.M), we use $\tilde{M}$ to approximate $\mathbb{E}[\boldsymbol{M}]$, as in Eq. (56).

We followed the experiment setting proposed by Dziugaite & Roy (2017) in general. In all the results we present, we first trained the models from Gaussian random initialization $w_0$ to the initial posterior mean estimate $w$ using SGD (lr=0.01) with batch-size 128 and epoch number 1000.

We then optimize the posterior mean and variance with layer-wise Hessian information using Algorithm 1, where $\delta = 0.025$, $b = 100$, and $c = 0.1$. We train for 2000 epochs, with learning rate $\tau$ initialized at 0.001 and decays with ratio 0.1 every 400 epochs. For *Approximated Hessian* algorithm, we set $\eta = 1$. For *Iterative Hessian* algorithm, we set $\eta = 10$. We also tried $\eta$ with the same decay schedule as learning rate (multiply $\eta$ by 10 every time the learning rate is multiplied by 0.1) and the results are similar to those without decay. We also used the same Monte Carlo method as in Dziugaite & Roy (2017) to calculate the final PAC-Bayes bound. Except that we used 50000 iterations instead of 150000 iterations because extra iterations do not further tighten the bound significantly. We use sample frequency 100 and $\delta' = 0.01$ as in that paper.

The complete experiment results are listed in Table 9. We follow the same naming convention as in Dziugaite & Roy (2017) except adding T-200$^2$ we introduced in Section 4. T-600$_{10}$, T-600$^2_{10}$, and T-200$^2_{10}$ are trained on standard MNIST with 10 classes, and others are trained on MNIST-2 (see Appendix D.1), in which we combined class 0-4 and class 5-9.

In Table 9, Prev means the previous results in Dziugaite & Roy (2017), APPR means *Approximated Hessian*, ITER means *Iterative Hessian*, ITER (D) means *Iterative Hessian* with decaying $\eta$, ITER.M means *Iterative Hessian with approximated output Hessian*. BASE are Base PAC-Bayes optimization as in the previous paper.

We also plotted the final posterior variance, $\boldsymbol{s}$. Fig. 30 shown below is for T-200$^2_{10}$. For posterior variance optimized with our algorithms (APPR, ITER, and ITER.M) we can see that direction associated with larger eigenvalue has a smaller variance. This agrees with our presumption that top eigenvectors are aligned with sharper directions and should have smaller variance after optimization. The effect is more significant and consistent for Iterative Hessian, where the PAC-Bayes bound is also tighter.

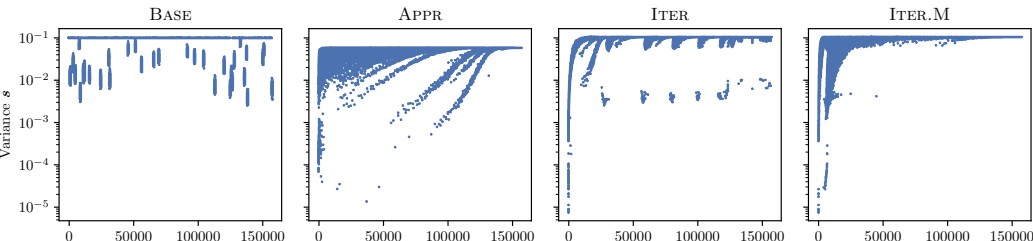

Figure 30: Optimized posterior variance, $\boldsymbol{s}$. (fc1:T-200$^2$, trained on MNIST), the horizontal axis is ordered with decreasing eigenvalues.

Table 9: Full PAC-Bayes bound optimization results

| Network | Method | PAC-Bayes Bound | KL Divergence | SNN loss | $\lambda$ (prior) | Test Error |
|---|---|---|---|---|---|---|
| T-600 | Prev | 0.161 | 5144 | 0.028 | - | 0.017 |
| | Base | 0.154 | 4612.6 | 0.03373 | -1.3313 | 0.0153 |
| | Appr | 0.1432 | 3980.6 | 0.03417 | -1.6063 | 0.0153 |
| | Iter | **0.1198** | 3766.1 | 0.02347 | -1.2913 | 0.0153 |
| | Iter(D) | 0.1199 | 3751.1 | 0.02366 | -1.2913 | 0.0153 |
| | Iter.M | 0.1255 | 3929.9 | 0.02494 | -1.3213 | 0.0153 |
| T-$600^2$ | Prev | 0.186 | 6534 | 0.028 | - | 0.016 |
| | Base | 0.1921 | 6966.6 | 0.03262 | -1.4163 | 0.0148 |
| | Appr | 0.1658 | 5176.1 | 0.03468 | -2.0963 | 0.0148 |
| | Iter | 0.1456 | 5086.5 | 0.02473 | -1.7963 | 0.0148 |
| | Iter(D) | **0.1443** | 4956.8 | 0.02523 | -1.7963 | 0.0148 |
| | Iter.M | 0.1502 | 5024.5 | 0.02767 | -1.8363 | 0.0148 |
| T-1200 | Prev | 0.179 | 5977 | 0.027 | - | 0.016 |
| | Base | 0.1754 | 5917.6 | 0.03295 | -1.5463 | 0.0161 |
| | Appr | 0.1725 | 5318.8 | 0.03701 | -1.8313 | 0.0161 |
| | Iter | 0.1417 | 5071 | 0.02292 | -1.4763 | 0.0161 |
| | Iter(D) | **0.1413** | 5021.1 | 0.02316 | -1.4763 | 0.0161 |
| | Iter.M | 0.1493 | 5185.4 | 0.02576 | -1.5363 | 0.0161 |
| T-$300^2$ | Prev | 0.17 | 5791 | 0.027 | - | 0.015 |
| | Base | 0.1686 | 5514.9 | 0.03329 | -1.1513 | 0.015 |
| | Appr | 0.1434 | 4105.4 | 0.03296 | -1.8063 | 0.015 |
| | Iter | 0.1249 | 3873.2 | 0.02514 | -1.4763 | 0.015 |
| | Iter(D) | **0.1244** | 3833.7 | 0.02526 | -1.4763 | 0.015 |
| | Iter.M | 0.1308 | 3987.2 | 0.02721 | -1.5713 | 0.015 |
| R-600 | Prev | 1.352 | 201131 | 0.112 | - | 0.501 |
| | Base | 0.6046 | 1144.8 | 0.507 | -1.8263 | 0.4925 |
| | Appr | 0.5653 | 390.25 | 0.5066 | -2.4713 | 0.4925 |
| | Iter(D) | 0.5681 | 431.62 | 0.5066 | -2.4513 | 0.4925 |
| | Iter.M | **0.5616** | 340.62 | 0.5065 | -2.5263 | 0.4925 |
| T-$200^2_{10}$ | Base | 0.4165 | 21896 | 0.04706 | -1.1513 | 0.0208 |
| | Appr | 0.2621 | 11068 | 0.0366 | -1.4213 | 0.0208 |
| | Iter | **0.2145** | 9821 | 0.02229 | -1.1513 | 0.0208 |
| | Iter(D) | 0.2311 | 9758.5 | 0.03071 | -1.1513 | 0.0208 |
| | Iter.M | 0.2728 | 13406 | 0.02605 | -1.1513 | 0.0208 |
| T-$600_{10}$ | Base | 0.2879 | 12674 | 0.03854 | -1.1513 | 0.018 |
| | Appr | 0.2424 | 9095.8 | 0.04159 | -1.6013 | 0.018 |
| | Iter | **0.2132** | 8697.9 | 0.02947 | -1.3063 | 0.018 |
| | Iter.M | 0.2227 | 8870.9 | 0.03294 | -1.4613 | 0.018 |
| T-$600^2_{10}$ | Base | 0.3472 | 17212 | 0.03884 | -1.1513 | 0.0186 |
| | Appr | 0.2896 | 11618 | 0.04723 | -2.0563 | 0.0186 |
| | Iter | **0.2431** | 10568 | 0.03057 | -1.5713 | 0.0186 |

# H Proofs and experiment for the low rank structure of the output Hessian

## H.1 Preliminaries

### H.1.1 Notations

We use $[n]$ to denote the set $\{1, \cdots, n\}$, and $\|M\|$ to denote the spectral norm of a matrix $M$. We use $\langle A, B \rangle$ to denote the Frobenius inner product of two matrices $A$ and $B$, namely $\langle A, B \rangle \triangleq \sum_{i,j} A_{i,j} B_{i,j}$. Denote $\mathrm{tr}(M)$ the trace of a matrix $M$ and denote $\mathbf{1}_c$ the all-one vector of dimension $c$ (the subscript may be omitted when it's clear from the context). Furthermore, for notation simplicity, we will say "with probability 1 over $\boldsymbol{W}^{(1)}/\boldsymbol{W}^{(2)}$, event $E$ is true" to denote

$$\lim_{n \to \infty} \lim_{d \to \infty} \Pr_{\boldsymbol{W}^{(1)} \sim \mathcal{N}(0, \frac{1}{d} \boldsymbol{I}_{nd}), \boldsymbol{W}^{(2)} \sim \mathcal{N}(0, \frac{1}{n} \boldsymbol{I}_{cn})} [E] = 1. \tag{66}$$

### H.1.2 Problem Setting

Consider a two layer fully connected ReLU neural network with input dimension $d$, hidden layer dimension $n$ and output dimension $c$. The network is trained with cross-entropy objective $\mathcal{L}$. Let $\sigma$ denote the element-wise ReLU activation function which acts as $\sigma(x) = x \cdot \mathbf{1}_{x \geq 0}$. Let $\boldsymbol{W}^{(1)} \in \mathbb{R}^{n \times d}$ and $\boldsymbol{W}^{(2)} \in \mathbb{R}^{c \times n}$ denote the weight matrices of the first and second layer respectively.

Let the neural network have standard normal input $\mathbf{x} \sim \mathcal{N}(0, \boldsymbol{I}_d)$. Denoting the output of the first and second layer as $\mathbf{y}$ and $\mathbf{z}$ respectively, we have $\mathbf{y} = \sigma(\boldsymbol{W}^{(1)}\mathbf{x})$ and $\mathbf{z} = \boldsymbol{W}^{(2)}\mathbf{y}$. Let $\mathbf{p} = \mathrm{softmax}(\mathbf{z})$ denote the softmax output of the network. Let $\mathbf{A} := \mathrm{diag}(\mathbf{p}) - \mathbf{p}\mathbf{p}^T$. From the previous analysis of Hessian, we have the output Hessian of the second layer can be written as $\boldsymbol{M} := \mathbb{E}[\mathbf{D}\boldsymbol{W}^{(2)T}\mathbf{A}\boldsymbol{W}^{(2)}\mathbf{D}]$, where $\mathbf{D} := \mathrm{diag}(\mathbf{1}_{\mathbf{y} \geq 0})$ is the diagonal random matrix representing the activations of ReLU function after the first layer.

In this problem, we look into the state of random Gaussian initialization, in which entries of both matrices are i.i.d. sampled from a standard normal distribution, and then re-scaled such that each row of $\boldsymbol{W}^{(1)}$ and $\boldsymbol{W}^{(2)}$ has norm 1. When taking $n$ and $d$ to infinity, with the concentration of norm in high-dimensional Gaussian random variables, in this problem we assume that entries of $\boldsymbol{W}^{(1)}$ are iid sampled from a zero-mean distribution with variance $1/d$, and entries of $\boldsymbol{W}^{(2)}$ are iid sampled from a zero-mean distribution with variance $1/n$. This initialization is standard in training neural networks.

Since our formula for the top eigenspace is going to depend on $\boldsymbol{W}^{(2)}$, throughout the section when we take expectation we condition on the value of $\boldsymbol{W}^{(1)}$ and $\boldsymbol{W}^{(2)}$. The expectation is only taken over the input $\mathbf{x} \sim \mathcal{N}(0, \boldsymbol{I}_d)$ (due to concentration taking expectation on $\mathbf{x}$ is similar to having many samples from the input distribution). In this case, the output Hessian is defined as:

$$\boldsymbol{M} \triangleq \mathbb{E}[\mathbf{D}\boldsymbol{W}^{(2)T}\mathbf{A}\boldsymbol{W}^{(2)}\mathbf{D}]. \tag{67}$$

## H.2 Main Theorem and Proof Sketch

In this section, we will provide a formal statement of our main theorem and its proof sketch. First, we state our main theorem:

**Theorem H.1.** For all $\epsilon > 0$,

$$\lim_{n \to \infty} \lim_{d \to \infty} \Pr_{\boldsymbol{W}^{(1)} \sim \mathcal{N}(0, \frac{1}{d} \boldsymbol{I}_{nd}), \boldsymbol{W}^{(2)} \sim \mathcal{N}(0, \frac{1}{n} \boldsymbol{I}_{cn})} \left[ \left( \frac{\lambda_c(\boldsymbol{M})}{\lambda_{c-1}(\boldsymbol{M})} \bigg|_{\boldsymbol{W}^{(1)}, \boldsymbol{W}^{(2)}} \right) < \epsilon \right] = 1. \tag{68}$$

Besides, for all $\epsilon > 0$, if we define $S_1$ as the top $c - 1$ eigenspace of $\boldsymbol{M}$, and $S_2$ as $\mathcal{R}(\boldsymbol{W}) \backslash \{\boldsymbol{W} \cdot \mathbf{1}\}$ where $\mathcal{R}(\boldsymbol{W})$ is the row space of $\boldsymbol{W}$, then

$$\lim_{n \to \infty} \lim_{d \to \infty} \Pr_{\boldsymbol{W}^{(1)} \sim \mathcal{N}(0, \frac{1}{d} \boldsymbol{I}_{nd}), \boldsymbol{W}^{(2)} \sim \mathcal{N}(0, \frac{1}{n} \boldsymbol{I}_{cn})} [\mathrm{Overlap}\,(S_1, S_2) > 1 - \epsilon] = 1. \tag{69}$$

**Proof of Theorem H.1.**

First of all, let us repeat the expression for the output Hessian $\boldsymbol{M}$:

$$\boldsymbol{M} \triangleq \mathbb{E}[\mathbf{D}\boldsymbol{W}^{(2)T}\mathbf{A}\boldsymbol{W}^{(2)}\mathbf{D}]. \tag{70}$$

In the proof, we will first analyze the properties of $\mathbf{D}, \boldsymbol{W}^{(2)}, \mathbf{A}$ separately:

Firstly, $\mathbf{D}$ is a diagonal matrix with $0/1$ entries, and the following lemma shows that its entries are independent when the input dimension tends to infinity.

**Lemma 1.** *When $d \to \infty$, with probability 1 over $\boldsymbol{W}^{(1)}$, the entries of $\mathbf{D}$ are independent.*

Secondly, since each entry of $\boldsymbol{W}^{(2)}$ is sampled i.i.d. from a spherical Gaussian distribution, this matrix enjoys some very nice properties when the network width $n$ goes to infinity. We have the following lemma for $\boldsymbol{W}^{(2)}$.

**Lemma 2.** *(informal) When $n$ is large enough, each row of $\boldsymbol{W}^{(2)}$ has norm very close to 1, and these rows are nearly orthogonal to each other. Besides, the entries (and the average of the entries) cannot be too large.*

These properties (along with other useful properties) will be formally stated and proved in Appendix H.3.2.

As for matrix $\mathbf{A}$, it's very hard to compute its expectation explicitly because the generation of $\mathbf{A}$ involves softmax, but we are able to prove some useful properties of $\mathbb{E}[\mathbf{A}]$ as shown in the following lemma.

**Lemma 3.** $\tilde{\mathbf{A}} \triangleq \lim_{n \to \infty} \mathbb{E}[\mathbf{A}]$ *exist and is rank-$(c-1)$ with probability 1 over $\boldsymbol{W}^{(2)}$.*

This lemma is true because $\mathbf{A}$ is positive semi-definite (PSD) and is almost always of rank $(c-1)$. Besides, its null space always contains the all-one vector $\mathbf{1}_c$.

Having these properties of these three matrices, we then look at the expression of $M$ again to see how to compute this expectation. This expectation is not easy to compute as we condition on $\boldsymbol{W}^{(1)}$ and $\boldsymbol{W}^{(2)}$. With this conditioning, $\mathbf{D}$ and $\mathbf{A}$ are correlated and hard to decompose. This is when we need the most important observation in our proof: When the input dimension and the network width tend to infinity, $\mathbf{A}$ and $\mathbf{D}$ can be considered independent when computing $M$. These two matrices are actually not independent even when we take the limit: For example, if $\mathbf{D}$ happens to be 1 whenever the first row of $\boldsymbol{W}^{(2)}$ is positive, then the first logit output is going to be much larger than the rest which significantly skews the distribution of $\mathbf{A}$. However, since the computation of $M$ only contains finite-degree polynomials of $\mathbf{A}$ and $\mathbf{D}$, we only need a weaker form of independence, i.e., the distribution of $\mathbf{A}$ is approximately invariant condition on finite entries of $\mathbf{D}$, as shown in the following lemma:

**Lemma 4.** *Let $\mathcal{D}_\mathbf{X}$ denote the distribution of $\mathbf{X}$, and $TV(\mathcal{D}_1, \mathcal{D}_2)$ denote the total variation distance between $\mathcal{D}_1$ and $\mathcal{D}_2$. Then $\forall i, j \in [n], \forall \epsilon > 0$*

$$\lim_{n \to \infty} \lim_{d \to \infty} \Pr_{\boldsymbol{W}^{(1)} \sim \mathcal{N}(0, \frac{1}{d} \boldsymbol{I}_{nd}), \boldsymbol{W}^{(2)} \sim \mathcal{N}(0, \frac{1}{n} \boldsymbol{I}_{cn})} \left[ TV(\mathcal{D}_\mathbf{A}, \mathcal{D}_{\mathbf{A} | \mathbf{D}_{i,i} = \mathbf{D}_{j,j} = 1}) > \epsilon \right] = 0. \quad (71)$$

The intuition behind this is that $\mathbf{A}$ is uniquely determined by the output of the second layer $\mathbf{z}$, where $\mathbf{z} = \boldsymbol{W}^{(2)} \mathbf{y}$ is $c$-dimensional and $\mathbf{y}$ is an $n$-dimensional vector. Since $\mathbf{D} = \text{diag}(\mathbf{1}_{\mathbf{y} \geq 0})$, fixing finite number of entries in $\mathbf{D}$ is equivalent to fixing the signs of finite entries of $\mathbf{y}$. When $n$ is large enough compare to $c$, only constraining finite entries of $\mathbf{y}$ shouldn't change the distribution of $\mathbf{z}$ by much. The formal proof of this theorem is given in Appendix H.3.4.

Having Lemma 4, we can equivalently consider $\mathbf{A}$ and $\mathbf{D}$ as independent matrices. To formalize this, we need the following definition:

**Definition H.1.** Let $\mathbf{D}'$ be an independent copy of $\mathbf{D}$ and also independent of $\mathbf{A}$. Define $M^* \triangleq \mathbb{E}[\mathbf{D}' \boldsymbol{W}^{(2)T} \mathbf{A} \boldsymbol{W}^{(2)} \mathbf{D}']$.

In other words, $M^*$ is the matrix which has the same expression as $M$ except that we assume $\mathbf{D}$ is independent of $\mathbf{A}$ in $M^*$. Then we know that $M$ and $M^*$ are essentially the same. Since $\mathbf{D}$ is a diagonal matrix with $0/1$ entries, multiplying $\mathbf{D}$ at both sides of a matrix is equivalent to independently zero out each row and corresponding column with probability $\frac{1}{2}$. Thus, the probability of each diagonal entry to be kept is $\frac{1}{2}$ while for off-diagonal ones it's $\frac{1}{4}$. Formally, we have

$$M^* \approx \frac{1}{4} \left( \mathbb{E} \left[ \boldsymbol{W}^{(2)T} \mathbf{A} \boldsymbol{W}^{(2)} \right] + \text{diag}(\mathbb{E}[\boldsymbol{W}^{(2)T} \mathbf{A} \boldsymbol{W}^{(2)}]) \right). \quad (72)$$

We have two terms on the right hand side of this equation: $T_1 \triangleq \mathbb{E}\left[W^{(2)T}\mathbf{A}W^{(2)}\right]$ and $T_2 \triangleq$ diag($\mathbb{E}[W^{(2)T}\mathbf{A}W^{(2)}]$). We make two observations of these two terms. On the one hand, they have the same trace. This is because the diagonal entries of these two matrices are exactly the same. On the other hand, $T_1$ is a low rank matrix but $T_2$ is approximately full rank. For $T_1$, since the expectation is taken over $\mathbf{x}$ only, we know that $T_1 = W^{(2)T}\mathbb{E}[\mathbf{A}]W^{(2)}$. Note that $\mathbb{E}[A]$ is a rank $(c-1)$ PSD matrix, so $T_1$ is also PSD and has rank at most $(c-1)$. $T_2$ is a diagonal matrix, and each diagonal entry equals a quadratic form $w_i^{(2)T}\mathbb{E}[A]w_i^{(2)}$ where $w_i^{(2)}$ is the $i$-th column of $W^{(2)}$ ($i \in [n]$). This term is always positive unless $w_i^{(2)}$ lies in the span of $\mathbf{1}_c$, which happens with probability 0. Actually, due to the random nature of $w_i^{(2)}$'s, the diagonal terms in $T_2$ do not differ too much from one another.

To summarize, $T_1$ and $T_2$ are both PSD matrices with the same trace, while $T_1$ is low-rank but $T_2$ is approximately full rank. This intuitively indicates that the positive eigenvalues of $T_1$ is significantly large than those of $T_2$, making the positive eigenvalues of $T_1$ the dominating eigenvalues of $M^*$ and those of $T_2$ the thin but long tail of $M^*$'s eigenvalue spectrum.

Now we have know that $T_1$ is almost the only contributing term to the top eigenvalues and eigenspaces of $M^*$, so we only need to analyze these for $T_1 = W^{(2)T}\mathbb{E}[\mathbf{A}]W^{(2)}$. Since the rows of $W^{(2)}$ are close to rank 1 and almost mutually orthogonal, the matrix $T_1$ will roughly keeps all the eigenvalues of $\mathbb{E}[A]$, and the top $(c-1)$ eigenspace should roughly be the "$W^{(2)}$-rotated" version of $\mathbb{R}^c \backslash \{\mathbf{1}_c\}$, i.e., $\mathcal{R}(W^{(2)}) \backslash \{W^{(2)} \cdot \mathbf{1}_c\}$.

Despite the arguments above, we have some technical difficulties, the biggest of which is that the dimensions of $M$ and $M^*$ will become infinite when $n$ goes to infinity. To tackle this problem, we introduce an indirect and more complicated way to do the proof. We first project these matrices onto the row span of $W^{(2)}$ and show that this projection roughly keeps all the information of these matrices. Formally, we have the following lemma:

**Lemma 5.** *With probability 1 over $W^{(2)}$,*

$$\lim_{n\to\infty} \frac{\left\|WMW^T\right\|_F^2}{\|M\|_F^2} = 1.$$

After that, we do the analysis in this finite-dimensional span and finish the proof of our main theorem.

$\square$

## H.3 Detailed Proof

### H.3.1 Proof of Lemma 1

We first restate Lemma 1 here:

**Lemma 1.** *When $d \to \infty$, with probability 1 over $W^{(1)}$, the entries of $\mathbf{D}$ are independent.*

**Proof of Lemma 1.** Remember that $\mathbf{D} := \text{diag}(\mathbf{1}_{\mathbf{y}\geq 0})$. The off-diagonal entries of $\mathbf{D}$ are always 0 and independent of anything. For the diagonal entries, each diagonal entry is decided by a corresponding entry of $\mathbf{y}$. Therefore, we only need to prove that the entries of $\mathbf{y}$ are independent, and we have the following lemma:

**Lemma 6.** *When $d \to \infty$, with probability 1 over $W^{(1)}$, the entries of $\mathbf{y}$ are independent.*

**Proof of Lemma 6.** We will prove this lemma using the multivariate Lindeberg-Feller CLT. For each $i \in [n]$, let $w_i \in \mathbb{R}^d$ denote the $i$-th column vector of $W^{(1)}$. Let $\mathbf{u}_i = w_i \mathbf{x}_i$, then we have

$$\mathbf{u} = \sum_{i=1}^d \mathbf{u}_i = \sum_{i=1}^d w_i \mathbf{x}_i = W^{(1)}\mathbf{x} \tag{73}$$

Note that $\mathbf{x}_i$'s are i.i.d standard Gaussian. It has the moments:

$$\mathbb{E}[\mathbf{x}_i] = 0, \qquad \mathbb{E}[(\mathbf{x}_i - \mathbb{E}[\mathbf{x}_i])^2] = 1, \qquad \mathbb{E}[(\mathbf{x}_i - \mathbb{E}[\mathbf{x}_i])^4] = 3. \tag{74}$$

It follows that

$$\text{Var}[\mathbf{u}_i] = \text{Var}[w_i \mathbf{x}_i] = w_i w_i^T. \tag{75}$$

Let $\boldsymbol{V} = \sum_{i=1}^{n} Var[\mathbf{u}_i]$,

$$\boldsymbol{V} = \sum_{i=1}^{d} \boldsymbol{w}_i \boldsymbol{w}_i^T = \boldsymbol{W}^{(1)} \boldsymbol{W}^{(1)T}. \tag{76}$$

As $d \to \infty$, from Lemma 9 (we replace $n$, $\boldsymbol{W}$ with $d$ and $\boldsymbol{W}^{(1)}$) we have $\boldsymbol{W}^{(1)} \boldsymbol{W}^{(1)T} \to \boldsymbol{I}_n$ in probability, therefore $\lim_{d\to\infty} \boldsymbol{V} = \boldsymbol{I}_n$.

We now verify the Lindeberg condition of independent random vectors $\{\mathbf{u}_1, \ldots, \mathbf{u}_n\}$. First observe that the fourth moments of the $\mathbf{u}_i$ are sufficiently small.

$$
\begin{aligned}
\lim_{d\to\infty} \sum_{i=1}^{d} \mathbb{E}\left[\|\mathbf{u}_i\|^3\right] &= \lim_{d\to\infty} \sum_{i=1}^{d} \mathbb{E}\left[\left(\sum_{j=1}^{n} \left(\boldsymbol{W}_{ji}^{(1)} \mathbf{x}_i\right)^2\right)^2\right] \\
&\leq \lim_{d\to\infty} \sum_{i=1}^{d} \mathbb{E}\left[n^2 \left(\left(\max_{j\in[n]} \boldsymbol{W}_{ji}^{(2)}\right)^2 \mathbf{x}_i^2\right)^2\right] \\
&\leq \lim_{d\to\infty} n^2 \left(\max_{i\in[d],j\in[n]} \boldsymbol{W}_{ji}^{(2)}\right)^4 \sum_{i=1}^{n} \mathbb{E}\left[(\mathbf{x}_i - \mathbb{E}[\mathbf{x}_i])^4\right]
\end{aligned}
\tag{77}
$$

Since $\mathbb{E}[(\mathbf{x}_i - \mathbb{E}[\mathbf{x}_i])^4] = 3$ and $\max_{i\in[d],j\in[n]} |\boldsymbol{W}_{ji}^{(2)}| < 2d^{-\frac{1}{3}}$ with probability 1 from Lemma 11, it follows from above that

$$\lim_{d\to\infty} \sum_{i=1}^{d} \mathbb{E}\left[\|\mathbf{u}_i\|^4\right] \leq n^2 \lim_{d\to\infty} \left(2d^{-\frac{1}{3}}\right)^4 \sum_{i=1}^{d} 3 = 48n^2 \lim_{d\to\infty} d^{-\frac{4}{3}} d = 48n^2 \lim_{d\to\infty} d^{-\frac{1}{3}} = 0. \tag{78}$$

For any $\epsilon > 0$, since $\|\mathbf{u}_i\| > \epsilon$ in the domain of integration,

$$
\begin{aligned}
\lim_{d\to\infty} \sum_{i=1}^{d} \mathbb{E}\left[\|\mathbf{u}_i\|^2 \mathbf{1}\left[\|\mathbf{u}_i\| > \epsilon\right]\right] &< \lim_{d\to\infty} \sum_{i=1}^{d} \mathbb{E}\left[\frac{\|\mathbf{u}_i\|^2}{\epsilon^2} \|\mathbf{u}_i\|^2 \mathbf{1}\left[\|\mathbf{u}_i\| > \epsilon\right]\right] \\
&\leq \frac{1}{\epsilon^2} \lim_{d\to\infty} \sum_{i=1}^{n} \mathbb{E}\left[\|\mathbf{u}_i\|^4\right] = 0.
\end{aligned}
\tag{79}
$$

As the Lindeberg Condition is satisfied, with $\lim_{n\to\infty} \boldsymbol{V} = \boldsymbol{I}_c$ in probability we have

$$\lim_{d\to\infty} \mathbf{u} = \lim_{d\to\infty} \sum_{i=1}^{d} \mathbf{u}_i \xrightarrow{D} \mathcal{N}\left(0, \boldsymbol{I}_n\right). \tag{80}$$

Thus, since $\mathbf{u}$ converges to $\mathcal{N}\left(0, \boldsymbol{I}_n\right)$ in distribution with probability 1 over $\boldsymbol{W}^{(1)}$, entries of $\mathbf{u}$ are independent. Since $\mathbf{y} = \sigma(\mathbf{u})$ and ReLu is an entry-wise operator, entries of $\mathbf{y}$ are independent. $\square$

Since the diagonal entries of $\mathbf{D}$ are uniquely determined by the corresponding entries of $\mathbf{y}$, we know that when $d \to \infty$, with probability 1 over $\boldsymbol{W}^{(1)}$, the entries of $\mathbf{D}$ are independent. This finishes the proof of Lemma 1. $\square$

### H.3.2 Proof of Lemma 2

We first restate Lemma 2 here:

**Lemma 2.** *(informal) When $n$ is large enough, each row of $\boldsymbol{W}^{(2)}$ has norm very close to 1, and these rows are nearly orthogonal to each other. Besides, the entries (and the average of the entries) cannot be too large.*

This is not a formal lemma and will act as the intuition behind the properties of $\boldsymbol{W}^{(2)}$. In this section, we will formally state the properties we need and prove them. Besides, for simplicity of notations, we use $\boldsymbol{W}$ to denote $\boldsymbol{W}^{(2)}$ from now on unless otherwise stated.

**Lemma 7.** *For all $i \in [c]$, for all $\epsilon > 0$, $\lim_{n\to\infty} \Pr\left(\left|\sum_{j=1}^{n} \boldsymbol{W}_{ij}\right| \geq \epsilon\right) = 0$.*

**Proof of Lemma 7.** Since each entry of $\boldsymbol{W}$ is initialized independently from $\mathcal{N}(0, \frac{1}{n})$, by Central Limit Theorem we have $\sum_{j=1}^n \boldsymbol{W}_{ij} \sim \mathcal{N}(0, \frac{1}{n})$. For any $\epsilon > 0$, fix $\epsilon$. By chebyshev's inequality,

$$\lim_{n \to \infty} \mathbb{P}\left( \left| \sum_{j=1}^n \boldsymbol{W}_{ij} \right| \geq \epsilon \right) < \lim_{n \to \infty} \frac{1}{n\epsilon^2} = 0. \tag{81}$$

$\square$

**Lemma 8.** *For all $\epsilon > 0$, $\lim_{n \to \infty} \Pr\left( \left| \|\boldsymbol{W}\|_F^2 - c \right| \geq \epsilon \right) = 0$.*

*Besides, for all $i \in [c]$, $\lim_{n \to \infty} \Pr\left( \left| \|\boldsymbol{W}_i\|^2 - 1 \right| \geq \epsilon \right) = 0$.*

**Proof of Lemma 8.** Since each entry of $\boldsymbol{W}$ is initialized independently from $\mathcal{N}(0, \frac{1}{n})$, we know that $n\|\boldsymbol{W}\|_F^2 = \sum_{i=1}^c \sum_{j=1}^n n\boldsymbol{W}_{i,j}^2$ follows a $\chi_{cn}^2$-distribution. Using the tail bound provided by Lemma 1 in Laurent & Massart (2000), we know that for large enough $n$,

$$\Pr\left( |n\|\boldsymbol{W}\|_F^2 - cn| \geq n\epsilon \right) \geq \Pr\left( |n\|\boldsymbol{W}\|_F^2 - cn| \geq 2\sqrt{c}n^{3/4} + 2n^{1/2} \right) \leq 2\exp(-n^{1/2}). \tag{82}$$

In other words,

$$\lim_{n \to \infty} \Pr\left( \left| \|\boldsymbol{W}\|_F^2 - c \right| \geq \epsilon \right) = \lim_{n \to \infty} \Pr\left( |n\|\boldsymbol{W}\|_F^2 - cn| \geq n\epsilon \right) = 0. \tag{83}$$

Similarly, $\forall i \in [c]$, $n\|\boldsymbol{W}_i\|_F^2$ follows a $\chi_n^2$-distribution, so for large enough $n$,

$$\Pr\left( |n\|\boldsymbol{W}_i\|_F^2 - n| \geq n\epsilon \right) \leq \Pr\left( |n\|\boldsymbol{W}_i\|_F^2 - n| \geq 2n^{3/4} + 2n^{1/2} \right) \leq 2\exp(-n^{1/2}), \tag{84}$$

which indicates that

$$\lim_{n \to \infty} \Pr\left( \left| \|\boldsymbol{W}_i\|^2 - 1 \right| \geq \epsilon \right) = \lim_{n \to \infty} \Pr\left( |n\|\boldsymbol{W}_i\|^2 - n| \geq n\epsilon \right) = 0. \tag{85}$$

$\square$

**Lemma 9.** *For all $\epsilon > 0$, $\lim_{n \to \infty} \Pr\left( \|\boldsymbol{W}\boldsymbol{W}^T - \boldsymbol{I}_c\| \geq \epsilon \right) = 0$.*

*Besides, for all $i, j \in [c]$, $\lim_{n \to \infty} \Pr\left( |(\boldsymbol{W}\boldsymbol{W}^T)_{i,j} - \delta_{i,j}| \geq \epsilon \right) = 0$. Here $\delta$ is the Kronecker delta function, i.e., $\delta_{i,j} = \mathbf{1}_{[i=j]}$.*

**Proof of Lemma 9.** Since each entry of $\boldsymbol{W}$ is initialized independently from $\mathcal{N}(0, \frac{1}{n})$, we know that $\boldsymbol{W}\boldsymbol{W}^T$ follows Wishart distribution $\boldsymbol{W}_c(\frac{1}{n}\boldsymbol{I}_c, n)$. Using the third tail bound in Theorem 1 of Zhu (2012), for large enough $n$, we get

$$\begin{aligned}
\Pr\left( \|\boldsymbol{W}\boldsymbol{W}^T - \boldsymbol{I}_c\| \geq \epsilon \right) &= \Pr\left( \left\| \frac{1}{n}\boldsymbol{W}\boldsymbol{W}^T - \frac{1}{n}\boldsymbol{I}_c \right\| \geq \frac{\epsilon}{n} \right) \\
&\leq \Pr\left( \left\| \frac{1}{n}\boldsymbol{W}\boldsymbol{W}^T - \frac{1}{n}\boldsymbol{I}_c \right\| \geq \frac{1}{n}\left( \sqrt{2(c+1)}n^{-1/4} + 2cn^{-1/2} \right) \right) \\
&\leq 2c\exp(-\sqrt{n}).
\end{aligned} \tag{86}$$

Therefore,

$$\forall \epsilon > 0, \lim_{n \to \infty} \Pr\left( \|\boldsymbol{W}\boldsymbol{W}^T - \boldsymbol{I}_c\| \geq \epsilon \right) = 0. \tag{87}$$

Moreover, for all $i, j \in [c]$, we have

$$\begin{aligned}
\Pr\left( |(\boldsymbol{W}\boldsymbol{W}^T)_{i,j} - \delta_{i,j}| \geq \epsilon \right) &\leq \Pr\left( \sum_{i,j=1}^c \left( (\boldsymbol{W}\boldsymbol{W}^T)_{i,j} - \delta_{i,j} \right)^2 \geq \epsilon^2 \right) \\
&= \Pr\left( \|\boldsymbol{W}\boldsymbol{W}^T - \boldsymbol{I}_c\|_F^2 \geq \epsilon^2 \right) \\
&= \Pr\left( \|\boldsymbol{W}\boldsymbol{W}^T - \boldsymbol{I}_c\| \geq \frac{\epsilon}{\sqrt{c}} \right),
\end{aligned} \tag{88}$$

which implies that for all $i, j \in [c]$,

$$\lim_{n \to \infty} \Pr\left(|(\boldsymbol{W}\boldsymbol{W}^T)_{i,j} - \delta_{i,j}| \geq \epsilon\right) = 0. \tag{89}$$

$\square$

**Lemma 10.** *Let $P_{\boldsymbol{W}}$ be the projection matrix onto the row span of $\boldsymbol{W}$, then for all $\epsilon > 0$,*

$$\lim_{n \to \infty} \Pr\left[\left\|\boldsymbol{W}^T\boldsymbol{W} - P_{\boldsymbol{W}}\right\|_F^2 > \epsilon\right] = 0. \tag{90}$$

**Proof of Lemma 10.** Without loss of generality, we assume that $\epsilon < 1$. Let $\boldsymbol{W}_i(i \in [c])$ be the $i$-th row of $\boldsymbol{W}$, and we will do the Gram–Schmidt process for the rows of $\boldsymbol{W}$. Specifically, the Gram–Schmidt process is as following: Assume that $\{\overline{\boldsymbol{W}}_i\}_{i=1}^k$ are the already normalized basis, we set $\boldsymbol{W}'_{k+1} \triangleq \boldsymbol{W}_{k+1} - \sum_{i=1}^k \langle \boldsymbol{W}_{k+1}, \overline{\boldsymbol{W}}_i \rangle$ and $\overline{\boldsymbol{W}}_{k+1} \triangleq \frac{\boldsymbol{W}'_{k+1}}{\|\boldsymbol{W}'_{k+1}\|}$. Finally, from the definition of projection matrix, we know that $P_{\boldsymbol{W}} = \overline{\boldsymbol{W}}^T \overline{\boldsymbol{W}}$.

Let $\epsilon' \triangleq \frac{\epsilon^2}{c^3 \cdot 16^{2c+1}}$, from Lemma 9 we know that $\forall i, j \in [c], \lim_{n \to \infty} \Pr\left(|\boldsymbol{W}_i \boldsymbol{W}_j^T - \delta_{i,j}| \geq \epsilon'\right) = 0$. Besides, from Lemma 8 we know that $\forall i \in [c], \lim_{n \to \infty} \Pr\left(|\|\boldsymbol{W}_i\|^2 - 1| \geq \epsilon\right) = 0$. Then we use induction to bound the difference between $\boldsymbol{W}$ and $\overline{\boldsymbol{W}}$. Specifically, we will show that $\forall i \in [c], \|\overline{\boldsymbol{W}}_i - \boldsymbol{W}_i\| \leq 8^i \epsilon'$. For notation simplicity, in the following proof we will not repeat the probability argument and assume that $\forall i, j \in [c], |\boldsymbol{W}_i \boldsymbol{W}_j^T - \delta_{i,j}| \leq \epsilon'$ and $\forall i \in [c], |\|\boldsymbol{W}_i\|^2 - 1| \leq \epsilon'$. We will only use these inequalities finite times so applying a union bound will give the probability result.

For $i = 1$, we know that $\overline{\boldsymbol{W}}_1 = \frac{\boldsymbol{W}_1}{\|\boldsymbol{W}_1\|}$ and $|\|\boldsymbol{W}_1\| - 1| \leq \epsilon'$, so $\|\overline{\boldsymbol{W}}_i - \boldsymbol{W}_i\| \leq \epsilon'$.

If our inductive hypothesis holds for $i \leq k$, then for $i = k + 1$, we have

$$\begin{aligned}
\forall j \leq k, |\langle \boldsymbol{W}_i, \overline{\boldsymbol{W}}_j \rangle| &\leq |\langle \boldsymbol{W}_i, \boldsymbol{W}_j \rangle| + |\langle \boldsymbol{W}_i, \overline{\boldsymbol{W}}_j - \boldsymbol{W}_j \rangle| \\
&\leq \epsilon' + \|\boldsymbol{W}_i\| \cdot \|\overline{\boldsymbol{W}}_j - \boldsymbol{W}_j\| \\
&\leq \epsilon' + (1 + \epsilon')8^j \epsilon' \\
&\leq (2^{3j+1} + 1)\epsilon'.
\end{aligned} \tag{91}$$

Therefore,

$$\|\boldsymbol{W}'_i - \boldsymbol{W}_i\| \leq \sum_{j \in [k]} |\langle \boldsymbol{W}_i, \overline{\boldsymbol{W}}_j \rangle| \leq \epsilon' + \sum_{j \in [k]} (2^{3j+1} + 1)\epsilon' \leq (2^{3k+2} - 1)\epsilon', \tag{92}$$

and

$$|\|\boldsymbol{W}'_i\| - 1| \leq |\|\boldsymbol{W}_i\| - 1| + \|\boldsymbol{W}'_i - \boldsymbol{W}_i\| \leq 2^{3k+2}\epsilon'. \tag{93}$$

Thus,

$$\begin{aligned}
\|\overline{\boldsymbol{W}}_i - \boldsymbol{W}_i\| &\leq \|\overline{\boldsymbol{W}}_i - \boldsymbol{W}'_i\| + \|\boldsymbol{W}'_i - \boldsymbol{W}_i\| \\
&\leq |\|\boldsymbol{W}'_i\| - 1| + \|\boldsymbol{W}'_i - \boldsymbol{W}_i\| \\
&\leq 8^{k+1}\epsilon',
\end{aligned} \tag{94}$$

which finishes the induction and implies that $\forall \epsilon > 0, \forall i \in [c], \|\overline{\boldsymbol{W}}_i - \boldsymbol{W}_i\| \leq 8^i \epsilon'$. Thus,

$$\|\overline{\boldsymbol{W}} - \boldsymbol{W}\|_F^2 = \sum_{i \in [c]} \|\overline{\boldsymbol{W}}_i - \boldsymbol{W}_i\|^2 \leq c \cdot 16^c \epsilon'. \tag{95}$$

This means that

$$\begin{aligned}
\|\boldsymbol{W}^T\boldsymbol{W} - P_{\boldsymbol{W}}\|_F &= \|\boldsymbol{W}^T\boldsymbol{W} - \overline{\boldsymbol{W}}^T\overline{\boldsymbol{W}}\|_F \\
&\leq 2\|\boldsymbol{W} - \overline{\boldsymbol{W}}\|_F \|\overline{\boldsymbol{W}}\|_F + \|\boldsymbol{W} - \overline{\boldsymbol{W}}\|_F^2 \\
&\leq 2c \cdot \sqrt{c} \cdot 8^c \sqrt{\epsilon'} + c \cdot 16^c \epsilon' \leq \epsilon.
\end{aligned} \tag{96}$$

$\square$

**Lemma 11.** *The largest entry of $\boldsymbol{W}^{(2)}$ is reasonably small with high probability as $n$ goes to infinity, namely,*

$$\lim_{n\to\infty} \mathbb{P}\left[\max_{i\in[c],j\in[n]} \left|\boldsymbol{W}_{ij}^{(2)}\right| > 2n^{-\frac{1}{3}}\right] = 0 \tag{97}$$

**Proof of Lemma 11.** For i.i.d. random variables $\mathbf{x}_1,\cdots,\mathbf{x}_n \sim \mathcal{N}(0,1)$, by concentration inequality on maximum of Gaussian random variables, for any $t > 0$, we have

$$\mathbb{P}\left[\max_{i=1}^{n} \mathbf{x}_i > \sqrt{2\log(2n)} + t\right] < 2e^{-\frac{t^2}{2}}. \tag{98}$$

For any $i,j$, since $\boldsymbol{W}_{ij}^{(2)}$ are i.i.d. sampled from $\mathcal{N}(0,\frac{1}{n})$, with rescaling of $1/\sqrt{n}$ we may substitute $\mathbf{x}_j$ with $\boldsymbol{W}_{ij}^{(2)}$. It follows that

$$\mathbb{P}\left[\max_{i\in[c],j\in[n]} \boldsymbol{W}_{ij}^{(2)} > \frac{\sqrt{2\log(2cn)} + t}{\sqrt{n}}\right] < 2e^{-\frac{t^2}{2}}. \tag{99}$$

Taking $t = n^{\frac{1}{6}}$, with $c$ as constant, for large $n$ we have $\sqrt{2\log(2cn)} < n^{\frac{1}{6}}$. Thus for large $n$,

$$\begin{aligned}
\mathbb{P}\left[\max_{i\in[c],j\in[n]} \boldsymbol{W}_{ij}^{(2)} > 2n^{-\frac{1}{3}}\right] &= \mathbb{P}\left[\max_{i\in[c],j\in[n]} \boldsymbol{W}_{ij}^{(2)} > \frac{n^{\frac{1}{6}} + n^{\frac{1}{6}}}{\sqrt{n}}\right] \\
&< \mathbb{P}\left[\max_{i\in[c],j\in[n]} \boldsymbol{W}_{ij}^{(2)} > \frac{\sqrt{2\log(2n)} + n^{\frac{1}{6}}}{\sqrt{n}}\right] < 2e^{-\frac{n^{\frac{1}{3}}}{2}}.
\end{aligned} \tag{100}$$

With the same argument, we have

$$\mathbb{P}\left[\min_{i\in[c],j\in[n]} \boldsymbol{W}_{ij}^{(2)} < -2n^{-\frac{1}{3}}\right] < 2e^{-\frac{n^{\frac{1}{3}}}{2}}. \tag{101}$$

Passing $n$ to infinity completes the proof. $\qquad\square$

### H.3.3 Proof of Lemma 3

We first restate Lemma 3 here:

**Lemma 3.** $\tilde{\mathbf{A}} \triangleq \lim_{n\to\infty} \mathbb{E}[\mathbf{A}]$ *exist and is rank-$(c-1)$ with probability 1 over $\boldsymbol{W}^{(2)}$.*

Before proving Lemma 3, we need some knowledge about the distribution of $\mathbf{A}$. Since $\mathbf{A}$ is determined by the vector $\mathbf{z}$, it suffice to know the distribution of $\mathbf{z}$:

**Lemma 12.** $\lim_{n\to\infty} \mathbf{z} \xrightarrow{d} \mathcal{N}(0, \frac{\pi-1}{2\pi}\boldsymbol{I}_c)$ *with probability 1 over $\boldsymbol{W}^{(2)}$.*

**Proof of Lemma 12.** We will prove this lemma using the multivariate Lindeberg-Feller CLT. For each $i \in [n]$, let $\boldsymbol{w}_i \in \mathbb{R}^c$ denote the $i$-th column vector of $\boldsymbol{W}^{(2)}$. Let $\mathbf{v}_i = \boldsymbol{w}_i(\mathbf{y}_i - \mathbb{E}[\mathbf{y}_i])$, then we have

$$\mathbf{z} = \sum_{i=1}^{n} \boldsymbol{w}_i \mathbf{y}_i = \sum_{i=1}^{n} \mathbf{v}_i + \mathbb{E}[\mathbf{y}_i] \sum_{i=1}^{n} \boldsymbol{w}_i. \tag{102}$$

From Lemma 6 we know $\mathbf{y}_i$'s are i.i.d. rectified half standard normal. It has the moments:

$$\mathbb{E}[\mathbf{y}_i] = \frac{1}{\sqrt{2\pi}}, \qquad \mathbb{E}[(\mathbf{y}_i - \mathbb{E}[\mathbf{y}_i])^2] = \frac{\pi-1}{2\pi}, \qquad \mathbb{E}[(\mathbf{y}_i - \mathbb{E}[\mathbf{y}_i])^4] = \frac{6\pi^2 - 10\pi - 3}{4\pi^2} < 1. \tag{103}$$

It follows that

$$Var[\mathbf{v}_i] = Var[\boldsymbol{w}_i \mathbf{y}_i] = \frac{\pi-1}{2\pi} \boldsymbol{w}_i \boldsymbol{w}_i^T. \tag{104}$$

Let $\boldsymbol{V} = \sum_{i=1}^{n} Var[\mathbf{v}_i]$,

$$\boldsymbol{V} = \frac{\pi-1}{2\pi} \sum_{i=1}^{n} \boldsymbol{w}_i \boldsymbol{w}_i^T = \frac{\pi-1}{2\pi} \boldsymbol{W}^{(2)} \boldsymbol{W}^{(2)T}. \tag{105}$$

As $n \to \infty$, from Lemma 9 we have $W^{(2)}W^{(2)T} \to I_c$ in probability, therefore $\lim_{n\to\infty} V = \frac{\pi-1}{2\pi}I_c$.

We now verify the Lindeberg condition of independent random vectors $\{\mathbf{v}_1, \ldots, \mathbf{v}_n\}$. First observe that the fourth moments of the $\mathbf{v}_i$'s are sufficiently small.

$$\lim_{n\to\infty} \sum_{i=1}^{n} \mathbb{E}\left[\|\mathbf{v}_i\|^4\right] = \lim_{n\to\infty} \sum_{i=1}^{n} \mathbb{E}\left[\left(\sum_{j=1}^{c}\left(W_{ji}^{(2)}(\mathbf{y}_i - \mathbb{E}[\mathbf{y}_i])\right)^2\right)^2\right]$$

$$\leq \lim_{n\to\infty} \sum_{i=1}^{n} \mathbb{E}\left[c^2\left(\left(\max_{j\in[c]} W_{ji}^{(2)}\right)^2 (\mathbf{y}_i - \mathbb{E}[\mathbf{y}_i])^2\right)^2\right] \tag{106}$$

$$\leq \lim_{n\to\infty} c^2\left(\max_{i\in[n],j\in[c]} W_{ji}^{(2)}\right)^4 \sum_{i=1}^{n} \mathbb{E}\left[(\mathbf{y}_i - \mathbb{E}[\mathbf{y}_i])^4\right].$$

Since $\mathbb{E}[(\mathbf{y}_i - \mathbb{E}[\mathbf{y}_i])^4] < 1$ and $\max_{i\in[n],j\in[c]}|W_{ji}^{(2)}| < 2n^{-\frac{1}{3}}$ with probability 1 from Lemma 11, it follows that

$$\lim_{n\to\infty} \sum_{i=1}^{n} \mathbb{E}\left[\|\mathbf{v}_i\|^4\right] \leq c^2 \lim_{n\to\infty}\left(2n^{-\frac{1}{3}}\right)^4 \sum_{i=1}^{n} 1 = c^2 \lim_{n\to\infty} 16n^{-\frac{4}{3}}n = 16c^2 \lim_{n\to\infty} n^{-\frac{1}{3}} = 0. \tag{107}$$

For any $\epsilon > 0$, since $\|\mathbf{v}_i\| > \epsilon$ in the domain of integration,

$$\lim_{n\to\infty} \sum_{i=1}^{n} \mathbb{E}\left[\|\mathbf{v}_i\|^2 \mathbf{1}\left[\|\mathbf{v}_i\| > \epsilon\right]\right] < \lim_{n\to\infty} \sum_{i=1}^{n} \mathbb{E}\left[\frac{\|\mathbf{v}_i\|^2}{\epsilon^2}\|\mathbf{v}_i\|^2 \mathbf{1}\left[\|\mathbf{v}_i\| > \epsilon\right]\right]$$

$$\leq \frac{1}{\epsilon^2} \lim_{n\to\infty} \sum_{i=1}^{n} \mathbb{E}\left[\|\mathbf{v}_i\|^4\right] = 0. \tag{108}$$

As the Lindeberg Condition is satisfied, with $\lim_{n\to\infty} V = \frac{\pi-1}{2\pi}I_c$ we have

$$\lim_{n\to\infty} \sum_{i=1}^{n} \mathbf{v}_i \xrightarrow{d} \mathcal{N}\left(0, \frac{\pi-1}{2\pi}I_c\right). \tag{109}$$

By Lemma 7, we have $\lim_{n\to\infty} \boldsymbol{w}_i = \vec{0}$ with probability 1 over $W^{(2)}$, therefore plugging (Eq. (109)) into (Eq. (102)) we have

$$\lim_{n\to\infty} \mathbf{z} \xrightarrow{d} \mathcal{N}\left(0, \frac{\pi-1}{2\pi}I_c\right). \tag{110}$$

$\square$

After that, we can proceed to prove Lemma 3.

**Proof of Lemma 3.** Note that each entry of $\mathbf{A}$ is a quadratic function of $p$, and $p$ is a continuous function of $\mathbf{z}$. Therefore, we consider $\mathbf{A}$ as a function of $\mathbf{z}$ and write $\mathbf{A}(\mathbf{z})$ when necessary. From Lemma 12 we know that $\lim_{n\to\infty} \mathbf{z}$ follows a standard normal distribution $\mathcal{N}(0, \alpha I_c)$ with probability 1 over $W$, where $\alpha$ is some absolute constant. Therefore, $\tilde{\mathbf{A}} \triangleq \lim_{n\to\infty} \mathbb{E}[\mathbf{A}]$ exist and it equals $\mathbb{E}[\mathbf{A}(\lim_{n\to\infty} \mathbf{z})] = \mathbb{E}_{\mathbf{z}\sim\mathcal{N}(0,\alpha I_c)}[\mathbf{A}(\mathbf{z})]$. For notation simplicity, we will omit the statement "with probability 1 over $W$" when there is no confusion.

From the definition of $\mathbf{A}$ we know that $\mathbf{A} \triangleq \text{diag}(p) - pp^T$ where $p$ is the vector obtained by applying softmax to $\mathbf{z}$, so $\sum_{i=1}^{c} p_i = 1$ and $\forall i \in [c], p_i \in (0, 1)$. Therefore, for any vector $p$ satisfying the previous conditions, we have

$$\mathbf{1}^T\mathbf{A}\mathbf{1} = \sum_{i=1}^{c}\left(p_i - \sum_{j=1}^{c} p_i p_j\right) = \sum_{i=1}^{c}(p_i - p_i) = 0, \tag{111}$$

where $\mathbf{1}$ is the all-one vector. Therefore, we know that $\mathbf{A}$ has an eigenvalue 0 with eigenvector $\frac{1}{\sqrt{c}}\mathbf{1}$. This means that $\mathbb{E}[\mathbf{A}]$ also has an eigenvalue 0 with eigenvector $\frac{1}{\sqrt{c}}\mathbf{1}$. Thus, $\mathbb{E}[\mathbf{A}]$ is at most of rank $(c-1)$.

Then we analyze the other $(c-1)$ eigenvalues of $\tilde{\mathbf{A}}$. Since $\mathbf{A} = QQ^T$ where $Q = \text{diag}(\sqrt{p})(\mathbf{I}_c - \mathbf{1}p^T)$, we know that $\mathbf{A}$ is always a positive semi-definite (PSD) matrix, which indicates that $E[\mathbf{A}]$ must also be PSD. Assume the $c$ eigenvalues of $\tilde{\mathbf{A}}$ are $\lambda_1 \geq \lambda_2 \geq \cdots \geq \lambda_{c-1} \geq \lambda_c = 0$. Therefore, by definition, we have

$$\lambda_{c-1} = \min_{\boldsymbol{v} \in S, \|\boldsymbol{v}\|=1} \boldsymbol{v}^T \tilde{\mathbf{A}} \boldsymbol{v} = \mathbb{E}_{\mathbf{z} \sim \mathcal{N}(0, \alpha \boldsymbol{I}_c)} \left[ \min_{\boldsymbol{v} \in S, \|\boldsymbol{v}\|=1} \boldsymbol{v}^T \mathbf{A} \boldsymbol{v} \right], \tag{112}$$

where $S \triangleq \mathbb{R}^c \backslash \mathcal{R}\{\mathbf{1}^T\}$ is the orthogonal subspace of the span of $\mathbf{1}$. $\boldsymbol{v} \in S$ implies that $\boldsymbol{v} \perp \mathbf{1}$, i.e., $\sum_{i=1}^c \boldsymbol{v}_i = 0$.

Direct computation gives us

$$\boldsymbol{v}^T \mathbf{A} \boldsymbol{v} = \sum_{i=1}^c \boldsymbol{v}_i^2 p_i - \left( \sum_{i=1}^c \boldsymbol{v}_i p_i \right)^2. \tag{113}$$

Define two vectors $\boldsymbol{a}, b \in \mathbb{R}^c$ as $\forall i \in [c]$, $\boldsymbol{a}_i \triangleq \boldsymbol{v}_i \sqrt{p_i}$, $\boldsymbol{b}_i \triangleq \sqrt{p_i}$, then $\|\boldsymbol{b}\|^2 = \sum_{i=1}^c p_i = 1$ and

$$\boldsymbol{v}^T \mathbf{A} \boldsymbol{v} = \|\boldsymbol{a}\|^2 - \langle \boldsymbol{a}, \boldsymbol{b} \rangle^2 = \|\boldsymbol{a}\|^2 \cdot \|\boldsymbol{b}\|^2 - \langle \boldsymbol{a}, \boldsymbol{b} \rangle^2. \tag{114}$$

Therefore,

$$\boldsymbol{v}^T \mathbf{A} \boldsymbol{v} \geq \|\boldsymbol{a}\|^2 \|\boldsymbol{b}\|^2 \sin^2 \theta(\boldsymbol{a}, \boldsymbol{b}), \tag{115}$$

where $\theta(\boldsymbol{a}, b)$ is the angle between $\boldsymbol{a}$ and $\boldsymbol{b}$, i.e., $\theta(\boldsymbol{a}, \boldsymbol{b}) \triangleq \arccos \frac{\langle \boldsymbol{a}, \boldsymbol{b} \rangle}{\|\boldsymbol{a}\| \|\boldsymbol{b}\|}$. Define $p_0 \triangleq \min_{i \in [c]} p_i$, then

$$\|\boldsymbol{a}\|^2 = \sum_{i=1}^c \boldsymbol{v}_i^2 p_i \geq \sum_{i=1}^c \boldsymbol{v}_i^2 p_0 = p_0 \|\boldsymbol{v}\|^2 = p_0. \tag{116}$$

Since $\|\boldsymbol{b}\| = 1$, we have

$$\sin^2 \theta(\boldsymbol{a}, \boldsymbol{b}) = \frac{\|\boldsymbol{a} - \langle \boldsymbol{a}, \boldsymbol{b} \rangle \cdot \boldsymbol{b}\|^2}{\|\boldsymbol{a}\|^2}. \tag{117}$$

Besides,

$$\begin{aligned}
\|\boldsymbol{a} - \langle \boldsymbol{a}, \boldsymbol{b} \rangle \cdot \boldsymbol{b}\|^2 &= \sum_{i=1}^c \left( \boldsymbol{v}_i \sqrt{p_i} - \left( \sum_{j=1}^c \boldsymbol{v}_j p_j \right) \sqrt{p_i} \right)^2 \\
&= \sum_{i=1}^c p_i \left( \boldsymbol{v}_i - \sum_{j=1}^c \boldsymbol{v}_j p_j \right)^2 \\
&\geq p_0 \sum_{i=1}^c \left( \boldsymbol{v}_i - \sum_{j=1}^c \boldsymbol{v}_j p_j \right)^2.
\end{aligned} \tag{118}$$

Define $s \triangleq \arg \max_{i \in [c]} \boldsymbol{v}_i$ and $t \triangleq \arg \min_{i \in [c]} \boldsymbol{v}_i$, then

$$\sum_{i=1}^c \left( \boldsymbol{v}_i - \sum_{j=1}^c \boldsymbol{v}_j p_j \right)^2 \geq \left( \boldsymbol{v}_s - \sum_{j=1}^c \boldsymbol{v}_j p_j \right)^2 + \left( \boldsymbol{v}_t - \sum_{j=1}^c \boldsymbol{v}_j p_j \right)^2 \geq \frac{(\boldsymbol{v}_s - \boldsymbol{v}_t)^2}{2}. \tag{119}$$

From $\|\boldsymbol{v}\| = 1$ we know that $\max_{i \in [c]} |\boldsymbol{v}_i| \geq \frac{1}{\sqrt{c}}$. Besides, since $\sum_{i=1}^c \boldsymbol{v}_i = 0$, we have $\boldsymbol{v}_s > 0 > \boldsymbol{v}_t$. Therefore, $\boldsymbol{v}_s - \boldsymbol{v}_t > \max_{i \in [c]} |\boldsymbol{v}_i| \geq \frac{1}{\sqrt{c}}$. As a result,

$$\|\boldsymbol{a} - \langle \boldsymbol{a}, \boldsymbol{b} \rangle \cdot \boldsymbol{b}\|^2 \geq p_0 \cdot \frac{(\boldsymbol{v}_s - \boldsymbol{v}_t)^2}{2} > \frac{p_0}{2c}. \tag{120}$$

Moreover,

$$\|\boldsymbol{a}\|^2 = \sum_{i=1}^c \boldsymbol{v}_i^2 p_i \leq \sum_{i=1}^c p_i = 1. \tag{121}$$

Thus,

$$\sin^2\theta(\boldsymbol{a}, \boldsymbol{b}) \geq \frac{\frac{p_0}{2c}}{1} = \frac{p_0}{2c}, \tag{122}$$

which means that

$$\boldsymbol{v}^T \mathbf{A} \boldsymbol{v} \geq p_0 \cdot 1 \cdot \frac{p_0}{2c} = \frac{p_0^2}{2c}. \tag{123}$$

Now we analyze the distribution of $p_0$. Since $\mathbf{z}$ follows a spherical Gaussian distribution $\mathcal{N}(0, \alpha \boldsymbol{I}_c)$, we know that the entries of $\mathbf{z}$ are totally independent. Besides, for each entry $\mathbf{z}_i (i \in [c])$, we have $|\mathbf{z}_i| < \alpha$ with probability $\beta$, where $\beta \approx 0.68$ is an absolute constant. Therefore, with probability $\beta^c$, forall entries $\mathbf{z}_i (i \in [c])$, we have $|\mathbf{z}_i| < \alpha$. In this case,

$$p_0 = \frac{\exp(\min_{i \in [c]} \mathbf{z}_i)}{\sum_{i=1}^c \exp(\mathbf{z}_i)} \geq \frac{\exp(-\alpha)}{c \exp(\alpha)}. \tag{124}$$

In other cases, we know that $p_0 > 0$. Thus,

$$\lambda_{c-1} = \mathbb{E}_{\mathbf{z} \sim \mathcal{N}(0, \alpha \boldsymbol{I}_c)} \left[ \min_{\boldsymbol{v} \in S, \|v\|=1} \boldsymbol{v}^T \mathbf{A} \boldsymbol{v} \right] \geq \beta^c \cdot \frac{\left( \frac{\exp(-\alpha)}{c \exp(\alpha)} \right)^2}{2c}. \tag{125}$$

The right hand side is independent of $n$. Therefore, $\lambda_{c-1} > 0$, which means that $\tilde{\mathbf{A}}$ has exactly $(c-1)$ positive eigenvalues and a 0 eigenvalue, and the eigenvalue gap between the smallest positive eigenvalue and 0 is independent of $n$. $\qquad\square$

### H.3.4 Proof of Lemma 4

We first restate Lemma 4 here:

**Lemma 4.** *Let $\mathcal{D}_{\mathbf{X}}$ denote the distribution of $\mathbf{X}$, and $TV(\mathcal{D}_1, \mathcal{D}_2)$ denote the total variation distance between $\mathcal{D}_1$ and $\mathcal{D}_2$. Then $\forall i, j \in [n], \forall \epsilon > 0$*

$$\lim_{n \to \infty} \lim_{d \to \infty} \Pr_{\boldsymbol{W}^{(1)} \sim \mathcal{N}(0, \frac{1}{d} \boldsymbol{I}_{nd}), \boldsymbol{W}^{(2)} \sim \mathcal{N}(0, \frac{1}{n} \boldsymbol{I}_{cn})} \left[ TV(\mathcal{D}_{\mathbf{A}}, \mathcal{D}_{\mathbf{A}|\mathbf{D}_{i,i}=\mathbf{D}_{j,j}=1}) > \epsilon \right] = 0. \tag{71}$$

**Proof of Lemma 4.** The proof of this lemma requires knowledge about the distributions of $\mathbf{A}$ condition on two diagonal entries of $\mathbf{D}$. Since $\mathbf{A}$ can be uniquely determined by $\mathbf{z}$, it is enough for us to know the distribution of $\mathbf{z}$ condition on the two entries of $\mathbf{D}$. We use the following lemma to analyze this:

**Lemma 13.** *With probability 1 over $\boldsymbol{W}^{(2)}$, for any $i, j \in [n]$, for any $(p, q) \in \{0, 1\}^2$,*

$$\lim_{n \to \infty} P(\mathbf{z}|D_{ii} = p, D_{jj} = q) \xrightarrow{d} \mathbf{z}. \tag{126}$$

**Proof of Lemma 13.** With $\{\boldsymbol{w}_1, \ldots, \boldsymbol{w}_n\}$ and $\{\mathbf{v}_1, \ldots, \mathbf{v}_n\}$ defined as above, since different summands contributing to $\mathbf{z}$ are independent, and every $\mathbf{v}_i$ is only affected by its corresponding $D_{ii}$,

$$\begin{aligned} P(\mathbf{z}|D_{ii} = p, D_{jj} = q) &= \mathbf{z} - \mathbf{v}_i + P(\mathbf{v}_i|D_{ii} = p) - \mathbf{v}_j + P(\mathbf{v}_j|D_{jj} = q) \\ &= \mathbf{z} - \boldsymbol{w}_i(\mathbf{y}_i - P(\mathbf{y}_i|D_{ii} = p)) + \boldsymbol{w}_j(\mathbf{y}_j - P(\mathbf{y}_j|D_{jj} = q)). \end{aligned} \tag{127}$$

For any $i \in [n]$, with the condition of $D_{ii} = p$, when $p = 0$, $P(\mathbf{y}_i|D_{ii} = p) = 0$; when conditioned with $p = 1$, $P(\mathbf{y}_i|D_{ii} = p)$ is of a half standard normal distribution truncated at 0. In both cases the conditional distribution of $P(\mathbf{y}_i|D_{ii} = p)$ and hence $\mathbf{y}_i - P(\mathbf{y}_i|D_{ii} = p)$ has bounded mean and variance. For any $\boldsymbol{w}_i$, by Lemma 11 we have

$$\|\boldsymbol{w}_i\| \leq \sqrt{c \left( \max_{i \in [c], j \in [n]} \boldsymbol{W}_{ij}^{(2)} \right)^2} < \sqrt{4cn^{-\frac{2}{3}}} \text{ with probability 1 over } \boldsymbol{W}^{(2)}. \tag{128}$$

Since $\lim_{n \to \infty} \sqrt{4cn^{-\frac{2}{3}}} = 0$, as $n$ goes to infinity we have

$$\boldsymbol{w}_i(\mathbf{y}_i - P(\mathbf{y}_i|D_{ii} = p)) \xrightarrow{d} 0 \text{ with probability 1 over } \boldsymbol{W}^{(2)}. \tag{129}$$

Therefore

$$P(\mathbf{z}|D_{ii} = p, D_{jj} = q) = \mathbf{z} - \boldsymbol{w}_i(\mathbf{y}_i - P(\mathbf{y}_i|D_{ii} = p)) + \boldsymbol{w}_j(\mathbf{y}_j - P(\mathbf{y}_j|D_{jj} = q)) \xrightarrow{d} \mathbf{z}. \quad (130)$$

$\square$

From Lemma 13 we conclude that

$$\lim_{n\to\infty} \lim_{d\to\infty} \Pr_{\boldsymbol{W}^{(1)}\sim\mathcal{N}(0,\frac{1}{d}\boldsymbol{I}_{nd}),\boldsymbol{W}^{(2)}\sim\mathcal{N}(0,\frac{1}{n}\boldsymbol{I}_{cn})} \left[ TV(\mathcal{D}_\mathbf{z}, \mathcal{D}_{\mathbf{z}|\mathbf{D}_{i,i}=\mathbf{D}_{j,j}=1}) > \epsilon \right] = 1. \quad (131)$$

Since $\mathbf{A}$ can be uniquely determined by $\mathbf{z}$, we have

$$TV(\mathcal{D}_\mathbf{z}, \mathcal{D}_{\mathbf{z}|\mathbf{D}_{i,i}=\mathbf{D}_{j,j}=1}) \geq TV(\mathcal{D}_\mathbf{A}, \mathcal{D}_{\mathbf{A}|\mathbf{D}_{i,i}=\mathbf{D}_{j,j}=1}). \quad (132)$$

Therefore,

$$\lim_{n\to\infty} \lim_{d\to\infty} \Pr_{\boldsymbol{W}^{(1)}\sim\mathcal{N}(0,\frac{1}{d}\boldsymbol{I}_{nd}),\boldsymbol{W}^{(2)}\sim\mathcal{N}(0,\frac{1}{n}\boldsymbol{I}_{cn})} \left[ TV(\mathcal{D}_\mathbf{A}, \mathcal{D}_{\mathbf{A}|\mathbf{D}_{i,i}=\mathbf{D}_{j,j}=1}) > \epsilon \right] = 0. \quad (133)$$

This finishes the proof of Lemma 4. $\square$

### H.3.5 Proof of Lemma 5

We first restate Lemma 5 here:

**Lemma 5.** *With probability 1 over $\boldsymbol{W}^{(2)}$,*

$$\lim_{n\to\infty} \frac{\left\|\boldsymbol{W}\boldsymbol{M}\boldsymbol{W}^T\right\|_F^2}{\left\|\boldsymbol{M}\right\|_F^2} = 1.$$

**Proof of Lemma 5.** To prove the equivalence between $\left\|\boldsymbol{W}\boldsymbol{M}\boldsymbol{W}^T\right\|_F^2$ and $\left\|\boldsymbol{M}\right\|_F^2$, we need some other terms, including terms containing $\boldsymbol{M}^*$, as bridges. To prove the equivalence between $\boldsymbol{M}$ and $\boldsymbol{M}^*$, we need the following lemma which explains the reason why we only need the weaker sense of independence (Lemma 4) instead of the total independence between $\mathbf{A}$ and $\mathbf{D}$.

**Lemma 14.** *Let $p(\mathbf{A}, \mathbf{D})$ be a homogeneous polynomial of $\mathbf{A}$ and $\mathbf{D}$ and is degree 1 in $\mathbf{A}$ and degree 2 in $\mathbf{D}$, and let the coefficients in $p$ are upper bounded in $\ell_1$-norm by an absolute constant. Also let $\mathbf{A}'$ be an independent copy of $\mathbf{A}$. Then*

$$\lim_{n\to\infty} \lim_{d\to\infty} \Pr_{\boldsymbol{W}^{(1)}\sim\mathcal{N}(0,\frac{1}{d}\boldsymbol{I}_{nd}),\boldsymbol{W}^{(2)}\sim\mathcal{N}(0,\frac{1}{n}\boldsymbol{I}_{cn})} \left[ |\mathbb{E}[p(\mathbf{A}, \mathbf{D})] - \mathbb{E}[p(\mathbf{A}', \mathbf{D})]| > \epsilon \right] = 0. \quad (134)$$

**Proof of Lemma 14.** Assume that $p(\mathbf{A}, \mathbf{D}) = \sum_{i=1}^m c_i \mathbf{A}_{s(i),t(i)} \mathbf{D}_{u(i),u(i)} \mathbf{D}_{v(i),v(i)}$, then from linearity of expectation we know

$$\mathbb{E}[p(\mathbf{A}, \mathbf{D})] = \sum_{i=1}^m c_i \mathbb{E}[\mathbf{A}_{s(i),t(i)} \mathbf{D}_{u(i),u(i)} \mathbf{D}_{v(i),v(i)}]. \quad (135)$$

Since the entries of $\mathbf{D}$ can only be 0 or 1, we have

$$\mathbb{E}[\mathbf{A}_{s(i),t(i)} \mathbf{D}_{u(i),u(i)} \mathbf{D}_{v(i),v(i)}] = \mathbb{E}[\mathbf{A}_{s(i),t(i)} | \mathbf{D}_{u(i),u(i)} = \mathbf{D}_{v(i),v(i)} = 1]. \quad (136)$$

Assume that the upper bound of the sum of $|c_i|$s is $\alpha$, i.e., $\sum_{i=1}^m |c_i| \geq \alpha$. Set $\epsilon' = \frac{\epsilon}{\alpha}$ and from Lemma 4 we know that

$$\lim_{n\to\infty} \lim_{d\to\infty} \Pr_{\boldsymbol{W}^{(1)}\sim\mathcal{N}(0,\frac{1}{d}\boldsymbol{I}_{nd}),\boldsymbol{W}^{(2)}\sim\mathcal{N}(0,\frac{1}{n}\boldsymbol{I}_{cn})} \left[ TV(\mathcal{D}_\mathbf{A}, \mathcal{D}_{\mathbf{A}|\mathbf{D}_{i,i}=\mathbf{D}_{j,j}=1}) > \epsilon' \right] = 0. \quad (137)$$

In other words, with probability 1 we have $TV(\mathcal{D}_\mathbf{A}, \mathcal{D}_{\mathbf{A}|\mathbf{D}_{i,i}=\mathbf{D}_{j,j}=1}) \leq \epsilon'$. Besides, since $\forall i, j \in [c], i \neq j, p_i, p_j, p_i + p_j \in (0, 1)$, each entry of $A$ (either $p_i - p_i^2$ or $-p_i p_j$) must be in $(-\frac{1}{4}, \frac{1}{4})$. Therefore, when $n$ and $d$ goes to infinity, with probability 1 over $\boldsymbol{W}^{(1)}$ and $\boldsymbol{W}^{(2)}$ we have

$$\|\mathbb{E}[\mathbf{A}_{s(i),t(i)} | \mathbf{D}_{u(i),u(i)} = \mathbf{D}_{v(i),v(i)} = 1] - \mathbb{E}[\mathbf{A}_{s(i),t(i)}]\|$$
$$\leq TV(\mathcal{D}_\mathbf{A}, \mathcal{D}_{\mathbf{A}|\mathbf{D}_{i,i}=\mathbf{D}_{j,j}=1}) \cdot \left( \frac{1}{4} - \left(-\frac{1}{4}\right) \right) \leq \frac{\epsilon'}{2}. \quad (138)$$

Thus,

$$|\mathbb{E}[p(\mathbf{A},\mathbf{D})] - \mathbb{E}[p(\mathbf{A}',\mathbf{D})]| \leq \sum_{i=1}^{m} |c_i| \cdot \|\mathbb{E}[\mathbf{A}_{s(i),t(i)}|\mathbf{D}_{u(i),u(i)} = \mathbf{D}_{v(i),v(i)} = 1] - \mathbb{E}[\mathbf{A}_{s(i),t(i)}]\|$$

$$\leq \left(\sum_{i=1}^{m} |c_i|\right) \cdot \frac{\epsilon'}{2} \leq \alpha \cdot \frac{\epsilon}{2\alpha} < \epsilon. \tag{139}$$

This finishes the proof of Lemma 14. $\qquad\square$

After this, using Lemma 4, we have the following lemmas:

**Lemma 15.** *with probability 1 over $\mathbf{W}$,*

$$\lim_{n \to \infty} \frac{\|\mathbf{M}\|_F^2}{\|\mathbf{M}^*\|_F^2} = 1.$$

**Proof of Lemma 15.** Let $(\mathbf{D}', \mathbf{A}')$ be an independent copy of $(\mathbf{D}, \mathbf{A})$, then

$$\begin{aligned}
\|\mathbf{M}\|_F^2 &= \left\|\mathbb{E}[\mathbf{D}\mathbf{W}^T\mathbf{A}\mathbf{W}\mathbf{D}]\right\|_F^2 \\
&= \mathbb{E}\left[\langle \mathbf{D}\mathbf{W}^T\mathbf{A}\mathbf{W}\mathbf{D}, \mathbf{D}'\mathbf{W}^T\mathbf{A}'\mathbf{W}\mathbf{D}'\rangle\right] \\
&= \mathbb{E}\left[\mathrm{tr}\left(\mathbf{D}\mathbf{W}^T\mathbf{A}\mathbf{W}\mathbf{D}\mathbf{D}'\mathbf{W}^T\mathbf{A}'\mathbf{W}\mathbf{D}'\right)\right] \\
&= \mathbb{E}\left[\mathrm{tr}\left(\mathbf{W}\mathbf{D}'\mathbf{D}\mathbf{W}^T\mathbf{A}\mathbf{W}\mathbf{D}\mathbf{D}'\mathbf{W}^T\mathbf{A}'\right)\right].
\end{aligned} \tag{140}$$

Expressing the term inside the expectation as a polynomial of entries of $\mathbf{A}$, $\mathbf{D}$, $\mathbf{A}'$ and $\mathbf{D}'$, we get

$$\begin{aligned}
&\mathrm{tr}\left(\mathbf{W}\mathbf{D}'\mathbf{D}\mathbf{W}^T\mathbf{A}\mathbf{W}\mathbf{D}\mathbf{D}'\mathbf{W}^T\mathbf{A}'\rangle\right) \\
&= \sum_{i=1}^{c} \left(\mathbf{W}\mathbf{D}'\mathbf{D}\mathbf{W}^T\mathbf{A}\mathbf{W}\mathbf{D}\mathbf{D}'\mathbf{W}^T\mathbf{A}'\right)_{i,i} \\
&= \sum_{i,j=1}^{c} \left(\mathbf{W}\mathbf{D}'\mathbf{D}\mathbf{W}^T\mathbf{A}\right)_{i,j} \left(\mathbf{W}\mathbf{D}\mathbf{D}'\mathbf{W}^T\mathbf{A}'\right)_{j,i} \\
&= \sum_{i,j=1}^{c} \left(\sum_{k=1}^{c}\sum_{l=1}^{n} \mathbf{W}_{i,l}\mathbf{W}_{k,l}\mathbf{D}'_{l,l}\mathbf{D}_{l,l}\mathbf{A}_{k,j}\right) \left(\sum_{s=1}^{c}\sum_{t=1}^{n} \mathbf{W}_{j,t}\mathbf{W}_{s,t}\mathbf{D}_{t,t}\mathbf{D}'_{t,t}\mathbf{A}'_{s,i}\right) \\
&= \sum_{i,j,k,s=1}^{c}\sum_{l,t=1}^{n} \mathbf{W}_{i,l}\mathbf{W}_{k,l}\mathbf{W}_{j,t}\mathbf{W}_{s,t}\mathbf{A}_{k,j}\mathbf{A}'_{s,i}\mathbf{D}_{l,l}\mathbf{D}'_{l,l}\mathbf{D}_{t,t}\mathbf{D}'_{t,t}.
\end{aligned} \tag{141}$$

Now we can bound the $\ell_1$ norm of the coefficient of this polynomial as follows (note that absolute value of each entry of $\mathbf{A}$ is bounded by 1):

$$\begin{aligned}
&\left\|\sum_{i,j,k,s=1}^{c}\sum_{l,t=1}^{n} \mathbf{W}_{i,l}\mathbf{W}_{k,l}\mathbf{W}_{j,t}\mathbf{W}_{s,t}\right\|_1 \\
&\leq \sum_{i,j,k,s=1}^{c}\sum_{l,t=1}^{n} |\mathbf{W}_{i,l}| \cdot |\mathbf{W}_{k,l}| \cdot |\mathbf{W}_{j,t}| \cdot |\mathbf{W}_{s,t}| \\
&= \left(\sum_{i,k=1}^{c}\sum_{l=1}^{n} |\mathbf{W}_{i,l}| \cdot |\mathbf{W}_{k,l}|\right) \left(\sum_{j,s=1}^{c}\sum_{t=1}^{n} |\mathbf{W}_{j,t}| \cdot |\mathbf{W}_{s,t}|\right) \\
&\leq \left(\sum_{i,k=1}^{c}\sum_{l=1}^{n} \frac{\mathbf{W}_{i,l}^2 + \mathbf{W}_{k,l}^2}{2}\right) \left(\sum_{j,s=1}^{c}\sum_{t=1}^{n} \frac{\mathbf{W}_{j,t}^2 + \mathbf{W}_{s,t}^2}{2}\right) \\
&= \left(\sum_{i,k=1}^{c} \frac{\|\mathbf{W}_i\|^2 + \|\mathbf{W}_k\|^2}{2}\right) \left(\sum_{j,s=1}^{c} \frac{\|\mathbf{W}_j\|^2 + \|\mathbf{W}_s\|^2}{2}\right) \\
&= (c\|\mathbf{W}\|_F^2)^2 = c^2 \|\mathbf{W}\|_F^4.
\end{aligned} \tag{142}$$

When $n \to \infty$, we know that $\|\boldsymbol{W}\|_F^2 = O(c)$ with probability 1 over $\boldsymbol{W}$, so the coefficient of this polynomial is $\ell_1$-norm bounded. We know from Lemma 4 that the distribution of $\boldsymbol{A}$ is invariant condition on two entries of $\boldsymbol{D}$. Furthermore, since $\boldsymbol{A}'$ and $\boldsymbol{D}'$ are independent copies of $\boldsymbol{A}$ and $\boldsymbol{D}$, we know that the distribution of $(\boldsymbol{A}, \boldsymbol{A}')$ is invariant conditioning on two entries of $\boldsymbol{D}$ and two entries of $\boldsymbol{D}'$. Each term in this polynomial is a 4-th order term containing two entries from $\boldsymbol{D}$ and two from $\boldsymbol{D}'$. This combined with Lemma 14 gives us

$$\lim_{n\to\infty} \frac{\|\boldsymbol{M}\|_F^2}{\|\boldsymbol{M}^*\|_F^2} = 1. \tag{143}$$

$\square$

**Lemma 16.** *For all $i, j \in [c], \lim_{n\to\infty}((\boldsymbol{W}\boldsymbol{M}\boldsymbol{W}^T)_{i,j} - (\boldsymbol{W}\boldsymbol{M}^*\boldsymbol{W}^T)_{i,j}) = 0$. Thus,*

$$\lim_{n\to\infty} \frac{\left\|\boldsymbol{W}\boldsymbol{M}\boldsymbol{W}^T\right\|_F^2}{\left\|\boldsymbol{W}\boldsymbol{M}^*\boldsymbol{W}^T\right\|_F^2} = 1.$$

**Proof of Lemma 16.** This proof is very similar to that of Lemma 15. First, we focus on a single entry of the matrix $\boldsymbol{W}\boldsymbol{M}\boldsymbol{W}^T$ and express it as a polynomial of entries of $\boldsymbol{D}$:

$$
\begin{aligned}
(\boldsymbol{W}\boldsymbol{M}\boldsymbol{W}^T)_{i,j} &= \mathbb{E}[(\boldsymbol{W}\boldsymbol{D}\boldsymbol{W}^T\boldsymbol{A}\boldsymbol{W}\boldsymbol{D}\boldsymbol{W}^T)_{i,j}] \\
&= \mathbb{E}\left[\sum_{k=1}^c (\boldsymbol{W}\boldsymbol{D}\boldsymbol{W}^T\boldsymbol{A})_{i,k}(\boldsymbol{W}\boldsymbol{D}\boldsymbol{W}^T)_{k,j}\right] \\
&= \mathbb{E}\left[\sum_{k=1}^c \left(\sum_{s=1}^c \sum_{l=1}^n \boldsymbol{W}_{i,l}\boldsymbol{W}_{s,l}\boldsymbol{D}_{l,l}\boldsymbol{A}_{s,k}\right)\left(\sum_{t=1}^n \boldsymbol{W}_{k,j}\boldsymbol{W}_{j,t}\boldsymbol{D}_{t,t}\right)\right] \\
&= \mathbb{E}\left[\sum_{k,s=1}^c \sum_{l,t=1}^n \boldsymbol{A}_{s,k}\boldsymbol{W}_{i,l}\boldsymbol{W}_{s,l}\boldsymbol{W}_{k,t}\boldsymbol{W}_{j,t}\boldsymbol{D}_{l,l}\boldsymbol{D}_{t,t}\right].
\end{aligned}
\tag{144}
$$

Then we bound the $\ell_1$ norm of the coefficients of this polynomial as follows:

$$
\begin{aligned}
&\left\|\sum_{k,s=1}^c \sum_{l,t=1}^n \boldsymbol{A}_{s,k}\boldsymbol{W}_{i,l}\boldsymbol{W}_{s,l}\boldsymbol{W}_{k,t}\boldsymbol{W}_{j,t}\right\|_1 \\
&\leq \sum_{k,s=1}^c \sum_{l,t=1}^n |\boldsymbol{W}_{i,l}| \cdot |\boldsymbol{W}_{s,l}| \cdot |\boldsymbol{W}_{k,t}| \cdot |\boldsymbol{W}_{j,t}| \\
&= \left(\sum_{s=1}^c \sum_{l=1}^n |\boldsymbol{W}_{i,l}| \cdot |\boldsymbol{W}_{s,l}|\right)\left(\sum_{k=1}^c \sum_{t=1}^n |\boldsymbol{W}_{k,t}| \cdot |\boldsymbol{W}_{j,t}|\right) \\
&\leq \left(\sum_{s=1}^c \sum_{l=1}^n \frac{\boldsymbol{W}_{i,l}^2 + \boldsymbol{W}_{s,l}^2}{2}\right)\left(\sum_{k=1}^c \sum_{t=1}^n \frac{\boldsymbol{W}_{k,t}^2 + \boldsymbol{W}_{j,t}^2}{2}\right) \\
&= \left(c\|\boldsymbol{W}_i\|^2 + \|\boldsymbol{W}\|_F^2\right)\left(c\|\boldsymbol{W}_j\|^2 + \|\boldsymbol{W}\|_F^2\right) \\
&\leq (2c\|\boldsymbol{W}\|_F^2)^2 = 4c^2\|\boldsymbol{W}\|_F^4.
\end{aligned}
\tag{145}
$$

Similar to Lemma 15, this coefficient is $\ell_1$-norm bounded. Therefore, using Lemma 14, we have with probability 1 over $\boldsymbol{W}$, for all $i, j \in [c], \lim_{n\to\infty}((\boldsymbol{W}\boldsymbol{M}\boldsymbol{W}^T)_{i,j} - (\boldsymbol{W}\boldsymbol{M}^*\boldsymbol{W}^T)_{i,j}) = 0$, which indicates that

$$\lim_{n\to\infty} \frac{\left\|\boldsymbol{W}\boldsymbol{M}\boldsymbol{W}^T\right\|_F^2}{\left\|\boldsymbol{W}\boldsymbol{M}^*\boldsymbol{W}^T\right\|_F^2} = 1.$$

$\square$

**Lemma 17.**

$$\lim_{n\to\infty} \frac{\left\|\boldsymbol{W}\boldsymbol{M}^*\boldsymbol{W}^T\right\|_F^2}{\|\boldsymbol{M}^*\|_F^2} = 1.$$

**Proof of Lemma 17.** The proof of this lemma will be divided into two parts. In the first part, we will estimate the Frobenius norm of $\boldsymbol{M}^*$, and in the second part we do the same thing for $\boldsymbol{W}\boldsymbol{M}^*\boldsymbol{W}^T$.

**Part 1:** We know from the definition of $\boldsymbol{M}^*$ that

$$\boldsymbol{M}^* = \frac{1}{4}\left(\mathbb{E}[\boldsymbol{W}^T\mathbf{A}\boldsymbol{W}] + \mathrm{diag}(\mathbb{E}[\boldsymbol{W}^T\mathbf{A}\boldsymbol{W}])\right). \tag{146}$$

Define $\tilde{\mathbf{A}} := \mathbb{E}[\mathbf{A}]$, then

$$\mathbb{E}[\boldsymbol{W}^T\mathbf{A}\boldsymbol{W}] = \boldsymbol{W}^T\mathbb{E}[\mathbf{A}]\boldsymbol{W} = \boldsymbol{W}^T\tilde{\mathbf{A}}\boldsymbol{W}. \tag{147}$$

From Lemma 9, $\forall \epsilon' > 0$, with probability 1 we have $\left\|\boldsymbol{W}\boldsymbol{W}^T - I_c\right\| \leq \epsilon'$. Besides, from Kleinman & Athans (1968) we know that for positive semi-definite matrices $A$ and $B$ we have $\lambda_{\min}(A)\mathrm{tr}(B) \leq \mathrm{tr}(AB) \leq \lambda_{\max}(A)\mathrm{tr}(B)$, so

$$
\begin{aligned}
\left|\left\|\boldsymbol{W}^T\tilde{\mathbf{A}}\boldsymbol{W}\right\|_F^2 - \left\|\tilde{\mathbf{A}}\right\|_F^2\right| &= \left|\mathrm{tr}(\boldsymbol{W}^T\tilde{\mathbf{A}}\boldsymbol{W}\boldsymbol{W}^T\tilde{\mathbf{A}}\boldsymbol{W}) - \mathrm{tr}(\tilde{\mathbf{A}}\tilde{\mathbf{A}})\right| \\
&= \left|\mathrm{tr}(\boldsymbol{W}\boldsymbol{W}^T\tilde{\mathbf{A}}\boldsymbol{W}\boldsymbol{W}^T\tilde{\mathbf{A}}) - \mathrm{tr}(\tilde{\mathbf{A}}\tilde{\mathbf{A}})\right| \\
&\leq \left|\left(\left\|\boldsymbol{W}\boldsymbol{W}^T - I_c\right\| + 1\right)\mathrm{tr}(\tilde{\mathbf{A}}\boldsymbol{W}\boldsymbol{W}^T\tilde{\mathbf{A}}) - \mathrm{tr}(\tilde{\mathbf{A}}\tilde{\mathbf{A}})\right| \\
&= \left|\left(\left\|\boldsymbol{W}\boldsymbol{W}^T - I_c\right\| + 1\right)\mathrm{tr}(\boldsymbol{W}\boldsymbol{W}^T\tilde{\mathbf{A}}\tilde{\mathbf{A}}) - \mathrm{tr}(\tilde{\mathbf{A}}\tilde{\mathbf{A}})\right| \\
&\leq \left|\left(\left\|\boldsymbol{W}\boldsymbol{W}^T - I_c\right\| + 1\right)^2\mathrm{tr}(\tilde{\mathbf{A}}\tilde{\mathbf{A}}) - \mathrm{tr}(\tilde{\mathbf{A}}\tilde{\mathbf{A}})\right| \\
&\leq \left\|\boldsymbol{W}\boldsymbol{W}^T - I_c\right\|^2\left\|\tilde{A}\right\|_F^2 + 2\left\|\boldsymbol{W}\boldsymbol{W}^T - I_c\right\|\left\|\tilde{A}\right\|_F^2.
\end{aligned}
\tag{148}
$$

For any $\epsilon > 0$, set $\epsilon' = \min\{\frac{\epsilon}{4}, \frac{\sqrt{\epsilon}}{2}\}$ gives us with probability 1,

$$\lim_{n\to\infty}\frac{\left|\left\|\boldsymbol{W}^T\tilde{\mathbf{A}}\boldsymbol{W}\right\|_F^2 - \left\|\tilde{\mathbf{A}}\right\|_F^2\right|}{\left\|\tilde{\mathbf{A}}\right\|_F^2} = 0, \tag{149}$$

i.e.,

$$\lim_{n\to\infty}\frac{\left\|\boldsymbol{W}^T\tilde{\mathbf{A}}\boldsymbol{W}\right\|_F^2}{\left\|\tilde{\mathbf{A}}\right\|_F^2} = 1. \tag{150}$$

Besides, if we denote the $i$-th column of $\boldsymbol{W}$ by $\boldsymbol{w}_i$, then

$$
\begin{aligned}
\left\|diag(\mathbb{E}[\boldsymbol{W}^T\mathbf{A}\boldsymbol{W}])\right\|_F^2 &= \sum_{i=1}^n (\boldsymbol{w}_i^T\tilde{\mathbf{A}}\boldsymbol{w}_i)^2 \\
&\leq \sum_{i=1}^n \left(\left\|\boldsymbol{w}_i\right\|^2 \cdot \left\|\tilde{\mathbf{A}}\right\|\right)^2 \\
&= \left\|\tilde{\mathbf{A}}\right\|^2 \sum_{i=1}^n \left\|\boldsymbol{w}_i\right\|^4.
\end{aligned}
\tag{151}
$$

Since $\mathbb{E}[n^2\left\|\boldsymbol{w}_i\right\|^4] = c^2 + 2c$, by the additive form of Chernoff bound we get

$$\Pr\left(\sum_{i=1}^n\left\|\boldsymbol{w}_i\right\|^4 \geq \frac{c^2+3c}{n}\right) = \Pr\left(\frac{\sum_{i=1}^n n^2\left\|\boldsymbol{w}_i\right\|^4}{n} - (c^2+2c) \geq c\right) \leq e^{-2nc^2}. \tag{152}$$

Therefore, when $n \to \infty$, with probability 1 we have

$$\left\|\mathrm{diag}(\mathbb{E}[\boldsymbol{W}^T\mathbf{A}\boldsymbol{W}])\right\|_F^2 \leq \left\|\tilde{\mathbf{A}}\right\|^2\sum_{i=1}^n\left\|\boldsymbol{w}_i\right\|^4 \leq \left\|\tilde{\mathbf{A}}\right\|^2 \cdot \frac{c^2+3c}{n}. \tag{153}$$

Thus, with probability 1,

$$\lim_{n \to \infty} \frac{\left\|\text{diag}\left(\mathbb{E}\left[\boldsymbol{W}^T \mathbf{A} \boldsymbol{W}\right]\right)\right\|_F^2}{\left\|\boldsymbol{W}^T \tilde{\mathbf{A}} \boldsymbol{W}\right\|_F^2} = 0, \tag{154}$$

i.e.,

$$\lim_{n \to \infty} \frac{\frac{1}{16}\left\|\tilde{\mathbf{A}}\right\|_F^2}{\|\boldsymbol{M}^*\|_F^2} = 1. \tag{155}$$

**Part 2:** Plug equation Eq. (146) into $\boldsymbol{W} \boldsymbol{M}^* \boldsymbol{W}$ and we get

$$\boldsymbol{W} \boldsymbol{M}^* \boldsymbol{W} = \frac{1}{4}\left(\mathbb{E}[\boldsymbol{W}\boldsymbol{W}^T \mathbf{A} \boldsymbol{W}\boldsymbol{W}^T] + \mathbb{E}[\boldsymbol{W}\text{diag}(\boldsymbol{W}^T \mathbf{A} \boldsymbol{W})\boldsymbol{W}^T]\right). \tag{156}$$

Similar to **Part 1**, when $n \to \infty$, with probability 1, we have

$$\lim_{n \to \infty} \frac{\left\|\mathbb{E}[\boldsymbol{W}\boldsymbol{W}^T \mathbf{A} \boldsymbol{W}\boldsymbol{W}^T]\right\|_F^2}{\left\|\tilde{\mathbf{A}}\right\|_F^2} = 1. \tag{157}$$

Besides, when $n \to \infty$, with probability 1 we have

$$\left\|\boldsymbol{W}\text{diag}(\mathbb{E}[\boldsymbol{W}^T \mathbf{A} \boldsymbol{W}])\boldsymbol{W}^T\right\|_F^2 \le \|\boldsymbol{W}\|_F^2 \left\|\tilde{\mathbf{A}}\right\|^2 \sum_{i=1}^n \|\boldsymbol{w}_i\|^4 \le \left\|\tilde{\mathbf{A}}\right\|^2 \cdot \frac{c^2 + 3c}{n} \|\boldsymbol{W}\|_F^2. \tag{158}$$

As a result, with probability 1,

$$\lim_{n \to \infty} \frac{\left\|\boldsymbol{W}\text{diag}(\mathbb{E}[\boldsymbol{W}^T \mathbf{A} \boldsymbol{W}])\boldsymbol{W}^T\right\|_F^2}{\left\|\boldsymbol{W}\boldsymbol{W}^T \tilde{\mathbf{A}} \boldsymbol{W}\boldsymbol{W}^T\right\|_F^2} = 0, \tag{159}$$

i.e.,

$$\lim_{n \to \infty} \frac{\frac{1}{16}\left\|\tilde{\mathbf{A}}\right\|_F^2}{\|\boldsymbol{W}\boldsymbol{M}^* \boldsymbol{W}^T\|_F^2} = 1. \tag{160}$$

Combining the results of **Part 1** and **Part 2** proves this lemma. $\qquad\square$

Combining Lemma 15, Lemma 16, and Lemma 17 directly finishes the proof of Lemma 5. $\qquad\square$

### H.3.6   Proof of Theorem Theorem H.1

**Proof of Theorem Theorem H.1.** Now we can proceed to the proof of our main theorem. In this proof, we will use the bounds for $\|\boldsymbol{M}\|_F$, which are formalized into the lemma below:

**Lemma 18.** *With probability 1, $\lim_{n \to \infty} \|\boldsymbol{M}\|_F$ is both lower bounded and upper bounded by constants that are independent of $n$.*

**Proof of Lemma 18.**   From Lemma 15 we know that $\lim_{n \to \infty} \frac{\|\boldsymbol{M}\|_F}{\|\boldsymbol{M}^*\|_F} = 1$, so we only need to bound $\lim_{n \to \infty} \|\boldsymbol{M}^*\|_F$. Since $\boldsymbol{M}^* = \frac{1}{4}\left(\mathbb{E}[\boldsymbol{W}^T \mathbf{A} \boldsymbol{W}] + \text{diag}(\mathbb{E}[\boldsymbol{W}^T \mathbf{A} \boldsymbol{W}])\right)$ and $\langle \mathbb{E}[\boldsymbol{W}^T \mathbf{A} \boldsymbol{W}], \text{diag}(\mathbb{E}[\boldsymbol{W}^T \mathbf{A} \boldsymbol{W}])\rangle \ge 0$, we have

$$\|\boldsymbol{M}^*\|_F \ge \left\|\mathbb{E}[\boldsymbol{W}^T \mathbf{A} \boldsymbol{W}]\right\|_F = \left\|\boldsymbol{W}^T \tilde{\mathbf{A}} \boldsymbol{W}\right\|_F. \tag{161}$$

From equation Eq. (125), we know that the first $(c-1)$ eigenvalues of $\tilde{\mathbf{A}}$ is bounded by some constant $\gamma \triangleq \beta^c \cdot \frac{\left(\frac{\exp(-\alpha)}{c \exp(\alpha)}\right)^2}{2c}$ which is independent of $n$. Then we analyze the eigenvalues of $\boldsymbol{W}^T \tilde{\mathbf{A}} \boldsymbol{W}$: From Lemma 9 and set $\epsilon = \frac{1}{2}$, we know that the smallest singular value of $\boldsymbol{W}$ is lower bounded by $\frac{1}{2}$. Therefore, for any unit vector $v$ in the row span of $\boldsymbol{W}$, we have

$$v^T \boldsymbol{W}^T \tilde{\mathbf{A}} \boldsymbol{W} v = (\boldsymbol{W}v)^T \tilde{\mathbf{A}}(\boldsymbol{W}v) \ge \gamma \|\boldsymbol{W}v\|^2 \ge \frac{\gamma}{4}. \tag{162}$$

Thus, $\left\|\boldsymbol{W}^T \tilde{\mathbf{A}} \boldsymbol{W}\right\|_F \geq \frac{\gamma}{4}$, which is some constant that is independent of $n$.

Besides, since $\mathbf{D}$ is a dignonal matrix with 0/1 entries, and the absolute value of each entry of $\mathbf{A}$ is bounded by 1, we have

$$\|\boldsymbol{M}\|_F = \left\|\mathbb{E}[\mathbf{D}\boldsymbol{W}^T\mathbf{A}\boldsymbol{W}\mathbf{D}]\right\|_F \leq \left\|\mathbb{E}[\boldsymbol{W}^T\mathbf{A}\boldsymbol{W}]\right\|_F \leq \|\boldsymbol{W}\|_F^2 \|\mathbf{A}\|_F \leq c\|\boldsymbol{W}\|_F^2. \tag{163}$$

From Lemma 8, we know that with probability 1, $\|\boldsymbol{W}\|_F^2 \leq 2c$, therefore, $\|\boldsymbol{M}\|_F$ is upper bounded by $2c^2$, which is independent of $n$. $\qquad\square$

Now we are ready to prove our main theorem.

From Lemma 5 we have

$$\lim_{n\to\infty} \frac{\left\|\boldsymbol{W}\boldsymbol{M}\boldsymbol{W}^T\right\|_F^2}{\|\boldsymbol{M}\|_F^2} = 1. \tag{164}$$

Then we consider $\left\|\boldsymbol{W}^T\boldsymbol{W}\boldsymbol{M}\boldsymbol{W}^T\boldsymbol{W}\right\|_F^2$. Note that

$$\begin{aligned}
\left\|\boldsymbol{W}^T\boldsymbol{W}\boldsymbol{M}\boldsymbol{W}^T\boldsymbol{W}\right\|_F^2 &= \mathrm{tr}(\boldsymbol{W}^T\boldsymbol{W}\boldsymbol{M}\boldsymbol{W}^T\boldsymbol{W}\boldsymbol{W}^T\boldsymbol{W}\boldsymbol{M}\boldsymbol{W}^T\boldsymbol{W}) \\
&= \mathrm{tr}(\boldsymbol{W}\boldsymbol{W}^T\boldsymbol{W}\boldsymbol{M}\boldsymbol{W}^T\boldsymbol{W}\boldsymbol{W}^T\boldsymbol{W}\boldsymbol{M}\boldsymbol{W}^T).
\end{aligned} \tag{165}$$

From Lemma 9 we know that for all $\epsilon' > 0$, $\lim_{n\to\infty} \Pr(\left\|\boldsymbol{W}\boldsymbol{W}^T - I_c\right\| \geq \epsilon') = 0$. For notation simplicity, in this proof we will omit the limit and probability arguments which can be dealt with using union bound. Therefore, we will directly state $\left\|\boldsymbol{W}\boldsymbol{W}^T - I_c\right\| \leq \epsilon'$. From Kleinman & Athans (1968) we know that for positive semi-definite matrices $A$ and $B$ we have $\lambda_{\min}(A)\mathrm{tr}(B) \leq \mathrm{tr}(AB) \leq \lambda_{\max}(A)\mathrm{tr}(B)$, so

$$\begin{aligned}
&|\mathrm{tr}(\boldsymbol{W}\boldsymbol{W}^T \cdot \boldsymbol{W}\boldsymbol{M}\boldsymbol{W}^T\boldsymbol{W}\boldsymbol{W}^T\boldsymbol{W}\boldsymbol{M}\boldsymbol{W}^T) - \mathrm{tr}(\boldsymbol{W}\boldsymbol{M}\boldsymbol{W}^T\boldsymbol{W}\boldsymbol{W}^T\boldsymbol{W}\boldsymbol{M}\boldsymbol{W}^T)| \\
&\leq \max\{1 - \lambda_{\min}(\boldsymbol{W}\boldsymbol{W}^T), \lambda_{\max}(\boldsymbol{W}\boldsymbol{W}^T) - 1\}\mathrm{tr}(\boldsymbol{W}\boldsymbol{M}\boldsymbol{W}^T\boldsymbol{W}\boldsymbol{W}^T\boldsymbol{W}\boldsymbol{M}\boldsymbol{W}^T) \\
&\leq \left\|\boldsymbol{W}\boldsymbol{W}^T - I_c\right\| \mathrm{tr}(\boldsymbol{W}\boldsymbol{M}\boldsymbol{W}^T\boldsymbol{W}\boldsymbol{W}^T\boldsymbol{W}\boldsymbol{M}\boldsymbol{W}^T) \leq \epsilon'\mathrm{tr}(\boldsymbol{W}\boldsymbol{M}\boldsymbol{W}^T\boldsymbol{W}\boldsymbol{W}^T\boldsymbol{W}\boldsymbol{M}\boldsymbol{W}^T).
\end{aligned} \tag{166}$$

Similarly,

$$\begin{aligned}
&|\mathrm{tr}(\boldsymbol{W}\boldsymbol{M}\boldsymbol{W}^T\boldsymbol{W}\boldsymbol{W}^T\boldsymbol{W}\boldsymbol{M}\boldsymbol{W}^T) - \mathrm{tr}(\boldsymbol{W}\boldsymbol{M}\boldsymbol{W}^T\boldsymbol{W}\boldsymbol{M}\boldsymbol{W}^T)| \\
&= |\mathrm{tr}(\boldsymbol{W}\boldsymbol{W}^T \cdot \boldsymbol{W}\boldsymbol{M}\boldsymbol{W}^T\boldsymbol{W}\boldsymbol{M}\boldsymbol{W}^T) - \mathrm{tr}(\boldsymbol{W}\boldsymbol{M}\boldsymbol{W}^T\boldsymbol{W}\boldsymbol{M}\boldsymbol{W}^T)| \\
&\leq \left\|\boldsymbol{W}\boldsymbol{W}^T - I_c\right\| \mathrm{tr}(\boldsymbol{W}\boldsymbol{M}\boldsymbol{W}^T\boldsymbol{W}\boldsymbol{M}\boldsymbol{W}^T) \leq \epsilon'\mathrm{tr}(\boldsymbol{W}\boldsymbol{M}\boldsymbol{W}^T\boldsymbol{W}\boldsymbol{M}\boldsymbol{W}^T).
\end{aligned} \tag{167}$$

Therefore,

$$\begin{aligned}
&|\left\|\boldsymbol{W}^T\boldsymbol{W}\boldsymbol{M}\boldsymbol{W}^T\boldsymbol{W}\right\|_F^2 - \left\|\boldsymbol{W}\boldsymbol{M}\boldsymbol{W}^T\right\|_F^2| \\
&= |\mathrm{tr}(\boldsymbol{W}\boldsymbol{W}^T \cdot \boldsymbol{W}\boldsymbol{M}\boldsymbol{W}^T\boldsymbol{W}\boldsymbol{W}^T\boldsymbol{W}\boldsymbol{M}\boldsymbol{W}^T) - \mathrm{tr}(\boldsymbol{W}\boldsymbol{M}\boldsymbol{W}^T\boldsymbol{W}\boldsymbol{M}\boldsymbol{W}^T)| \\
&\leq |\mathrm{tr}(\boldsymbol{W}\boldsymbol{W}^T \cdot \boldsymbol{W}\boldsymbol{M}\boldsymbol{W}^T\boldsymbol{W}\boldsymbol{W}^T\boldsymbol{W}\boldsymbol{M}\boldsymbol{W}^T) - \mathrm{tr}(\boldsymbol{W}\boldsymbol{M}\boldsymbol{W}^T\boldsymbol{W}\boldsymbol{W}^T\boldsymbol{W}\boldsymbol{M}\boldsymbol{W}^T)| \\
&\quad + |\mathrm{tr}(\boldsymbol{W}\boldsymbol{M}\boldsymbol{W}^T\boldsymbol{W}\boldsymbol{W}^T\boldsymbol{W}\boldsymbol{M}\boldsymbol{W}^T) - \mathrm{tr}(\boldsymbol{W}\boldsymbol{M}\boldsymbol{W}^T\boldsymbol{W}\boldsymbol{M}\boldsymbol{W}^T)| \\
&\leq \epsilon'\mathrm{tr}(\boldsymbol{W}\boldsymbol{M}\boldsymbol{W}^T\boldsymbol{W}\boldsymbol{W}^T\boldsymbol{W}\boldsymbol{M}\boldsymbol{W}^T) + \epsilon'\mathrm{tr}(\boldsymbol{W}\boldsymbol{M}\boldsymbol{W}^T\boldsymbol{W}\boldsymbol{M}\boldsymbol{W}^T) \\
&\leq \epsilon'(1 + \epsilon')\mathrm{tr}(\boldsymbol{W}\boldsymbol{M}\boldsymbol{W}^T\boldsymbol{W}\boldsymbol{M}\boldsymbol{W}^T) + \epsilon'\mathrm{tr}(\boldsymbol{W}\boldsymbol{M}\boldsymbol{W}^T\boldsymbol{W}\boldsymbol{M}\boldsymbol{W}^T) \\
&\leq (2\epsilon' + (\epsilon')^2)\mathrm{tr}(\boldsymbol{W}\boldsymbol{M}\boldsymbol{W}^T\boldsymbol{W}\boldsymbol{M}\boldsymbol{W}^T) = (2\epsilon' + (\epsilon')^2)\left\|\boldsymbol{W}\boldsymbol{M}\boldsymbol{W}^T\right\|_F^2.
\end{aligned} \tag{168}$$

For all $\epsilon > 0$, select $\epsilon' < \min\{\frac{\sqrt{\epsilon}}{2}, \frac{\epsilon}{4}\}$, we have

$$|\left\|\boldsymbol{W}^T\boldsymbol{W}\boldsymbol{M}\boldsymbol{W}^T\boldsymbol{W}\right\|_F^2 - \left\|\boldsymbol{W}\boldsymbol{M}\boldsymbol{W}^T\right\|_F^2| < \epsilon\left\|\boldsymbol{W}\boldsymbol{M}\boldsymbol{W}^T\right\|_F^2. \tag{169}$$

In other words,

$$\lim_{n\to\infty} \frac{\left\|\boldsymbol{W}^T\boldsymbol{W}\boldsymbol{M}\boldsymbol{W}^T\boldsymbol{W}\right\|_F^2}{\left\|\boldsymbol{W}\boldsymbol{M}\boldsymbol{W}^T\right\|_F^2} = 1. \tag{170}$$

Hence we get

$$\lim_{n\to\infty} \frac{\left\|\boldsymbol{W}^T\boldsymbol{W}\boldsymbol{M}\boldsymbol{W}^T\boldsymbol{W}\right\|_F^2}{\|\boldsymbol{M}\|_F^2} = 1. \tag{171}$$

Next, consider the orthogonal projection matrix $P_{\boldsymbol{W}}$ that projects vectors in $R^n$ into the subspace spanned by all rows of $\boldsymbol{W}$. We will consider the matrix $P_{\boldsymbol{W}} M P_{\boldsymbol{W}}$. Define $\delta \triangleq \boldsymbol{W}^T \boldsymbol{W} - P_{\boldsymbol{W}}$, then from Lemma 10 we get $\|\delta\|_F^2 \le \epsilon'$. Therefore,

$$
\begin{aligned}
\left| \left\| \boldsymbol{W}^T \boldsymbol{W} M \boldsymbol{W}^T \boldsymbol{W} \right\|_F - \left\| P_{\boldsymbol{W}} M P_{\boldsymbol{W}} \right\|_F \right| &\le \|P_{\boldsymbol{W}} M \delta\|_F + \|\delta M P_{\boldsymbol{W}}\|_F + \|\delta M \delta\|_F \\
&\le \|M\|_F \left( 2 \|P_{\boldsymbol{W}}\|_F \|\delta\|_F + \|\delta\|_F^2 \right) \\
&\le \|M\|_F \left( 2 \cdot 4c^2 \epsilon' + (\epsilon')^2 \right) .
\end{aligned}
\tag{172}
$$

For all $\epsilon > 0$, we choose $\epsilon' < \min\{\frac{\sqrt{\epsilon}}{2}, \frac{\epsilon}{16c^2}\}$ and have

$$
\frac{\left| \left\| \boldsymbol{W}^T \boldsymbol{W} M \boldsymbol{W}^T \boldsymbol{W} \right\|_F - \left\| P_{\boldsymbol{W}} M P_{\boldsymbol{W}} \right\|_F \right|}{\|M\|_F} < \epsilon,
\tag{173}
$$

which means that

$$
\lim_{n\to\infty} \frac{\left| \left\| \boldsymbol{W}^T \boldsymbol{W} M \boldsymbol{W}^T \boldsymbol{W} \right\|_F - \left\| P_{\boldsymbol{W}} M P_{\boldsymbol{W}} \right\|_F \right|}{\left\| \boldsymbol{W}^T \boldsymbol{W} M \boldsymbol{W}^T \boldsymbol{W} \right\|_F} = \lim_{n\to\infty} \frac{\left| \left\| \boldsymbol{W}^T \boldsymbol{W} M \boldsymbol{W}^T \boldsymbol{W} \right\|_F - \left\| P_{\boldsymbol{W}} M P_{\boldsymbol{W}} \right\|_F \right|}{\|M\|_F} = 0.
\tag{174}
$$

Thus,

$$
\lim_{n\to\infty} \frac{\left\| P_{\boldsymbol{W}} M P_{\boldsymbol{W}} \right\|_F}{\|M\|_F} = \lim_{n\to\infty} \frac{\left\| P_{\boldsymbol{W}} M P_{\boldsymbol{W}} \right\|_F}{\left\| \boldsymbol{W}^T \boldsymbol{W} M \boldsymbol{W}^T \boldsymbol{W} \right\|_F} = 1.
\tag{175}
$$

Note that $\|M\|_F^2 = \left\| P_{\boldsymbol{W}} M P_{\boldsymbol{W}} \right\|_F^2 + \left\| P_{\boldsymbol{W}} M P_{\boldsymbol{W}}^\perp \right\|_F^2 + \left\| P_{\boldsymbol{W}}^\perp M P_{\boldsymbol{W}} \right\|_F^2 + \left\| P_{\boldsymbol{W}}^\perp M P_{\boldsymbol{W}}^\perp \right\|_F^2$. Therefore,

$$
\lim_{n\to\infty} \frac{\left\| P_{\boldsymbol{W}} M P_{\boldsymbol{W}}^\perp \right\|_F^2 + \left\| P_{\boldsymbol{W}}^\perp M P_{\boldsymbol{W}} \right\|_F^2 + \left\| P_{\boldsymbol{W}}^\perp M P_{\boldsymbol{W}}^\perp \right\|_F^2}{\|M\|_F^2} = \lim_{n\to\infty} \frac{\|M\|_F^2 - \left\| P_{\boldsymbol{W}} M P_{\boldsymbol{W}} \right\|_F^2}{\|M\|_F^2} = 0.
\tag{176}
$$

In other words,

$$
\lim_{n\to\infty} \frac{\left\| P_{\boldsymbol{W}} M P_{\boldsymbol{W}}^\perp \right\|_F}{\|M\|_F} = \lim_{n\to\infty} \frac{\left\| P_{\boldsymbol{W}}^\perp M P_{\boldsymbol{W}} \right\|_F}{\|M\|_F} = \lim_{n\to\infty} \frac{\left\| P_{\boldsymbol{W}}^\perp M P_{\boldsymbol{W}}^\perp \right\|_F}{\|M\|_F} = 0.
\tag{177}
$$

From Lemma 18 we know that $\lim_{n\to\infty} \|M\|_F$ (this hasn't proved to be exist, so we perhaps need to say "for large enough $n$") is lower bounded by some constant that is independent of $n$, so

$$
\lim_{n\to\infty} \left\| P_{\boldsymbol{W}} M P_{\boldsymbol{W}}^\perp \right\|_F = \lim_{n\to\infty} \left\| P_{\boldsymbol{W}}^\perp M P_{\boldsymbol{W}} \right\|_F = \lim_{n\to\infty} \left\| P_{\boldsymbol{W}}^\perp M P_{\boldsymbol{W}}^\perp \right\|_F = 0.
\tag{178}
$$

Note that

$$
M = P_{\boldsymbol{W}} M P_{\boldsymbol{W}} + P_{\boldsymbol{W}} M P_{\boldsymbol{W}}^\perp + P_{\boldsymbol{W}}^\perp M P_{\boldsymbol{W}} + P_{\boldsymbol{W}}^\perp M P_{\boldsymbol{W}}^\perp.
\tag{179}
$$

Thus,

$$
\lim_{n\to\infty} \left\| M - P_{\boldsymbol{W}} M P_{\boldsymbol{W}} \right\|_F = 0.
\tag{180}
$$

For any $\epsilon > 0$, set $\delta < \min\{\frac{\epsilon\gamma}{8c^2}, \frac{\sqrt{\epsilon\gamma}}{2c}]\}$, from Lemma 10, we know that with probability 1, $\left\| P_{\boldsymbol{W}} - \boldsymbol{W}^T \boldsymbol{W} \right\|_F \le \delta$. Therefore,

$$
\begin{aligned}
&\left\| P_{\boldsymbol{W}} M P_{\boldsymbol{W}} - \boldsymbol{W}^T \boldsymbol{W} M \boldsymbol{W}^T \boldsymbol{W} \right\|_F \\
\le& \left\| P_{\boldsymbol{W}} - \boldsymbol{W}^T \boldsymbol{W} \right\|_F^2 \|M\|_F + 2 \left\| P_{\boldsymbol{W}} - \boldsymbol{W}^T \boldsymbol{W} \right\|_F \|M\|_F \|P_{\boldsymbol{W}}\|_F \\
\le& \delta^2 \cdot 2c^2 + 2\delta \cdot 2c^2 \\
<& \epsilon.
\end{aligned}
\tag{181}
$$

In other words,

$$
\lim_{n\to\infty} \left\| P_{\boldsymbol{W}} M P_{\boldsymbol{W}} - \boldsymbol{W}^T \boldsymbol{W} M \boldsymbol{W}^T \boldsymbol{W} \right\|_F = 0.
\tag{182}
$$

Now we conclude that

$$
\lim_{n\to\infty} \left\| M - \boldsymbol{W}^T \boldsymbol{W} M \boldsymbol{W}^T \boldsymbol{W} \right\|_F = 0.
\tag{183}
$$

From Lemma 16 we know that $\forall i, j \in [c], \lim_{n \to \infty}((\boldsymbol{WMW}^T)_{i,j} - (\boldsymbol{WM}^*\boldsymbol{W}^T)_{i,j}) = 0$, i.e.,

$$\lim_{n \to \infty} \left\| \boldsymbol{WMW}^T - \boldsymbol{WM}^*\boldsymbol{W}^T \right\|_F = 0. \tag{184}$$

Since

$$\left\| \boldsymbol{W}^T\boldsymbol{WMW}^T\boldsymbol{W} - \boldsymbol{W}^T\boldsymbol{WM}^*\boldsymbol{W}^T\boldsymbol{W} \right\|_F \le \|\boldsymbol{W}\|_F^2 \left\| \boldsymbol{WMW}^T - \boldsymbol{WM}^*\boldsymbol{W}^T \right\|_F, \tag{185}$$

from Lemma 8 which bounds the Frobenius norm of $\boldsymbol{W}$ we know that

$$\lim_{n \to \infty} \left\| \boldsymbol{W}^T\boldsymbol{WMW}^T\boldsymbol{W} - \boldsymbol{W}^T\boldsymbol{WM}^*\boldsymbol{W}^T\boldsymbol{W} \right\|_F = 0. \tag{186}$$

Thus,

$$\lim_{n \to \infty} \left\| \boldsymbol{M} - \boldsymbol{W}^T\boldsymbol{WM}^*\boldsymbol{W}^T\boldsymbol{W} \right\|_F = 0. \tag{187}$$

Note that $\boldsymbol{M}^* = \frac{1}{4}\left(\mathbb{E}[\boldsymbol{W}^T\boldsymbol{AW}] + \mathrm{diag}(\mathbb{E}[\boldsymbol{W}^T\boldsymbol{AW}])\right)$, so

$$4\boldsymbol{W}^T\boldsymbol{WM}^*\boldsymbol{W}^T\boldsymbol{W} = \boldsymbol{W}^T\boldsymbol{WW}^T\tilde{\boldsymbol{A}}\boldsymbol{WW}^T\boldsymbol{W} + \boldsymbol{W}^T\boldsymbol{W}\mathrm{diag}(\mathbb{E}[\boldsymbol{W}^T\boldsymbol{AW}])\boldsymbol{W}^T\boldsymbol{W}. \tag{188}$$

We will first analyze the second term on the RHS of equation Eq. (188). $\forall \epsilon > 0$, set $\epsilon' = \frac{\epsilon}{\sqrt{c}}$, and from Lemma 9 we know that $\left\| \boldsymbol{WW}^T - I_c \right\| < \epsilon'$ with probability 1, which means that $\left| \left\| \boldsymbol{WW}^T \right\|_F - c \right| < \epsilon$ with probability 1. Set $\epsilon = c$, we know that $\left\| \boldsymbol{WW}^T \right\|_F < 2c$ with probability 1. Note that

$$\begin{aligned}
\left\| \boldsymbol{W}^T\boldsymbol{W}\mathrm{diag}(\mathbb{E}[\boldsymbol{W}^T\boldsymbol{AW}])\boldsymbol{W}^T\boldsymbol{W} \right\|_F &\le \left\| \boldsymbol{W}^T\boldsymbol{W} \right\|_F^2 \left\| \mathrm{diag}(\mathbb{E}[\boldsymbol{W}^T\boldsymbol{AW}]) \right\|_F \\
&= \left\| \boldsymbol{WW}^T \right\|_F^2 \left\| \mathrm{diag}(\mathbb{E}[\boldsymbol{W}^T\boldsymbol{AW}]) \right\|_F \\
&\le 4c^2 \left\| \mathrm{diag}(\mathbb{E}[\boldsymbol{W}^T\boldsymbol{AW}]) \right\|_F.
\end{aligned} \tag{189}$$

Combine this with equation Eq. (154) and we have

$$\lim_{n \to \infty} \frac{\left\| \boldsymbol{W}^T\boldsymbol{W}\mathrm{diag}(\mathbb{E}[\boldsymbol{W}^T\boldsymbol{AW}])\boldsymbol{W}^T\boldsymbol{W} \right\|_F}{\left\| \boldsymbol{W}^T\tilde{\boldsymbol{A}}\boldsymbol{W} \right\|_F} = 0. \tag{190}$$

From equation Eq. (162) we know that $\left\| \boldsymbol{W}^T\tilde{\boldsymbol{A}}\boldsymbol{W} \right\|_F \ge \frac{\gamma}{4}$ with probability 1, so

$$\lim_{n \to \infty} \left\| 4\boldsymbol{W}^T\boldsymbol{WM}^*\boldsymbol{W}^T\boldsymbol{W} - \boldsymbol{W}^T\boldsymbol{WW}^T\tilde{\boldsymbol{A}}\boldsymbol{WW}^T\boldsymbol{W} \right\|_F = 0. \tag{191}$$

Similarly, define $\delta \triangleq \boldsymbol{WW}^T - I_c$, then

$$\begin{aligned}
& \left\| \boldsymbol{W}^T\boldsymbol{WW}^T\tilde{\boldsymbol{A}}\boldsymbol{WW}^T\boldsymbol{W} - \boldsymbol{W}^T\tilde{\boldsymbol{A}}\boldsymbol{W} \right\|_F \\
\le\ & \left\| \boldsymbol{W}^T\delta\tilde{\boldsymbol{A}}\delta\boldsymbol{W} \right\|_F + 2\left\| \boldsymbol{W}^T\tilde{\boldsymbol{A}}\delta \right\|_F \\
\le\ & \|\boldsymbol{W}\|_F^2 \|\delta\|_F^2 \left\| \tilde{\boldsymbol{A}} \right\|_F + 2\|\boldsymbol{W}\|_F \|\delta\|_F \left\| \tilde{\boldsymbol{A}} \right\|_F.
\end{aligned} \tag{192}$$

Set $\epsilon' < \min\{\frac{\epsilon}{8c^2}, \sqrt{\frac{\epsilon}{8c^3}}\}$, then from Lemma 9 we know that $\|\delta\|_F < \epsilon'$ with probability 1, and from Lemma 8 we have $\|\boldsymbol{W}\|_F \le 2c$ with probability 1. We also have $\left\| \tilde{\boldsymbol{A}} \right\|_F \le c$ since each entry of $\boldsymbol{A}$ is bounded by 1 in absolute value. Therefore,

$$\left\| \boldsymbol{W}^T\boldsymbol{WW}^T\tilde{\boldsymbol{A}}\boldsymbol{WW}^T\boldsymbol{W} - \boldsymbol{W}^T\tilde{\boldsymbol{A}}\boldsymbol{W} \right\|_F \le 4c^2(\epsilon')^2 \cdot c + 2 \cdot 2c\epsilon' \cdot c < \frac{\epsilon}{2} + \frac{\epsilon}{2} = \epsilon, \tag{193}$$

which means that

$$\lim_{n \to \infty} \left\| \boldsymbol{W}^T\boldsymbol{WW}^T\tilde{\boldsymbol{A}}\boldsymbol{WW}^T\boldsymbol{W} - \boldsymbol{W}^T\tilde{\boldsymbol{A}}\boldsymbol{W} \right\|_F = 0. \tag{194}$$

From equations Eq. (187), Eq. (191), and Eq. (194) we get

$$\lim_{n \to \infty} \left\| \boldsymbol{M} - \frac{1}{4}\boldsymbol{W}^T\tilde{\boldsymbol{A}}\boldsymbol{W} \right\|_F = 0. \tag{195}$$

Besides, from equation Eq. (95) in Lemma 10 we know that for any $\epsilon' > 0$,

$$\left\| \overline{W} - W \right\|_F^2 = \sum_{i \in [c]} \left\| \overline{W}_i - W_i \right\|^2 < \epsilon', \tag{196}$$

where $\overline{W}$ is the orthogonal version of $W$, i.e., we run the Gram-Schmidt process for the rows of $W$. Define $\delta \triangleq \overline{W} - W$, for any $\epsilon > 0$, set $\epsilon' = \min\{\frac{\epsilon}{8c^2}, \sqrt{\frac{\epsilon}{2c}}\}$, we have with probability 1,

$$\left\| W^T \tilde{\mathbf{A}} W - \overline{W}^T \tilde{\mathbf{A}} \overline{W} \right\|_F \leq 2 \left\| \delta \right\|_F \left\| \tilde{\mathbf{A}} \right\|_F \left\| W \right\|_F + \left\| \delta \right\|_F^2 \left\| \tilde{\mathbf{A}} \right\|_F \tag{197}$$
$$4c^2 \epsilon' + c(\epsilon')^2 < \epsilon.$$

Therefore,

$$\lim_{n \to \infty} \left\| W^T \tilde{\mathbf{A}} W - \overline{W}^T \tilde{\mathbf{A}} \overline{W} \right\|_F = 0, \tag{198}$$

which implies

$$\lim_{n \to \infty} \left\| M - \frac{1}{4} \overline{W}^T \tilde{\mathbf{A}} \overline{W} \right\|_F = 0. \tag{199}$$

From Lemma 3 we know that with probability 1, $\tilde{\mathbf{A}}$ is of rank $(c-1)$. Since $\mathbf{A} \cdot \mathbf{1} = 0$ is always true, the top $(c-1)$ eigenspace of $\tilde{\mathbf{A}}$ is $\mathbb{R}^c \backslash \{\mathbf{1}\}$. Note that the rows in $\overline{W}$ are of unit norm and orthogonal to each other, we conclude that $\overline{W}^T \tilde{\mathbf{A}} \overline{W}$ is of rank $(c-1)$ and the corresponding eigenspace is $\mathcal{R}\{\overline{W}_i\}_{i=1}^c \backslash \{\overline{W} \cdot \mathbf{1}\}$. Moreover, the minimum positive eigenvalue of $\overline{W}^T \tilde{\mathbf{A}} \overline{W}$ is lower bounded by $\gamma$.

As for the top $c-1$ eigenvectors of $M$, define $\delta \triangleq M - \frac{1}{4} \overline{W}^T \tilde{\mathbf{A}} \overline{W}$, then $M = \frac{1}{4} \overline{W}^T \tilde{\mathbf{A}} \overline{W} + \delta$. Define $S_1$ as the top $c-1$ eigenspaces for $M$, and $S_2$ to be the top $c-1$ eigenspaces for $\frac{1}{4} \overline{W}^T \tilde{\mathbf{A}} \overline{W}$. Then from Davis-Kahan Theorem we know that

$$\left\| \sin \Theta(S_1, S_2) \right\|_F \leq \frac{\|\delta\|_F}{\lambda_{c-1}(\frac{1}{4} \overline{W}^T \tilde{\mathbf{A}} \overline{W})}. \tag{200}$$

Here $\Theta(S_1, S_2)$ is a $(c-1) \times (c-1)$ diagonal matrix whose $i$-th diagonal entry is the $i$-th canonical angle between $S_1$ and $S_2$. Since $\lim_{n \to \infty} \|\delta\|_F = 0$, and with probability 1, $\lambda_{c-1}(\frac{1}{4} \overline{W}^T \tilde{\mathbf{A}} \overline{W}) \geq \gamma$ which is independent of $n$, we have with probability 1,

$$\lim_{n \to \infty} \left\| \sin \Theta(S_1, S_2) \right\|_F = 0, \tag{201}$$

which indicates that the top $c-1$ eigenspaces for $M$ and $\frac{1}{4} \overline{W}^T \tilde{\mathbf{A}} \overline{W}$ are the same when $n \to \infty$.

Notice that the top $c-1$ eigenspaces of $\overline{W}^T \tilde{\mathbf{A}} \overline{W}$ are $\mathcal{R}\{\overline{W}_i\}_{i=1}^c \backslash \{\overline{W} \cdot \mathbf{1}\}$, so $M$ will also have the same top $c-1$ eigenspaces. Besides, from equation Eq. (95) we know that $\lim_{n \to \infty} \left\| W - \overline{W} \right\|_F = 0$, so $\mathcal{R}\{\overline{W}_i\}_{i=1}^c \backslash \{\overline{W} \cdot \mathbf{1}\}$ are the same as $\mathcal{R}\{W_i\}_{i=1}^c \backslash \{W \cdot \mathbf{1}\}$. This completes the proof of this theorem. $\qquad\square$

### H.4 Experiment Results

Table 10: Overlap of $\mathcal{R}(S^{(k)}) \backslash \{S^{(k)} \cdot \mathbf{1}\}$ and the top $c-1$ dimension eigenspace of $\mathbb{E}[M^{(k)}]$ of different layers at minima.

| Dataset | MNIST | | MNIST-R | | CIFAR10 | | CIFAR10-R | |
| Network | F-$1500^3$ | LeNet5 | F-$1500^3$ | LeNet5 | F-$1500^3$ | LeNet5 | F-$1500^3$ | LeNet5 |
|---|---|---|---|---|---|---|---|---|
| fc1 | 0.602 | 0.890 | 0.235 | 0.518 | 0.880 | 0.951 | 0.903 | 0.213 |
| fc2 | 0.967 | 0.931 | 0.801 | 0.912 | 0.943 | 0.972 | 0.931 | 0.701 |
| fc3 | 0.982 | 0.999 | 0.998 | 0.999 | 0.993 | 0.999 | 0.996 | 0.999 |

Note that the overlap can be low for random-label datasets which do not have a clear eigengap (as in Fig. 4). Understanding how the data could change the behavior of the Hessian is an interesting open problem. Other papers have given alternative explanations which are not directly comparable to ours, however ours is the only one that gives a closed-form formula for top eigenspace. In Appendix F.1 we will discuss the other explanations in more details.

