# OpenReview forum: "Dissecting Hessian: Understanding Common Structure of Hessian in Neural Networks"
_ICLR.cc/2021/Conference — Reject_

### Official Review · AnonReviewer1 · 2020-10-28
**Interesting observations, but the paper lacks experiment interpretation and conclusions.**

**Rating:** 4
**Confidence:** 4

**Review:**

Summary:
The paper studies the structure of the Hessian matrix of loss functions by approximating Hessians using Kronecker factorizations. Combining the Kronecker factorization with PAC-Bayes, the authors provide a tighter bound on classification error.

Pros:
- The paper contains many experiments.
- Interesting observations on the models trained with batch normalization.
- The improvement of classification error bound.

Concerns:
- I suggest adding the Conclusion section. This section should highlight the contributions and summarize the study, and the paper will certainly benefit from it.
- The main paper contribution, optimized PAC-Bayes bound, is not emphasized enough; many important details are moved into the Appendix section, while most part of the paper is devoted to eigenspace overlap. While eigenspace overlap looks quite interesting, it does not give many insights. One of the most interesting properties of the Hessian structure is the gap in eigenvalues distribution around the number of classes, but this is prior work.
- The figure captions are not detailed enough; many figures lack axis labeling. For example, it is not apparent to me what is illustrated in Figure 3. What do the axes depict? Similarly, for Figure 4.
- The key concern is an incremental contribution comparing to Martens & Grosse (2015). While Martens & Grosse (2015) use the Kronecker factorization for training acceleration, your paper focuses more on eigenspace overlap, which is supported by many rigorous empirical results and illustrations, but not on the possible interpretations and insights.

UPD:
Thank you for your answer and for addressing the points raised in the review. Still, I agree with the concerns raised by AnonReviewer4 on the small scale of the networks used in the experiments (e.g., LeNet, fc). I am still not convinced by the experiments and the excess of hardly understandable plots given to support the main bulk of the paper's claims. Therefore, I leave my rating the same.

---

> ### Author Response · Authors · 2020-11-21
> **Response to Reviewer 1 with our interpretations and insights on the structures of Hessian eigenspace**
>
> Dear reviewer, thank you for your time and comment, and we highly appreciate your constructive advice.
>
> Following you suggestions, we have updated the paper such that it now includes a conclusion section. We have also updated the introduction section, which, together with the newly added conclusion section, summaries and clarifies our main contribution. In addition, we have highlighted our main contributions  and potential insights in the general response. Hopefully these will help you understand our contributions and interpretations better.
>
> We also refined the caption of several figures, and rewrote **Section 4.2** where we define and explain the eigenvector correspondence matrices in **Fig.3** in a more comprehensive way. Hope that will resolve some of the confusions. For the PAC-Bayes part, we move a figure showing the posterior variance to the main text, which could make our contribution clearer.
>
> We believe the structures of Hessian eigenspace are important and our explanations and interpretations can provide potential insights on this field. We will explain our point of views in details below.
>
> ### 1. Why do we think eigenspace overlap is interesting?
>
> Disregarding the peak dimension, we find the high eigenspace overlap between the models trained with different initialization is already very non-trivial. Because one would usually expect that in such a  highly non-convex regime, parameter-wise, everything about two differently initialized models would be different.
>
> Moreover, while there has been a rich literature investigating the gap of Hessian eigenspectrum at the number of classes, which has been considered as the "magic number" for neural network Hessian analysis, our work is the first to show that information of the network structure itself (such as the number neurons / channels for each layer) are also embeded in the Hessian matrix, and these information appears as the peak dimension of a non-trivial eigenspace overlap.
>
> The eigenspace overlap phenomenon shows that there is a subspace with large eigenvalues that is preserved across different models and throughout the training trajectory. Understanding the structure of such subspace likely has applications in both generalization and optimization.
>
> ### 2. Potential insights on the study of eigenspace and eigenspace overlap.
>
> Existing literature already uses Hessian information to understand both generalization and optimization, as we discuss below. We believe the new structures we find may also be useful in these discussions.
>
> * Generalization: It is a well known empirical result that the sharpness of minima is well correlated with its generalization capability[1][2]. Since the sharpest directions are just the basis for the dominating eigenspace of the Hessian. From a theoretical perspective, understanding the structure of this low dimensional subspace would provide valuable insight to the analysis on minima sharpness. (e.g. batch normalization is suppressing the $\mathbb{E}[\mathbf{xx}^T]$ term)
>
> * Optimization: There has been literature showing that the dominating eigenspace has a high overlap with the optimization trajectory of gradient descent [3]. The similarity of eigenspaces of different models may also imply the similarity between different optimization trajectories within some structured subspace. For optimization it is potentially easier to leverage the eigenspace overlap structure as it shows that the subspace is preserved across different models.
>
> ### 3. Regarding the incremental contributions on KFAC
> While KFAC [4] first utilized the idea of performing Kronecker factorization on second order information, the validity of such approximation was mostly validated by its applications in optimization. Second order optimization algorithms often uses the inverse of the Fisher matrix or Hessian matrix, which focuses on the lower end of the spectrum. On the other hand, our work focuses on the analysis of dominating eigenspace of the Hessian, therefore we give more empirical evidences that the factorization preserves the dominating eigenspace.
>
> **References**
>
> [1] Keskar et.al. On large-batch training for deep learning: Generalization gap and sharp minima. ICLR 2017. https://arxiv.org/abs/1609.04836
>
> [2] Jiang et.al. Fantastic generalization measures and where to find them. ICLR 2020.
> https://arxiv.org/abs/1912.02178
>
> [3] Gur-Ari et.al. Gradient descent happens in a tiny subspace. 2018. https://arxiv.org/abs/1812.04754
>
> [4] Martens & Grosse. Optimizing Neural Networks with Kronecker-factored Approximate Curvature. ICML 2015. https://arxiv.org/abs/1503.05671

---

### Official Review · AnonReviewer3 · 2020-10-28
**A review**

**Rating:** 7
**Confidence:** 4

**Review:**

### 1. Content summary
The authors study the layer-wise Hessian of several well-known vision networks on several vision classification datasets. They, to me be very surprisingly, observe that the subspaces formed by top eigenvectors of those layer-wise Hessians have non-trivial overlap between independently initialized and trained models. They explain this by proposing the usage of Kronecker factorization, whose validity they experimentally show. They trace the surprising top eigenspace overlap to the structure of the data and show that it weakens with the application of batch norm. They also show that this overlap can tighten PAC-Bayes bounds.

### 2. Strong points
* The paper is well written and motivated. I went from the intro to the conclusion in one go and felt I didn't have to jump around to fish out definitions etc.
* The question is well motivated and interesting.
* The proposed simplifications (Kronecker factorization) are validated on real networks.
* The networks and data used are non-trivial and close to practice, which makes this paper better than many others in bridging the gap between theory and experiments.
* The story fits nicely with existing observations about the effect of batch norm on class-specific structure of the Hessian.
* The obvious application to PAC-Bayes is nice!

### 3. Weaker points
* The paper would be strengthened by a broader suite of experiments: add ResNets, WideResNets, let's see what skip connections do, try CIFAR-100 to see the effect of >10 classes. I am not suggesting you work on ImageNet, but more experiments of even the easier variety would make the paper even more convincing. Perhaps FASHION MNIST and SVHN might be quick enough to run?
* I am missing a summary table or plot showing how well the approximations proposed match the reality for different architectures and datasets. This is a small detail, but I kept wondering how valid many things were. I was convinced by your detailed figures (e.g. Figure 3) and I am sure a figure like  that for each combination of net x architecture would clutter the paper, but a summary plot showing the match between the assumptions and reality would be very nice and convincing.

### 4. Points to clarify
* In Figure 4 the scales for the E[M] and H_L are different, the difference presumable coming from the scale of E[xx^T]. Is that the case?
* Am I understanding it correctly that the top eigenvector overlap is driven mainly by the structure of the data that through E[xx^T] \approx E[x]E[x]^T becomes almost rank 1?

### 5. Potential papers to look at
Stiffness: A New Perspective on Generalization in Neural Networks by Stanislav Fort, Paweł Krzysztof Nowak, Stanislaw Jastrzebski, Srini Narayanan (https://arxiv.org/abs/1901.09491) looks at the gradient tensorproduct gradient structure which is a part of the Hessian and its class dependence and training time dependence. It might be relevant.

### 6. Summary
This is a good, well-written and motivated paper with a (for me) surprising observation and well-formed explanation for it. More experiments would make it even stronger, but I already like it as it is!

---

> ### Author Response · Authors · 2020-11-21
> **Response to Reviewer 3 with our clarifications**
>
> Dear reviewer, thank you for your time and positive comment. We are very glad that you find our work valuable. We highly appreciate your constructive advices.
>
> For the two points for clarification, your understanding are all correct and accurate. We also modified the paper such that hopefully future readers will not have similar confusions. We also added the paper you mentioned relating to gradient alignment in the related works section as well as **Appendix E**, where we discussed about gradient clustering and structure of the output Hessian matrix $\mathbb{E}[\mathbf{M}]$.
>
> While we believe our current experiments are enough for showing the phenomena and verifying our explanations, we agree that a broader suite of experiments would potentially strengthen our paper. We will conduct more experiments in the future. We are currently planning to expand the experiment to larger networks and larger datasets. We plan to start by FASHION MNIST and the 20 super-classes of CIFAR100. However since our main observation, which is the non-trivial eigenspace overlap and position of its peak is related to the structure of network instead of the dataset information, we expect the experiment outcome will not be surprising.
>
> Regarding your question on how well the approximations are for different neural networks and datasets, we would say that the approximation is enough for our explanations for all the networks and datasets we experimented on. For a few instances where the approximation is not good, we also observe the disappearance or weakening of the phenomena we explained. Therefore, the approximation agrees with our explanations. The additional explanations in **Appendix E** give explanations for a more general situation, and can be helpful in answering your question.
>
> In addition, we have shown our experiment results on a variation of neural networks and datasets in **Appendix D**. We also listed all the neural networks and datasets we experimented in **Appendix B**. We agree that a summary table can be helpful. However, due to the page limit, we cannot add that into the main text. We believe what we show in the Appendix is clear and detailed.

---

### Official Review · AnonReviewer4 · 2020-10-29
**Attractive Narrative. Needs to better reflect prior research contributions**

**Rating:** 4
**Confidence:** 5

**Review:**

1.	I want to congratulate the authors on producing an attractive narrative, with a very nice flow to it and a very clear heuristic content.  I believe if I were seeing these ideas for the first time, I would definitely find this presentation friendlier and clearer than if I learned it from some other presentations.
2.	The topic is important. The analysis of the spectrum of the Hessian of the loss function gives us significant insight, and it is rather amazing and beautiful that the spectrum has definite mathematical properties which also have mathematical explanation. The narrative excels in making us feel this as something clear and natural.
3.	This is my first time ever reviewing for conferences of this nature; I am a bit unclear about the level of contribution we are looking for. My understanding is that researchers in this field are writing numerous very short papers each year and a paper doesn’t need to represent more than an incremental bump over previous work.
4.	One reason I bid to review this Mss. is that the topic is not new to me. In the last few years three separate researchers at my institution have written papers on this general topic; and so I feel more conversant with this topic than some other material I might have been assigned. So …
5.	I feel that this work may not know about or may not have fully assimilated the contributions by other authors. I can mention these papers:

arXiv:1901.10159
An Investigation into Neural Net Optimization via Hessian Eigenvalue Density
	Authors: Behrooz Ghorbani, Shankar Krishnan, Ying Xiao
Abstract: To understand the dynamics of optimization in deep neural networks, we develop a tool to study the evolution of the entire Hessian spectrum throughout the optimization process. Using this, we study a number of hypotheses concerning smoothness, curvature, and sharpness in the deep learning literature. We then thoroughly analyze a crucial structural feature of the spectra: in non-batch normalized networks, we observe the rapid appearance of large isolated eigenvalues in the spectrum, along with a surprising concentration of the gradient in the corresponding eigenspaces. In batch normalized networks, these two effects are almost absent. We characterize these effects, and explain how they affect optimization speed through both theory and experiments.


So arXiv:1901.10159 clearly calls out the phenomenon of batch norm changing the hessian. In comparison, the paper under review gives a heuristically very clear explanation of why such an effect should be present, i.e. why batch normalization should be effective at making the objective easier to optimize.  However, the paper under review does not fully discuss the above research contribution of arXiv:1901.10159,  and it would be very easy for a reader of this paper to imagine that the phenomenon being presented here originates with this paper.   I should point out that the authors of arXiv:1901.10159 have made company-wide presentations about that work. Since the company is Google, that means that their work is fairly well known.


arXiv:1811.07062
The Full Spectrum of Deepnet Hessians at Scale: Dynamics with SGD Training and Sample Size
Authors: Vardan Papyan
Abstract: We apply state-of-the-art tools in modern high-dimensional numerical linear algebra to approximate efficiently the spectrum of the Hessian of modern deepnets, with tens of millions of parameters, trained on real data. Our results corroborate previous findings, based on small-scale networks, that the Hessian exhibits "spiked" behavior, with several outliers isolated from a continuous bulk. We decompose the Hessian into different components and study the dynamics with training and sample size of each term individually

To my knowledge arXiv:1811.07062  is the first paper to show the empirical deep learning community that one can compute the eigenvalue density spectra of modern deepnet classifiers eg those of the kind that are in use on large datasets like imagenet and in real applications. The result of having the technology in arXiv:1811.07062  is that the author is able to show that some of the key features of eigenvalue spectra that were seen in small scale situations are also present at full scale. The examples shown in the paper under review, in contrast, are of quite limited scale, and to my understanding do not represent the current state of the art. Consequently, although tools re available to test out the authors’ ideas on realistic problems, we are left wondering what seen in these small examples might generalize.

arXiv:1901.08244 Measurements of Three-Level Hierarchical Structure in the Outliers in the Spectrum of Deepnet Hessians
Authors: Vardan Papyan
Abstract: We consider deep classifying neural networks. We expose a structure in the derivative of the logits with respect to the parameters of the model, which is used to explain the existence of outliers in the spectrum of the Hessian. Previous works decomposed the Hessian into two components, attributing the outliers to one of them, the so-called Covariance of gradients. We show this term is not a Covariance but a second moment matrix, i.e., it is influenced by means of gradients. These means possess an additive two-way structure that is the source of the outliers in the spectrum. This structure can be used to approximate the principal subspace of the Hessian using certain "averaging" operations, avoiding the need for high-dimensional eigenanalysis. We corroborate this claim across different datasets, architectures and sample sizes

To my knowledge arXiv:1901.08244 goes over some of the same territory as the present manuscript, but at full scale and in a much more penetrating way. arXiv:1901.08244  shows that not only do the C class means influence the eigenvalues, but actually there are C(C-1) eigenvalues that have structure deriving from means. The examples shown in the paper under review, in contrast, are of quite limited scale, and in contrast don’t seem able to show the full structure which we now know to be present in deepnet spectra across a very wide range of networks and datasets at full scale.  However, the paper under review does not discuss the research contribution of arXiv:1901.08244,  and in consequence a reader could get the misleading impression that this paper’s heuristic explanations of the mean structure are the only knowledge we have about this phenomenon, when we have actually dramatically more information.


arXiv:2008.11865  Traces of Class/Cross-Class Structure Pervade Deep Learning Spectra
Authors: Vardan Papyan
Abstract: Numerous researchers recently applied empirical spectral analysis to the study of modern deep learning classifiers. We identify and discuss an important formal class/cross-class structure and show how it lies at the origin of the many visually striking features observed in deepnet spectra, some of which were reported in recent articles, others are unveiled here for the first time. These include spectral outliers, "spikes", and small but distinct continuous distributions, "bumps", often seen beyond the edge of a "main bulk". The significance of the cross-class structure is illustrated in three ways: (i) we prove the ratio of outliers to bulk in the spectrum of the Fisher information matrix is predictive of misclassification, in the context of multinomial logistic regression; (ii) we demonstrate how, gradually with depth, a network is able to separate class-distinctive information from class variability, all while orthogonalizing the class-distinctive information; and (iii) we propose a correction to KFAC, a well-known second-order optimization algorithm for training deepnets.

To my knowledge arXiv:2008.11865  again goes over some of the same territory as the present manuscript, but at full scale and in a much more penetrating way. arXiv:2008.11865  in particular discusses the KFAC approximation of Martens and Grosse and in fact claims, with substantial evidence gleaned from many realistic examples, that KFAC is not a good approximation. The paper under review doesn’t cite arXiv:2008.11865, but does make heuristic claims about the adequacy of the KFAC approximation.  However, the evidence presented is much weaker and the implications much more sketchy than what is discussed at greater length  in arXiv:2008.11865.

---

> ### Author Response · Authors · 2020-11-21
> **Response to Reviewer 4 with the actual connections and differences between our work and previous works mentioned**
>
> Dear reviewer, thank you for your time and comment. We highly appreciate your constructive advice. We have edited the paper based on some of your advice and we will state them below.
>
> However, we would like to first point out that your review **ignores most of our contributions and only discussed the Hessian eigenspectrum**. Actually, the main contribution of our paper is the explanation of two surprising structures in the top eigenspace of layer-wise Hessians. The full list of our contributions are stated in the general response and our conclusion section in the revised version. We discuss the Hessian eigenspectrum because our explanations for the eigenspace structures can also be applied to the eigenspectrum. Since Hessian eigenspectrum is not our main focus, it is unfair to compare our work with these previous works which focuses on the eigenspectrum but not the eigenspace.
>
> Of course, we agree that the previous works mentioned are related to our paper. We cited [1][2][3] in our original paper and added [4] into the revised version. We would like to thank you for mentioning these papers and sharing your thoughts on the relation between our work and their works. However, we respectfully disagree with many of these comments. We think you might misinterpreted our paper and some parts of these previous works. We will explain our reasoning below.
>
> Contrary to what you might assume, we did not claim the phenomena on Hessian eigenspectrum as our discovery. We are sorry for this confusion and we have made our acknowledgement on previous works clearer in our revised paper. For the methods and explanations, however, we believe there are clear distinctions between our work and the previous works you mentioned.
>
> Below, we will share our thoughts on the relation between our work and each previous work mentioned in this review.
>
> 1. We agree with you that [1] presented the phenomenon that Batch Normalization suppresses the outliers in the Hessian eigenspectrum. We would like to thank you for making this comment. We have made it clear in our revised version. We would also like to note that we provided an explanation for this phenomenon but [1] did not.
>
> 2. Contrary to what you suggest, we do not think the technology presented in [2] can be applied to the experiments in our paper. This is because our experiments requires Hessian eigenvectors, but this technology cannot calculate eigenvectors. We use Hessian Eigenthings package [5] which is more suitable for our calculations. The only relation between [2] and our work is that [2] shows the existence of outliers in Hessian eigenspecturm, and we have clearly acknowledged that in our paper.
>
> 3. [3] did provide an inspiring explanation for the outliers in the Hessian eigenspectrum and we acknowledged that in the related works section. However, although we managed to conduct the experiment described in the paper on our experiment networks, the results do not agree with the explanations in the paper. We have shown that the clustering explanations do not apply to our networks at minima (**Fig. 21**). The discussion of the experiment results are in **Appendix E.1**. We added a brief discussion in **Section 4.4** in the revised version. Also note that we did not provide a new explanation for the outliers. Instead, we connected this phenomenon to the output Hessian matrix $\mathbb{E}[\mathbf{M}]$ instead.
>
> 4. We add [4] to the related works (**Section 2**). The part of this paper related to our work is mainly identical to what presented in [3], only with more details. Similarly, experiments in this paper only calculate the eigenspectrum but not the eigenspace. Contrary to what you suggest, this paper does not claim that the Kronecker factorization (KFAC) [6] is not a good approximation. It only shows that the approximation can be improved using class-distinct factorization (CFAC). However, CFAC requires a much larger space complexity to calculate and thus cannot be implemented in our experiments. We consider KFAC is a good approximation for our experiments and we think our empirical evidence is sufficient.
>
> **References**
>
> [1] Ghorbani et al. An Investigation into Neural Net Optimization via Hessian Eigenvalue Density. ICML 2019. https://arxiv.org/abs/1901.10159
>
> [2] Papyan. The Full Spectrum of Deepnet Hessians at Scale: Dynamics with SGD Training and Sample Size. 2018. https://arxiv.org/abs/1811.07062
>
> [3] Papyan. Measurements of Three-Level Hierarchical Structure in the Outliers in the Spectrum of Deepnet Hessians. ICML 2019. https://arxiv.org/abs/1901.08244
>
> [4] Papyan. Traces of Class/Cross-Class Structure Pervade Deep Learning Spectra. 2020. https://arxiv.org/abs/2008.11865
>
> [5] Golmant et al. pytorch-hessian-eigentings: efficient PyTorch Hessian eigendecomposition. 2018. https://github.com/noahgolmant/pytorch-hessian-eigenthings
>
> [6] Martens & Grosse. Optimizing Neural Networks with Kronecker-factored Approximate Curvature. ICML 2015. https://arxiv.org/abs/1503.05671

---

> > ### Comment · AnonReviewer4 · 2020-11-24
> > **Author Response still shows inability  to assimilate and give proper acknowledgement of prior contributions by others.**
> >
> > 1.  I am not the author of any of the papers cited or discussed.   I am dismayed that the authors make so little attempt to value the prior and much more extensive and insightful work of others.
> >
> > 2. Typical empirical measurements in this paper concern a 22-year-old network architecture that is seriously behind the times. So much so that experiments based on this cannot be considered evidentiary for making statements about modern neural nets. Moreover, the recent work of Ghorbani et al, of Papyan and by now of several others has now shown how to scale up eigencomputations to the scale of modern neural nets having millions of parameters.  The computations being performed in this paper are of such tiny scale in comparison to the computations that would be required for modern deepnets, and the model is so basic and small scale that there is serious  reason to doubt they reflect any important properties  of the modern field of deep nets and representation learning.
> >
> > 3. The authors claim that because I mention the Hessian spectrum I ignore their contributions. They  continue their response saying their paper discuss the top eigenspace structure, as if therefore my references don't apply. In fact the papers I refer to, such as [3]and [4], give very detailed information about the eigenspace structure.  It  seems to me that the authors' claims are either uninformed or just believe in naked  warrantless assertion.
> >
> > 4.  The authors say in their response:
> >
> > "We agree with you that [1] presented the phenomenon that Batch Normalization suppresses the outliers in the Hessian eigenspectrum. We would like to thank you for making this comment. We have made it clear in our revised version. We would also like to note that we provided an explanation for this phenomenon but [1] did not."
> >
> > [3] appeared before this paper, and  provides a clear argument for why batchnorm suppresses outliers in the spectrum of the Hessian. The authors ought to mention that reference as well. At any rate, the authors' claim to be making a contribution additional to [1] and [3] is unsubstantiated. It is just mere assertion.
> >
> > Morever, to me the so-called  author's "explanation" is just hand-waving. A proper explanation would involve showing a mechanism that specifically connects the dynamics of training with and without batch norm with the eigenproperty they wish to discuss. I don't see anything in this paper that remotely qualifies as such a demonstration.
> >
> > 5. The authors say:
> >
> > "Contrary to what you suggest, we do not think the technology presented in [2] can be applied to the experiments in our paper. This is because our experiments requires Hessian eigenvectors, but this technology cannot calculate eigenvectors. We use Hessian Eigenthings package [5] which is more suitable for our calculations. The only relation between [2] and our work is that [2] shows the existence of outliers in Hessian eigenspecturm, and we have clearly acknowledged that in our paper."
> >
> >  To make this clear, it is as if the authors were trying to tell me that the power method can calculate eigenvalues but not eigenvectors. Every textbook shows that the power method does both, inherently. Please! stop asserting nonsense.
> >
> > In fact the underlying Lanczos method used in [2] is the same method used in this paper!
> > 1. [2] implements the Lanczos iteration in PyTorch which can use GPUs, and Lanczos can obviously approximate eigenvectors.
> > 2. The authors use a package by Mahoney and co-authors that simply invokes the Lanczos implementation of SciPy (linear algebra package in Python). Therefore the authors also use Lanczos, just implemented on slow CPUs instead of extremely fast GPUs.
> >
> > 6. The authors say:
> >
> > "[3] did provide an inspiring explanation for the outliers in the Hessian eigenspectrum and we acknowledged that in the related works section. However, although we managed to conduct the experiment described in the paper on our experiment networks, the results do not agree with the explanations in the paper. We have shown that the clustering explanations do not apply to our networks at minima (Fig. 21). The discussion of the experiment results are in Appendix E.1. "
> >
> > Alternative interpretation: [3] studied modern networks at scale. The authors did not. The authors did not observe the phenomena that [3] observed at scale.
> >
> > The implication, for me, of this statement, is simply this: the author's experiments simply don't describe the phenomena modern deepnets.
> >
> > Appendix E of their revised paper contains an experiment - for LeNet5, a 22 year-old architecture.  On this small antiquated example, the logit derivatives don’t cluster well -- as [3] would suggest anyway, in this special case. This simply cements the irrelevance to modern deep learning of the evidence and the arguments adduced by the authors.  Multiple teams around the world have observed phenomena at scale in massive modern networks that that these authors can't produce on their toy problems. Should we be surprised? Interested?

---

> > > ### Author Response · Authors · 2020-11-24
> > > **Response to Reviewer 4: This review is purposely attacking this paper with unjustified claims - Part 2**
> > >
> > > ### 4. On Methods Used for Eigenvector Computation
> > >
> > > To address the confusion on algorithms in [2], our previous response is based on the assumption that the reviewer was referring to the full eigenspecturm estimation algorithm, which cannot be used for computing information of the eigenbasis. We were not discriminating the two Lanczos algorithm since both algorithms shares the same bottleneck in our setting as described below.
> > >
> > > FastLanczos algorithm [2], just as the original Lanczos algorithm and any other algorithms, shares the same bottleneck in our setting: the top eigenbasis for large networks are too large to be stored and analyzed. Note that storing 3 eigenvectors of the Lanczos tridiagonal matrix already takes more space than storing the tridiagonal matrix itself. As the scale of the network goes up, so does its output widths, and hence the dimension of the top eigenspace that we care about. (e.g. storing the top 500 eigenvectors of ResNet-18, which has approximately 11 million parameters, takes at least 22 GB in memory).
> > >
> > > The bottleneck for computation time for both FastLanczos and the original Lanczos in our setting is in fact the action of Hessian on vectors, for which both implementations use autograd on GPUs. In our experience the runtime of two algorithms does not make a big difference in our setting. Since FastLanczos takes a further approximation step which may introduce more uncertainty, we did not choose to use it in our work.
> > >
> > > ### 5. Gradient Clustering of [3] and [4] may not Generally Apply
> > >
> > > Based on the open-source code [7] provided by the author of [3] and [4]. We have reproduced the experiment in [3] and [4] using our experiment networks (fc networks, LeNet5, and VGG11) and network used in [3] and [4] (ResNet-18).
> > >
> > > Following the default hyperparameter setting of the author, the results of very significant clustering was successfully reproduces when ResNet-18 is trained with a weight-decay of 0.0005 ([Image 1](https://i.ibb.co/M1cYGXk/CIFAR10-Exp1-Resnet18-papyan-net-fixlr0-01-wd0-0005-m0-9-E-1-delta-tsne.jpg)). But the clustering is significantly weakened to a level comparable to **Fig. 21** when we remove the weight-decay regularization ([Image 2](https://i.ibb.co/dPXnpg4/CIFAR10-Exp1-Resnet18-papyan-net-fixlr0-01-E-1-delta-tsne.jpg)).
> > > Similarly, the clustering does not work well for VGG11 trained without weight-decay ([Image 3](https://i.ibb.co/D97ZRBy/VGG11.jpg)) and the approximation of eigenvalues failed consequently ([Image 4](https://i.ibb.co/3d410xT/VGG11-approx.jpg)). Here both models were trained to convergence on CIFAR-10 with comparable accuracy.
> > >
> > > If the theory in [3] is well-established, we would expect the second and third figure to be similar to the first one instead of **Fig. 21**. Hence the limitation of the current gradient clustering theory cannot be simply explained by the difference between large and small models.
> > >
> > > Since this observation is far from the main focus of our work, we did not include it in the paper, but we would like to show it here regarding the reviewer's questions. Why weight-decay may cause this difference is not clear and is an interesting open problem, but that also deviates from our analysis based on the decoupling conjecture.
> > >
> > > **References**
> > >
> > > [1] Ghorbani et al. An Investigation into Neural Net Optimization via Hessian Eigenvalue Density. ICML 2019. https://arxiv.org/abs/1901.10159
> > >
> > > [2] Papyan. The Full Spectrum of Deepnet Hessians at Scale: Dynamics with SGD Training and Sample Size. 2018. https://arxiv.org/abs/1811.07062
> > >
> > > [3] Papyan. Measurements of Three-Level Hierarchical Structure in the Outliers in the Spectrum of Deepnet Hessians. ICML 2019. https://arxiv.org/abs/1901.08244
> > >
> > > [4] Papyan. Traces of Class/Cross-Class Structure Pervade Deep Learning Spectra. 2020. https://arxiv.org/abs/2008.11865
> > >
> > > [5] Golmant et al. pytorch-hessian-eigentings: efficient PyTorch Hessian eigendecomposition. 2018. https://github.com/noahgolmant/pytorch-hessian-eigenthings
> > >
> > > [6] Martens \& Grosse. Optimizing Neural Networks with Kronecker-factored Approximate Curvature. ICML 2015. https://arxiv.org/abs/1503.05671
> > >
> > > [7] Papyan. DeepnetHessian. 2018. https://github.com/deep-lab/DeepnetHessian

---

> > > ### Author Response · Authors · 2020-11-24
> > > **Response to Reviewer 4: This review is purposely attacking this paper with unjustified claims - Part 1**
> > >
> > > We would again like to point out that this review ignores most of our contributions and only focus on the Hessian eigenspectrum under the theory of gradient clustering. The reviewer also has inaccurate understanding of previous works mentioned in the review, and made wrong claims on them.
> > >
> > > ### 1. [3] and [4] has Little Analysis on Eigenvectors of Hessian
> > > The reviewer claim that [3] and [4] has given "detailed information about the eigenspace structure."
> > > However both papers contains little information on the structure of the eigenvectors except some direct consequences of the structure of the eigenspectrum based on the theory of gradient clustering. [4] does include an analogue of the kronecker approximation of layerwise Hessians in its Section 7.2-7.3, but it did not do further analysis on the eigenspace of the two components.
> > >
> > > The theory of gradient clustering as proposed in [3][4] mainly focus on the structures for the top $C-1$ and $C(C-1)$ eigenvalues of the eigenspectrum ($C$ denoting the number of classes). This result is very different from our main result on the decoupling conjecture and the dominate eigenspace overlap.
> > >
> > > Moreover, we observed that the gradient clustering phenomenon which [3] and [4] builds upon with does not uniformly applies to our experiment models (including VGG11) and some large models (e.g. ResNet-18 as used in [3]). We will discuss this in detail in section 5 of this reply.
> > >
> > > ### 2. [3] has no Analysis on Batch Normalization
> > > Contrary to what the reviewer has claimed, [3] has no analysis on Batch Normalization (BN) at all rather than "providing a clear argument for why batchnorm suppresses outliers in the spectrum of the Hessian." We did found some explanations for the outlier suppression in [4] and acknowledged it in our revised paper.
> > >
> > > As emphasized in the introduction and the conclusion, the focus of this work is on the analysis of the structure of the eigenspaces at the minima. The main focus of our comments on BN is to show that BN affects the Hessian eigenspace structures we observed, instead of the suppression of outliers alone. Under this setting, we believe that our approximation model and supportive empirical results are sufficient to justify our claim.
> > >
> > > Also, the fact that [1] and [4] mentioned and explained related phenomena does not invalidate our contribution. Compared to these 2 papers, we observed and explained the effect of BN on Hessian eigenspace structures and the decoupling conjecture, which is more comprehensive than the suppression of outliers alone.
> > >
> > > It is unclear why the reviewer insists the explanation for a property of the minima must "involve showing a mechanism that specifically connects the dynamics of training with and without batch norm". Clearly, the explanations in [4] also does not satisfy this criteria.
> > >
> > > ### 3. Our Result can Generalize on Large Networks
> > > Since eigenspace information for large networks are not computationally available due to spatial complexity (will be elaborated in Section 4 of this reply), for most of the empirical studies we focused our analysis on the smaller network structures. In **Appendix D.3** we have also shown supportive evidence regarding our claim on the effect of $\mathbb{E}[\mathbf{xx}^T]$ on the layer-wise Hessian by examples of the first several layers of VGG11, which is a commonly recognized large network structure.
> > >
> > > Also, given that the strong clustering phenomenon in [3] and [4] does not apply uniformly to large models (see section 5 of this reply), it is unjustified to assume our explanations only apply for small networks simply because it does not reproduce this clustering effect of [3] and [4].

---

### Official Review · AnonReviewer2 · 2020-11-05
**Review for "Dissecting Hessian: Understanding Common Structure of Hessian in Neural Networks"**

**Rating:** 4
**Confidence:** 2

**Review:**

## Summary
The paper studies the empirical properties of the Kronecker Factorization based approximation to the Hessian of the loss. The paper makes the following two empirical observations:
1. Top eigenspace of the layer-wise hessian has a high overlap for neural networks trained with different initializations and hyper-parameter setting.
2. Top eigenvectors of the layerwise hessian as a matrix is approximately rank 1.
Paper makes the decoupling conjecture which states that $\nabla^2_{z^{(p)}}\ell(z, y)$, which is Hessian wrt loss evaluated at the output of layer p, is independent of $x_{(p)}x_{(p)}^T$, correlation of the input to the layer $p$. Under this assumption, the paper gives an explanation of why the above-stated phenomenon holds. Finally, the paper uses this approximation to optimize the PAC-Bayes bounds for the stochastic classifier where the eigenbasis of the proposed layer-wise Hessian is used for the covariance of the posterior.

## Evaluation

Overall, the paper is not clearly written. It really needs to improve on the notation and needs to explain the ideas more clearly. Starting from Eq(3) in the paper, we are computing a gradient wrt $v$ for the hessian, but it’s really not clear from the notation how the output of $f_\theta(x)$ is dependent on $v.$ Writing this precisely would ease the burden on the reader.

Having said that authors do make new empirical observations with regards to the layer-wise hessian approximation but they don’t do a good job in explaining why these observations are relevant in a more general context. I have some questions for the authors to help me evaluate this work.

1. How close is the layer-wise Hessian approximation to the true Hessian? There is no discussion in the paper about the approximation.
2. A majority of the paper hinges on the decoupling conjecture. How close are the matrices in Eq(6) in terms of the actual norm on matrices? In the paper, we’re only given visual proof for the closeness in terms of the eigenvector correspondence matrices. Honestly,  I had a hard time following how it was computed.
3. For the PAC-Bayes bound, how the posterior variance is parameterized? It’ll be useful to give a mathematical characterization of it. Is the prior variance wrt standard basis still? If so, how is the KL computed? I am happy to just see the expression because couldn't find it in the Appendix.

Minor comments:
Please review the citations added in references. For many of the papers, only Arxiv posts are cited whereas the papers are published at other venues.

---

> ### Author Response · Authors · 2020-11-21
> **Response to Reviewer 2 with our answers to the questions**
>
> Dear reviewer, thank you for your time and comment. We highly appreciate your constructive advice. Following you suggestions, we modified the preliminaries section for clearer notations. We also rewrote section 4.2. for a clearer definition of the correspondence matrices, and modified the Appendix F for PAC-Bayes bound. Hope that the current version will resolve some confusions in the previous one.
>
> Please see below for answers to your specific questions and concerns:
>
> 1. How close is the layer-wise Hessian approximation to the true Hessian?
>
>   In this work, we are not claiming any results on the full Hessian matrix. We focus on the analysis of the layer-wise Hessians. We believe the layer-wise Hessians by themselves already captures important information of the loss landscape. Previous works such as the K-FAC paper [1] has shown that layer-wise Fisher Information matrix alone can be used for optimization. The improved PAC-Bayes bound we proposed also demonstrated so. We listed the potential insights of our work in the general response.
>
>
> 2. How close are the matrices in **Eq (6)** in terms of the actual norm on matrices? How is the approximation computed?
>
>   Since the weight Hessian matrix of a layer is too large to be explicitly computed and stored, we do not have direct access to the norm of the difference between the true layer-wise Hessians and the approximated Hessians. Thus we can only compare them using their eigen-information as shown in **Fig. 2** and **Fig 3**. Note that the spectral norm of the two matrices are presented as the first eigenvalue in **Fig. 2(a)**.
>
>   The computation of the true layer-wise Hessian eigenvalues and eigenvectors are stated in **Appendix C**. The computation of approximation using the Decoupling Conjecture (eigen-information of $\mathbb{E}[\mathbf{M}]$ and $\mathbb{E}[\mathbf{xx}^T]$) can be done directly using SVD in PyTorch.
>
>
> 3. Questions on the PAC-Bayes bound
>
>   The variance of posterior $Q = \mathcal{N}(\mathbf{w}, \text{diag}(\mathbf{s}))$ is parameterized by $\text{diag}(\mathbf{s})$ in the eigenbasis. Here $\mathbf{s}$ is initialized as $\mathbf{\theta}^2$ (element-wise square of the parameter vector) and is updated through the optimization process. In the pseudocode Algorithm1 in **Appendix F**, we represented $\mathbf{s}=\exp(2\mathbf{\varsigma})$ where $\mathbf{\varsigma}$ is being updated in each iteration on Algorithm1:13.
>
>     For the prior $P = \mathcal{N}(\mathbf{\theta}_0, \lambda\mathbf{I}_p)$, since we are using a spherical Gaussian variance, the variance is invariant through the change of basis. In our implementation the KL-divergence is computed in the eigenbasis.
>
>   We have added the descriptions of the prior in **Section 6** and how we compute the KL divergence in **Appendix F**. Hopefully the revised version can be clearer and help your understanding.
>
> **References**
>
> [1] Martens & Grosse. Optimizing Neural Networks with Kronecker-factored Approximate Curvature. ICML 2015. https://arxiv.org/abs/1503.05671

---

### Author Response · Authors · 2020-11-18
**Revision and General Comment**

Dear reviewers, thank you for your time and constructive feedbacks and comments. Following your advice, we updated the first revision of the paper. In this revision, we have made the following major changes:

* Modified **Section 1** (Introduction) to state our contribution clearer.
* Added **Section 7** (Conclusion), which highlights our contribution and discuss about potential insights.
* Rewrote **Section 4.2** (Eigenvector Correspondence) to give a clearer definition of the eigenvector correspondence matrices as well as providing some intuitive interpretations.
* In **Section 6** (PAC-Bayes Bounds),  added descriptions for the prior used in the calculation PAC-Bayes bound and a figure showing the optimized posterior variance.
* In **Appendix F** (PAC-Bayes Bounds), added descriptions for the prior and how the KL divergence is computed on the eigenbasis.
* In **Section  5.2** (Batch  Normalization), acknowledged previous works [1][2] on discovering the phenomenon of BN on eigenspectrum.
* In **Appendix E.1** (Additional Explanations), acknowledged previous related works on class-centered gradient clustering and alignment.
* Moved comments on related previous works from the Appendix to **Section 4.4** (Hessian Eigenspectrum).
* Refined the caption of **fig.3** and **fig.4** to describe the figures more clearly.
* In **Section 2** (Related Works), added citations [2][3].
* Fixed the bibliography for published works.

This revision should address some of the questions and confusions in the reviews. We will post individual responses by this week. We respectfully disagree with some of the comments in the reviews and we will also address them in the individual responses.

---

In addition, we would like to again highlight our contribution and potential insights, which may not be clear from the original version of the paper. These information are also included in the modified introduction and newly added conclusion section.

### Main Contributions
1. We discovered two surprising structures in the top eigenspace of layer-wise Hessian for neural networks.
2. The top eigenspace of layer-wise Hessian for different models have a non-trivial high overlap which peaks at the dimension of the layer's output dimension (number of neurons / channels).
3. Top eigenvectors of layer-wise Hessian form low rank matrices when they are reshaped into the same shape as the corresponding weight matrix.
4. Under a decoupling conjecture, we show that these structures can be explained by Kronecker factorization.
5. As a proof of concept, we showed that the structure of the layer-wise Hessian can be used to find a better explicit generalization bounds with PAC-Bayes techniques.
6. We relate the eigenspectrum of the layer-wise weight Hessian to the eigenspectrum of the layer-wise output Hessian $\mathbb{E}[\mathbf{M}]$.
7. We provided an explanation for the effect of batch normalization on our two Hessian structures.

### Potential Insights
1. From a theoretical perspective, we hope this work would be a starting point towards formally proving the structures in the Hessians of neural networks (in particular, why the decoupling conjecture is correct and why the output Hessian matrix is often low rank).
2. From a generalization perspective, it is a well known empirical result that the sharpness of minima is well correlated with its generalization capability[4][5]. Since the sharpest directions are just the basis for the dominating eigenspace of the Hessian. From a theoretical perspective, understanding the structure of this low dimensional subspace would provide valuable insight to the analysis on minima sharpness. (e.g. batch normalization is suppressing the $\mathbb{E}[\mathbf{xx}^T]$ term)
3. From an optimization perspective, there has been literature showing that the dominating eigenspace has a high overlap with the optimization trajectory of gradient descent [6]. The similarity of eigenspaces of different models may also imply the similarity between different optimization trajectories within some structured subspace. For optimization it is potentially easier to leverage the eigenspace overlap structure as it shows that the subspace is preserved across different models.

**References**

[1] Ghorbani et al. An Investigation into Neural Net Optimization via Hessian Eigenvalue Density. ICML 2019. https://arxiv.org/abs/1901.10159

[2] Papyan. Traces of Class/Cross-Class Structure Pervade Deep Learning Spectra. 2020. https://arxiv.org/abs/2008.11865

[3] Fort et al. Stiffness: A New Perspective on Generalization in Neural Networks. 2019. https://arxiv.org/abs/1901.09491

[4] Keskar et al. On large-batch training for deep learning: Generalization gap and sharp minima. ICLR 2017. https://arxiv.org/abs/1609.04836

[5] Jiang et al. Fantastic generalization measures and where to find them. ICLR 2020. https://arxiv.org/abs/1912.02178

[6] Gur-Ari et al. Gradient descent happens in a tiny subspace. 2018. https://arxiv.org/abs/1812.04754

---

### Decision · Program_Chairs · 2021-01-07
**Final Decision**

**Decision:**

Reject

**Comment:**

This paper studies different properties of the top eigenspace of the Hessian of a deep neural network and their overlap. It raised quite a lot of discussion, which finally went in not very constructive way. The reviewers generally agree that the paper has potential, but the actual contribution is limited.

Pros:
- The idea that top eigenspaces between different models have high overlap is interesting
- The explanation that these structures can be explained by Kronecker-product approximation of the Hessian.

Cons:
- The connection to PAC-Bayes is unclear and seems artificial.
- Many of the related work is missing
- The models and datasets are too simple, and general conclusions can not be made on such kind of models. Much more testing is needed to verify the claims, including state-of-the art architectures and datasets.